# In vitro reconstitution of epigenetic reprogramming in the human germ line

Yusuke Murase[1,2,6], Ryuta Yokogawa[1,2,6], Yukihiro Yabuta[1,2], Masahiro Nagano[1,2], Yoshitaka Katou[1,2], Manami Mizuyama[1,2], Ayaka Kitamura[1,2], Pimpitcha Puangsricharoen[1,2], Chika Yamashiro[2], Bo Hu[1,2], Ken Mizuta[1,2], Taro Tsujimura[1], Takuya Yamamoto[1,3,4], Kosuke Ogata[5], Yasushi Ishihama[5] & Mitinori Saitou[1,2,3 ✉]

Epigenetic reprogramming resets parental epigenetic memories and differentiates primordial germ cells (PGCs) into mitotic pro-spermatogonia or oogonia. This process ensures sexually dimorphic germ cell development for totipotency[1]. In vitro reconstitution of epigenetic reprogramming in humans remains a fundamental challenge. Here we establish a strategy for inducing epigenetic reprogramming and differentiation of pluripotent stem-cell-derived human PGC-like cells (hPGCLCs) into mitotic pro-spermatogonia or oogonia, coupled with their extensive amplification (about >$10^{10}$-fold). Bone morphogenetic protein (BMP) signalling is a key driver of these processes. BMP-driven hPGCLC differentiation involves attenuation of the MAPK (ERK) pathway and both de novo and maintenance DNA methyltransferase activities, which probably promote replication-coupled, passive DNA demethylation. hPGCLCs deficient in TET1, an active DNA demethylase abundant in human germ cells[2,3], differentiate into extraembryonic cells, including amnion, with de-repression of key genes that bear bivalent promoters. These cells fail to fully activate genes vital for spermatogenesis and oogenesis, and their promoters remain methylated. Our study provides a framework for epigenetic reprogramming in humans and an important advance in human biology. Through the generation of abundant mitotic pro-spermatogonia and oogonia-like cells, our results also represent a milestone for human in vitro gametogenesis research and its potential translation into reproductive medicine.

Germ cells give rise to totipotency and ensure heredity and evolution. Human PGCs are thought to be specified at around embryonic day 12 (E12) to E16 (2 weeks post-fertilization (w.p.f.)) in the amnion or the posterior epiblast of early post-implantation embryos[4,5]. They migrate through the yolk sac and hindgut endoderm, colonizing genital ridges from around the 5–6 w.p.f. time frame[6]. During this period, human PGCs initiate epigenetic reprogramming, resetting parental epigenetic memories through genome-wide DNA demethylation (5-methylcytosine (5mC) demethylation) and histone modification remodelling[3,7]. By around 7–8 w.p.f., human PGCs complete the reprogramming process and differentiate into either mitotic pro-spermatogonia or oogonia, precursors for spermatogonia or oocyte differentiation, respectively[3,7,8] (Fig. 1a).

In vitro gametogenesis (IVG) from pluripotent stem (PS) cells provides a framework for clarifying the mechanism of germ cell development[9]. Accordingly, human PS cells have been induced into hPGCLCs[10–12], which, after aggregation culture with mouse embryonic testicular or ovarian somatic cells (xenogeneic reconstituted testes (xrTestes) or ovaries (xrOvaries), respectively), undergo epigenetic reprogramming and differentiate into either pro-spermatogonia or oogonia-like cells, respectively[12–14]. However, the xrTestis and xrOvary systems for hPGCLC differentiation are low in efficiency and bear limited capacity for scaling and experimental control. Although hPGCLCs can be co-cultured with human hindgut organoids for differentiation, this also achieves only limited differentiation[15]. Thus, to explore the mechanism for hPGCLC differentiation and accelerate human IVG, a more robust methodology is required. Here we embarked on establishing a system for signalling-molecule-driven hPGCLC differentiation.

## Signalling for hPGCLC differentiation

hPGCLCs cultured on m220 feeder cells under reported conditions propagate as early PGCs but tend to de-differentiate and require cell sorting for passage[12] (Extended Data Fig. 1a). We sought to determine a condition that minimizes de-differentiation. Human induced pluripotent stem (iPS) cells bearing *BLIMP1* (also known as *PRDM1*)–*tdTomato* (*BT*) and *TFAP2C*–*eGFP* (*AG*) alleles (585B1 *BTAG* (XY): M1-*BTAG*) (Fig. 1b) were induced into incipient mesoderm-like cells (iMeLCs) and then into BT⁺AG⁺ hPGCLCs, which were cultured and passaged around every 10 days by cell sorting using flow cytometry.

[1]Institute for the Advanced Study of Human Biology (ASHBi), Kyoto University, Kyoto, Japan. [2]Department of Anatomy and Cell Biology, Graduate School of Medicine, Kyoto University, Kyoto, Japan. [3]Center for iPS Cell Research and Application (CiRA), Kyoto University, Kyoto, Japan. [4]Medical-Risk Avoidance based on iPS Cells Team, RIKEN Center for Advanced Intelligence Project (AIP), Kyoto, Japan. [5]Department of Molecular Systems BioAnalysis, Graduate School of Pharmaceutical Sciences, Kyoto University, Kyoto, Japan. [6]These authors contributed equally: Yusuke Murase, Ryuta Yokogawa. ✉e-mail: saitou@anat2.med.kyoto-u.ac.jp

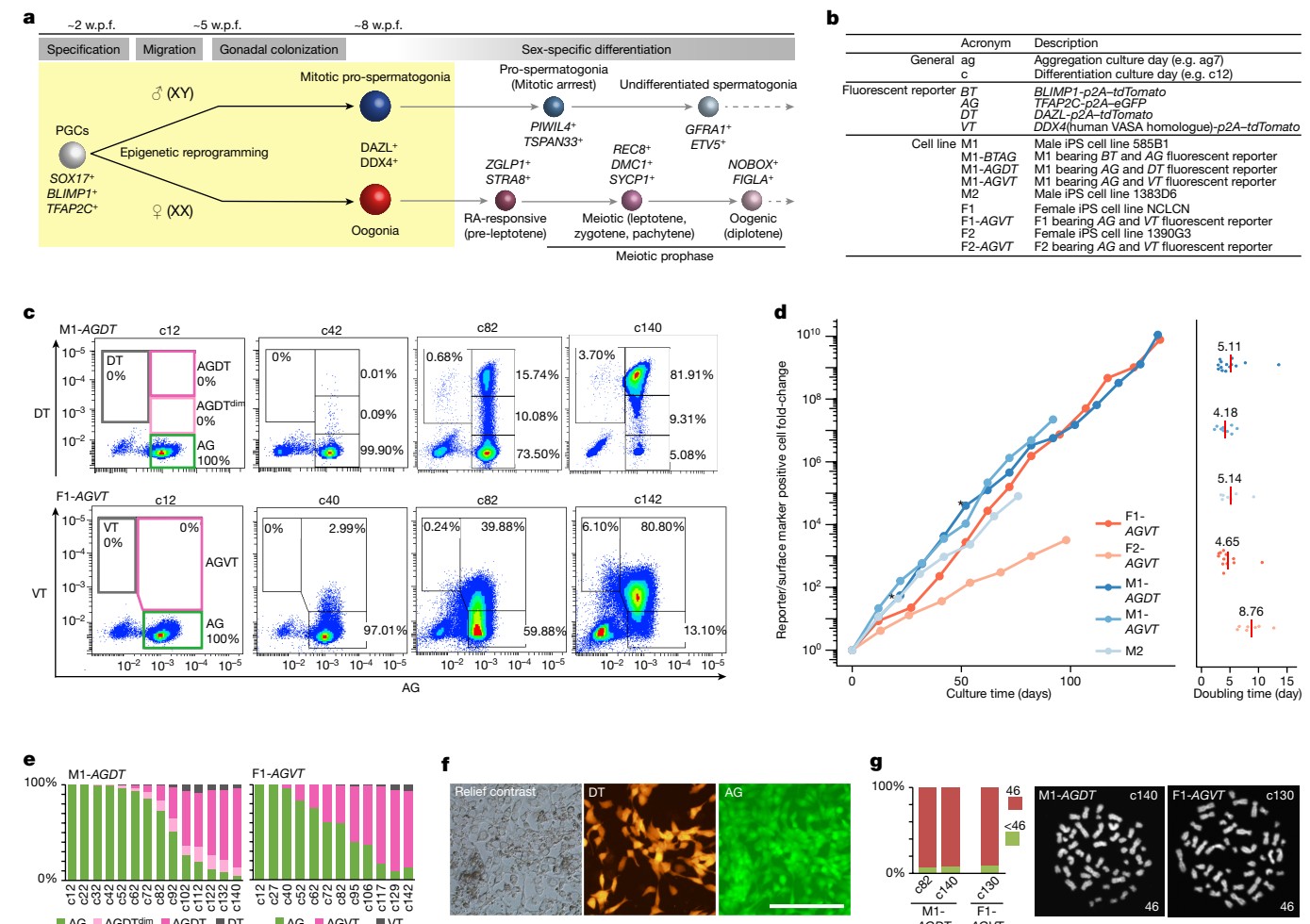

**Fig. 1 | BMP signalling promotes hPGCLC differentiation. a**, Schematic of human germ cell development. Differentiation stages, key markers and stages covered by this study (yellow) are shown. **b**, Summary of the acronyms used in this study. **c**, Flow cytometric analysis of the expression of AG and DT or VT during BMP-driven M1-*AGDT* or F1-*AGVT* hPGCLC differentiation on the indicated culture days. Percentages of the cells in each gate are shown. The data represent four (M1-*AGDT*) and eight (F1-*AGVT*) biological replicates. **d**, Growth curve (left) and doubling time (right) of hPGCLC-derived cells induced from the indicated human iPS cell lines. The number of hPGCLC-derived cells was calculated as the sum of reporter⁺ cells or EpCAM⁺ITGA6⁺ cells (for M2). For the doubling time, each dot represents a doubling time for one passage interval and the red bar represents the average of all passage intervals. Asterisk indicates cells passaged

by flow cytometry. Colour coding is as indicated. The data represent four (M1-*AGDT*), two (M1-*AGVT*), two (M2), eight (F1-*AGVT*) and two (F2-*AGVT*) biological replicates. **e**, Proportion of cells with the indicated reporter expression during BMP-driven M1-*AGDT* or F1-*AGVT* hPGCLC differentiation on the indicated culture days. The data represent four (M1-*AGDT*) and eight (F1-*AGVT*) biological replicates. **f**, Relief contrast and fluorescence (DT and AG) images of M1-*AGDT* hPGCLC-derived cells at c72. The images represent four biological replicates. Scale bar, 200 μm. **g**, Karyotype (left; percentage of cells with 46 or other chromosome numbers; right: chromosome spreads) of M1-*AGDT* and F1-*AGVT* hPGCLC-derived cells at the indicated culture days (one biological replicate at each time point).

Following passage, the culture consisted of three cell populations: BT⁺AG⁺ hPGCLCs; BT⁻AG⁻TRA-1-85⁺ (a human-specific antigen) de-differentiated cells; and TRA-1-85⁻ feeders (Extended Data Fig. 1a). The ratio of hPGCLCs to de-differentiated cells was estimated as the ratio of BT⁺AG⁺ cells to forward scatter^high cells (the hPGCLC enrichment score) (Extended Data Fig. 1a). Using this criterion, we evaluated the impact of inhibiting WNT, NODAL and BMP signals on hPGCLC de-differentiation. Inhibiting NODAL or BMP signalling resulted in a decrease in the enrichment score, whereas inhibiting WNT signalling with IWR1 dose-dependently increased the enrichment score (Extended Data Fig. 1b,c). We then examined the impact of basal medium on de-differentiation and found that advanced RPMI (advRPMI) led to increased enrichment scores (Extended Data Fig. 1d). Accordingly, hPGCLCs cultured with IWR1 and advRPMI exhibited vigorous propagation with high enrichment scores and could be passaged without sorting at least two times (Extended Data Fig. 1e).

During human PGC and hPGCLC differentiation, a set of genes for spermatogenesis and oogenesis are upregulated coupled with genome-wide DNA demethylation[3,13,14]. We determined the genes that showed both progressive upregulation during hPGCLC-to-oogonia-like cell differentiation and a ≥50% reduction in the promoter 5mC level from human iPS cell to oogonia-like cells (epigenetic reprogramming (ER)-activated genes)[14] (Extended Data Fig. 1f–h). Because ER genes such as *GTSF1*, *PRAME* and *MEG3* are upregulated early during hPGCLC differentiation, we screened signals that simultaneously upregulate these genes. hPGCLCs were cultured with IWR1 and advRPMI, together with relevant cytokines and chemicals, and the expression of the three genes was examined by quantitative PCR (qPCR) on culture day 22 (c22). The results showed that BMP ligands upregulated the three genes (Extended Data Fig. 1i,j). We reasoned that signals that induce human PGC differentiation should be active in tissues during PGC migration. Re-analyses of published single-cell RNA sequence (scRNA-seq) data for

human development[16–18] revealed that the relevant endoderm tissues expressed BMP family genes (Extended Data Fig. 2a–e).

We cultured hPGCLCs with progressively increasing doses of BMP2. hPGCLCs exhibited strong growth and high enrichment scores with low BMP2 doses (up to about 25 ng ml$^{-1}$), whereas their expansion was attenuated with high BMP2 (Extended Data Fig. 2f,g). We then cultured hPGCLCs with 25 ng ml$^{-1}$ BMP2 or without BMP2 until c55. hPGCLCs cultured with BMP2 continued to propagate with high enrichment scores, and immunofluorescence analysis revealed that a fraction of them expressed DAZL, a key ER gene product. By contrast, hPGCLCs cultured without BMP2 showed de-differentiation after around c32, and de-differentiated cells predominated at c55, with the remaining BT$^+$AG$^+$ cells barely expressing DAZL (Extended Data Fig. 2h). These findings suggest that BMP signalling stabilizes germ-cell fate and promotes epigenetic reprogramming and human PGC and hPGCLC differentiation into mitotic pro-spermatogonia or oogonia.

## BMP promotes hPGCLC differentiation

*DDX4* (human *VASA* homologue) and *DAZL* are two key genes that signify the differentiation of mitotic pro-spermatogonia or oogonia, and *DAZL* is expressed earlier and at a higher level[3,8,14,19] (Fig. 1a). We established human iPS cell lines bearing *AG* and *DAZL–tdTomato* (*DT*) or *DDX4–tdTomato* (*VT*) alleles (585B1 *AGDT* and *AGVT* (XY): M1-*AGDT* and M1-*AGVT*, respectively; NCLCN and 1390G3 *AGVT* (XX): F1-AGVT and F2-*AGVT*, respectively[14]) (Fig. 1b and Extended Data Fig. 3a–g).

First, we cultured hPGCLCs from M1-*AGDT* cells with 25 ng ml$^{-1}$ BMP2, and after the first passage, with three different doses of BMP2 (25, 100 or 200 ng ml$^{-1}$). Increasing the BMP2 dosage attenuated AG$^+$ cell expansion but accelerated the emergence of DT$^+$ cells (Extended Data Fig. 4a,b). We chose to culture hPGCLCs from this cell line first with 25 ng ml$^{-1}$ BMP2 and thereafter with 100 ng ml$^{-1}$ BMP2, with passage about every 10 days (a representative result is shown in Fig. 1c–e). hPGCLCs expanded stably and upregulated DT from about c32. Thereafter, the number of DT$^+$ cells progressively increased, and nearly all the cells became DT$^+$ at c140, with an overall expansion of at least 10$^{10}$-fold. By contrast, hPGCLCs cultured without BMP2 showed substantial degrees of de-differentiation after around c32, with AG$^+$ cells barely expressing DT at c42 (Extended Data Fig. 4c,d). During BMP-driven differentiation, hPGCLC-derived cells exhibited a spindle shape with an ovoid nucleus, formed loosely packed colonies with no clear AG$^+$DT$^-$ or AG$^+$DT$^+$ cell segregation, and were karyotypically normal (Fig. 1f,g). Under the same condition, hPGCLCs from M1-*AGVT* (Fig. 1b) expanded stably, upregulating VT from around c32, and nearly all the cells became VT$^+$ at c92, with an overall expansion of about >3 × 10$^7$-fold (Fig. 1d and Extended Data Fig. 4e,f). Similarly, hPGCLCs from M2 (1383D6 with no reporters) showed good expansion and differentiation into DDX4$^+$ cells, with an overall expansion of about 10$^5$-fold (Fig. 1b and Extended Data Fig. 4g–i). An orthogonal validation of DT and VT reporters revealed that DT and VT positivity are a quantitative indicator for DAZL and DDX4 expression, respectively. Conversely, about one-third of DT$^-$ cells at a late stage exhibited low-to-middle level DAZL expression, which is potentially due to sporadic transcriptional or post-transcriptional silencing of the *DT* allele (Supplementary Fig. 1 and Supplementary Discussion 1).

Next, we cultured hPGCLCs from F1-*AGVT* cells. These expanded with high enrichment scores at BMP2 concentrations higher than 50 ng ml$^{-1}$ (Extended Data Fig. 4j,k). We cultured them first with 50 ng ml$^{-1}$ BMP2 and thereafter with 100 ng ml$^{-1}$ BMP2, with passage every 10–15 days. hPGCLC-derived cells expanded stably and differentiated progressively into VT$^+$ cells with a normal karyotype, with an overall expansion of about 10$^{10}$-fold at c142 (Fig. 1c–e,g and Extended Data Fig. 4l). By contrast, hPGCLCs cultured without BMP2 showed poor expansion and substantial de-differentiation, with AG$^+$ cells barely expressing VT at c46 (Extended Data Fig. 4m,n). Under the same condition with BMP2, hPGCLCs from F2-*AGVT* cells exhibited a slower yet stable expansion

and progressively upregulated VT, with an overall expansion of about >3 × 10$^3$-fold (Fig. 1d and Extended Data Fig. 4o,p). Late in culture (about c82), we observed an emergence of AG$^-$DT$^+$ and AG$^-$VT$^+$ cells, which thereafter constituted a small fraction (5–10%) of the hPGCLC-derived cells (Fig. 1c,e and Extended Data Fig. 4e,f,o,p). Thus, BMP signalling reproducibly promotes hPGCLC differentiation from four independent human iPS cell lines, although their propagation and differentiation dynamics show line-dependent heterogeneity.

DT$^+$ and VT$^+$ cells expressed key ER genes and exhibited substantially reduced genomic 5mC levels (Extended Data Fig. 4q,r). Of note, hPGCLC-derived cells could be frozen and stored and were re-expandable after thawing (Extended Data Fig. 4s). They also propagated and differentiated into DT$^+$ or VT$^+$ cells without serum (Extended Data Fig. 4t,u). Although hPGCLC-derived cells occasionally differentiated into an AG$^-$DT$^-$ or AG$^-$VT$^-$ state, the sorted AG$^+$ cells propagated and differentiated into DT$^+$ or VT$^+$ cells (Fig. 1c,d). Collectively, these findings support the notion that BMP signalling stabilizes germ-cell fate, which promotes epigenetic reprogramming, hPGCLC differentiation into mitotic pro-spermatogonia or oogonia and their robust self-renewal.

## Transcriptome dynamics

We performed RNA-seq analysis (Supplementary Tables 1 and 2). hPGCLC-derived cells cultured with BMP2 maintained pluripotency and PGC gene expression, upregulated ER genes and some of the 13 genes reported to be upregulated in gonadal germ cells[19], but did not upregulate meiosis genes. These expression profiles were similar to those during oogonia-like cell differentiation in xrOvaries and in mitotic pro-spermatogonia or oogonia in vivo[3,14], and in male, but not female, hPGCLC-derived cells expressing Y-linked genes (Fig. 2a and Extended Data Fig. 5a). hPGCLC-derived cells expressed *CDH5* and *DMRT1*, markers for human germ cells from the migration stage onwards[20] (Extended Data Fig. 5b). Principal component analysis (PCA) revealed that BMP signalling enhanced the transcriptome maturation of hPGCLC-derived cells, which was similar to that of hPGCLC-derived cells in xrOvaries (Fig. 2b). Notably, c107 AG$^-$VT$^+$ cells were similar to AG$^-$VT$^+$ cells in xrOvaries (Fig. 2b), which show a retinoic acid (RA)-responsive, pre-leptotene state of the first meiotic prophase[14]. By contrast, BMP-driven and xrOvary-based hPGCLC showed distinct differentiation profiles when analysed with principal component 3. Moreover, genes upregulated during BMP-driven hPGCLC differentiation were enriched in gene ontology (GO) terms such as 'angiogenesis', whereas those for the xrOvary-based hPGCLCs were enriched for 'brain development' (Extended Data Fig. 5c).

We identified highly variable genes (HVGs) among hPGCLC-derived cells cultured with or without BMP2. The BMP upregulated genes included ER genes and were enriched for the GO terms 'negative regulation of transcription from RNA polymerase II promoter' and 'negative regulation of ERK1 and ERK2 cascade'. By contrast, the BMP downregulated genes were enriched for the terms 'positive regulation of transcription from RNA polymerase II promoter', 'cellular response to FGF stimulus', 'positive regulation of MAPK cascade' and 'positive regulation of ERK1 and ERK2 cascade' (Extended Data Fig. 5d–h and Supplementary Table 3). Accordingly, western blot analyses revealed that hPGCLC-derived cells cultured with BMP2 had reduced phosphorylated ERK levels (Extended Data Fig. 5i,j). Thus, BMP signalling promotes the upregulation of ER genes and attenuates MAPK (ERK) pathways.

Next, we performed scRNA-seq analysis of the female hPGCLC culture (F1-*AGVT*) and compared the results to oogonia and fetal meiotic oocyte development[21,22] (Supplementary Table 2). The analysed cells were classified into ten clusters and were annotated as follows: very early mitotic 1 (VEM1) and VEM2 (1 and 2 represent different cell cycle states); early mitotic 1 (EM1) and EM2; mitotic 1 (M1), M2 and M3; pre-leptotene and leptotene 1 (PLL1) and PLL2; and zygotene, pachytene

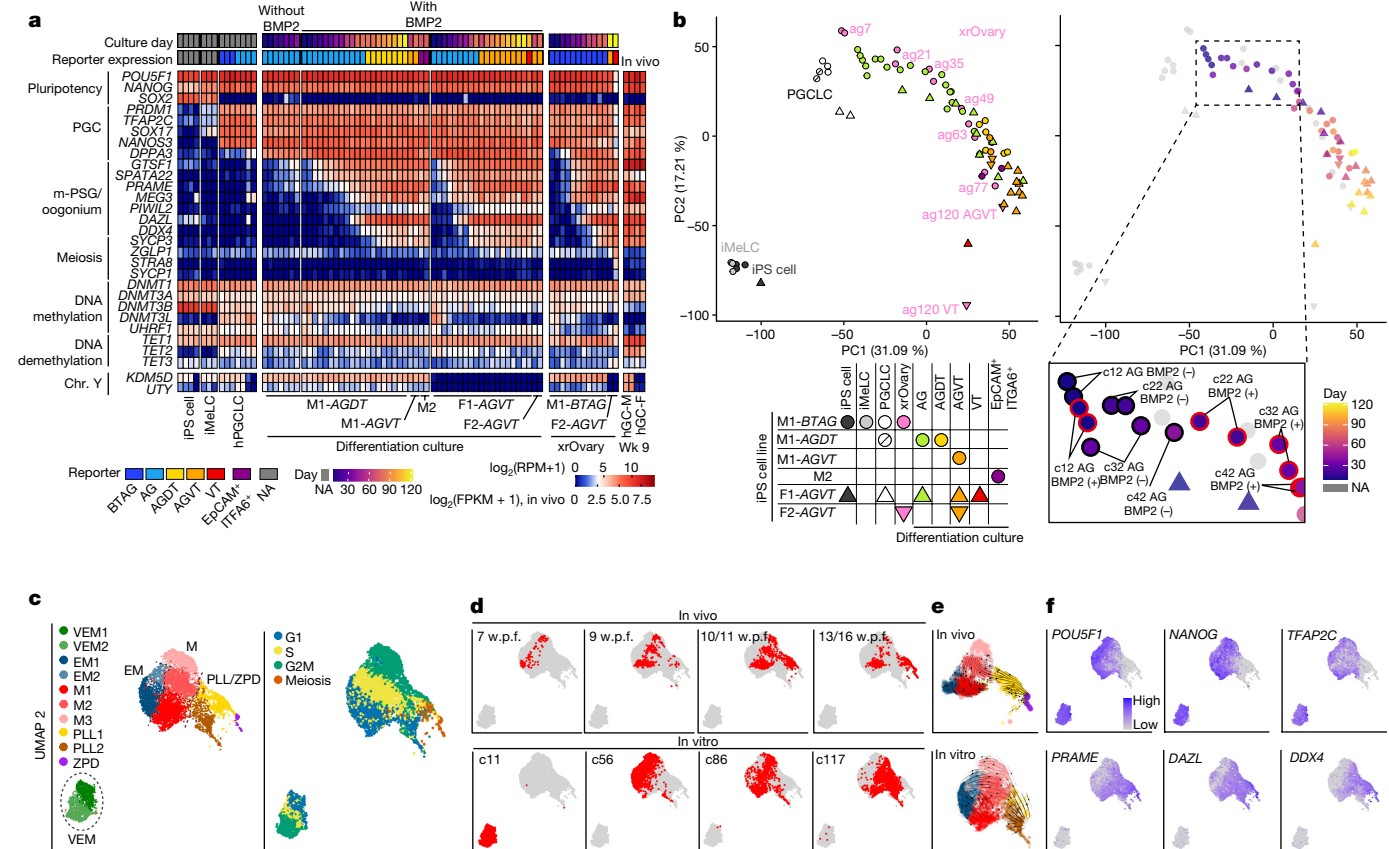

**Fig. 2 | Transcriptome dynamics during BMP-driven hPGCLC differentiation.**
**a**, Heatmap showing the expression levels of the indicated genes in the indicated cell types (see Supplementary Table 2 for full sample information). Colour coding is as indicated. FPKM, fragments per kilobase million; NA, not applicable; RPM, reads per million. **b**, PCA of transcriptomes during hPGCLC induction and BMP-driven or xrOvary-based hPGCLC differentiation. The left and right panels are colour coded with reporter expression and culture days, respectively, as indicated. The dotted area in the chart on the right is magnified to clarify the difference of transcriptome progression between cultures with BMP2 (BMP2 (+))

and without BMP2 (BMP2 (−)). **c**, Uniform manifold approximation and projection (UMAP) and Louvain clustering of scRNA-seq data of female germ cells at 7–16 w.p.f. in vivo[21,22] and F1-*AGVT* hPGCLC-derived cells at c11, c56, c86 and c117 in vitro. Cell-type (left) and cell cycle (right) annotations are shown. VEM, very early mitotic; EM, early mitotic; M, mitotic; PLL, preleptotene and leptotene; ZPD, zygotene, pachytene and diplotene. **d**–**f**, UMAP plots as in **c**, with the annotation of in vivo and in vitro samples (**d**), with potential developmental trajectories of in vivo and in vitro cell types analysed by RNA velocity (**e**), and with expression levels of the indicated genes (**f**). Colour coding is as indicated.

and diplotene (ZPD) (Fig. 2c and Extended Data Fig. 6a). VEM cells consisted nearly exclusively of AG⁺VT⁻ cells at c11. EM cells consisted primarily of AG⁺VT⁻ cells at c56 and of a small number of cells at 7–10 w.p.f. that began to upregulate oogonia markers. Mitotic cells constituted the major cell population and consisted of AG⁺VT⁺ cells at c56–c117 and oogonia at 7–16 w.p.f. in vivo, thereby representing the self-renewing oogonia. PLL cells consisted of AG⁻VT⁺ cells at c86–c117 and cells at 9–16 w.p.f. ZPD cells consisted of cells at 13–16 w.p.f. in vivo (Fig. 2c–f and Extended Data Fig. 6b,c). For mitotic cells, in vivo and in vitro cells contributed to all subclusters (M1: G1/S; M2: S/G2/M; M3: G2/M) and the differentially expressed genes (DEGs) between in vivo and in vitro cells were small in number. By contrast, for PLL cells, in vivo cells contributed only to PLL1 (S/G2/M and meiotic). Moreover, although in vitro cells contributed to a part of PLL1, they were the exclusive source for PLL2 (mostly G1), and the DEGs between in vivo and in vitro cells were large in number (Fig. 2c,d, Extended Data Fig. 6b–f and Supplementary Table 4). The PLL1 signature genes included *ZGLP1*, *STRA8* and *REC8* and exhibited substantial upregulation in in vivo cells, but showed moderate increases in in vitro cells (Extended Data Fig. 6g,h). The genes upregulated in PLL1 in vivo cells were enriched for the GO terms 'meiotic nuclear division', whereas those upregulated in PLL1 or PLL2 in vitro cells were enriched for 'regulation of cell differentiation' and 'cell development' (Extended Data Fig. 6d–h and Supplementary

Table 4). Collectively, BMP-driven hPGCLC differentiation recapitulates the transcriptome dynamics of human PGC differentiation into mitotic pro-spermatogonia or oogonia. Meanwhile, continued culture induces aberrant cells as a minor population; they express low levels of genes for meiotic entry and ectopically upregulate developmental regulators.

## DNA methylome reprogramming

As part of epigenetic reprogramming, human PGCs erase their genome-wide 5mCs to 5–10% and differentiate into either mitotic pro-spermatogonia or oogonia[3,7] (Fig. 1a). Consequently, both mitotic pro-spermatogonia and oogonia erase their imprints and oogonia undergo X chromosome reactivation (XCR)[3,7].

To examine epigenetic reprogramming, we examined genome-wide DNA methylation profiles (Supplementary Tables 2 and 5). We also performed long-read whole-genome sequencing of the female cell lines and reconstructed the active and inactive X chromosomes (Xa and Xi, respectively) (Supplementary Table 6). Human iPS cells showed autosome-wide 5mC levels of around 85%, which, after BMP-driven hPGCLC differentiation, decreased to as low as about 10%. DNA demethylation during BMP-driven differentiation occurred more slowly than that in vivo[3], but had similar or faster dynamics than that during xrOvary-based differentiation[14] (Fig. 3a and Extended Data Fig. 7). DNA

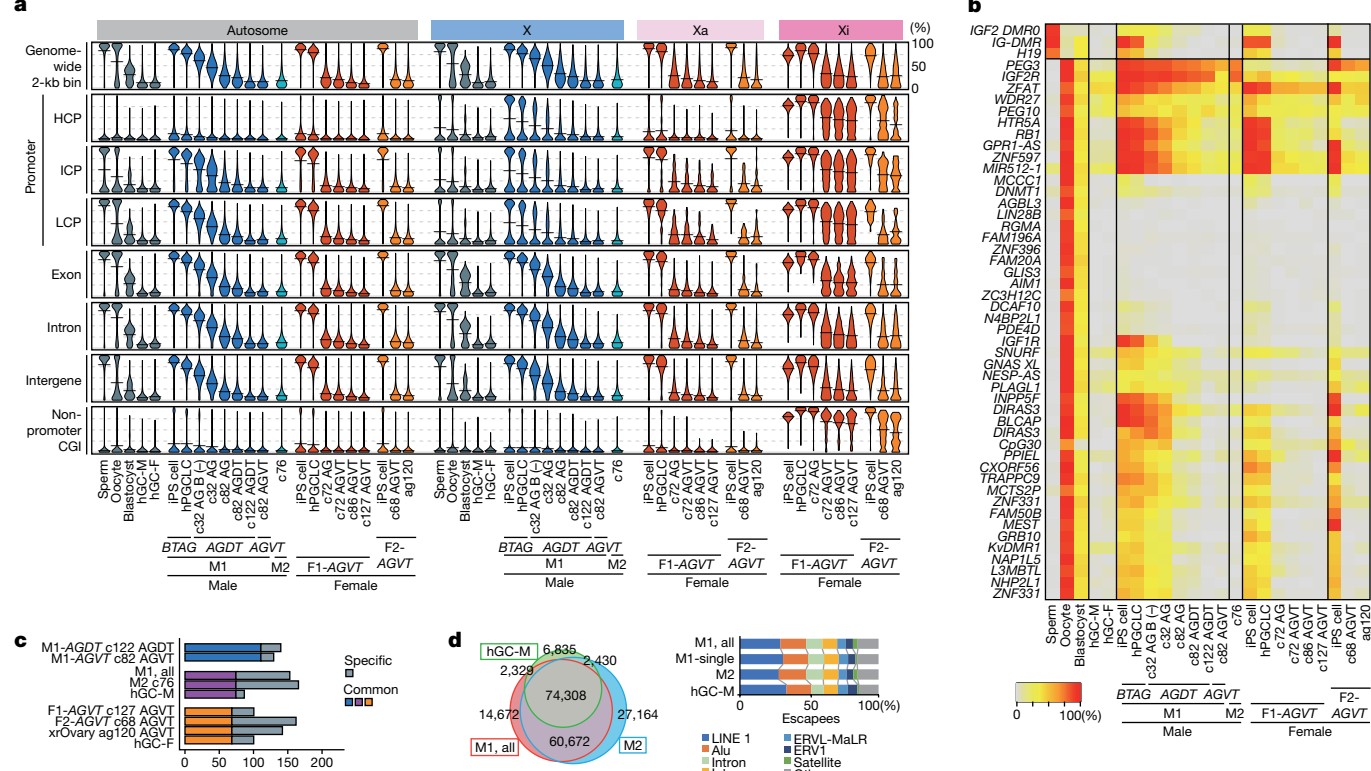

**Fig. 3 | DNA methylome reprogramming during BMP-driven hPGCLC differentiation. a**, Violin plots of the average 5mC levels on the indicated genomic loci in the indicated cell types (see Supplementary Table 2 for full sample information). Bars represent the average values. The DNA methylome data for human spermatozoa, oocytes and blastocytes are from ref. 51 and those for human male germ cells (hGC-M) and female germ cells (hGC-F) at 9 w.p.f. are from ref. 3. B (−) indicates hPGCLC culture without BMP2. HCP, high CpG promoter; ICP, intermediate CpG promoter; LCP, low CpG promoter. **b**, Heatmap of the 5mC levels of the imprint DMRs in the indicated samples.

Colour coding is as indicated. **c**, Escapee numbers common or specific between or in M1-*AGDT* c122 cells and M1-*AGVT* c82 cells (top), among or in the union of M1-*AGDT* c122 and M1-*AGVT* c82 cells, M2 c76 cells, and in vivo male germ cells at 9 w.p.f. (middle), and among or in F1-*AGVT* c127 cells, F2-*AGVT* c68 cells, ag120 cells and in vivo female germ cells at 9 w.p.f. (bottom). Colour coding is as indicated. **d**, Venn diagram showing the relationships of the DNA demethylation escapees among the indicated samples, and composition of the escapees in the indicated samples. Colour coding is as indicated.

demethylation occurred on all unique elements, including promoters of the 13 genes upregulated in gonadal germ cells[19]. DNA demethylation also occurred on genes for 'meiotic cell cycle' and differentially methylated regions (DMRs) of the imprinted genes, with nearly all imprints erased (except *PEG3*, *IGF2R* and *ZFAT*), irrespective of aberrant imprints in parental human iPS cells (Fig. 3a,b and Extended Data Fig. 8a). The 'escapees' that evaded DNA demethylation consisted primarily of evolutionarily young retrotransposons, which highly overlapped with those in mitotic pro-spermatogonia or oogonia and xrOvary-based oogonia-like cells (Fig. 3c,d, Extended Data Fig. 8b,c and Supplementary Table 7). This result suggested that there is a common mechanism for DNA demethylation. Additionally, paired-end sequencing identified centromeric and telomeric satellites as major escapees (Fig. 3d and Extended Data Fig. 8b–e). Note that previously reported hPGCLC induction[23] and hPGCLC culture[24] exhibited highly methylated profiles (Extended Data Fig. 8f).

Similarly, Xa exhibited chromosome-wide DNA demethylation (Fig. 3a and Extended Data Fig. 7b,c). By contrast, Xi showed distinct demethylation. In detail, Xi in F1-*AGVT* human iPS cells had 5mC levels of about 84%, which were reduced to around 26% in AG⁺VT⁺ cells at c86, but thereafter did not undergo further reduction, remaining at about 24% at c127 (Fig. 3a and Extended Data Fig. 7b,c). Promoters and non-promoter CpG islands (CGIs) were highly methylated in the iPS cells (about 87 and 93%, respectively), and most of them showed demethylation resistance, retaining about 44% and 65% 5mC levels on average, respectively, in AG⁺VT⁺ cells at c86–c127 (Fig. 3a and Extended

Data Fig. 7b,c). Xi in F2-*AGVT* human iPS cells exhibited similar demethylation resistance during both BMP-driven and xrOvary-based hPGCLC differentiation[25] (Fig. 3a and Extended Data Fig. 7b,c). Thus, as in human germ cells, BMP-driven hPGCLC differentiation results in comprehensive autosome-wide and Xa-wide DNA demethylation, whereas Xi of the human iPS cell lines we used here were resistant to demethylation.

## ER gene regulation and XCR

We redefined ER genes according to whether they satisfy the criteria in both xrOvary-based and BMP-driven hPGCLC differentiation (Extended Data Figs. 8g–j and 9a,b). Only minimal differences were observed in promoter-wide 5mC levels (about 2%) between c82 DT⁻ and DT⁺ cells, and between c72 VT⁻ and VT⁺ cells (Fig. 3a and Supplementary Table 5). By contrast, the promoter 5mC level of *DAZL* decreased by about 30%, and a ≥4-fold increase in *DAZL* expression was observed between c82 DT⁻ and DT⁺ cells (Extended Data Fig. 9c,d). Similarly, the overall promoter 5mC levels of ER genes were greater in c82 DT⁻ cells than in DT⁺ cells, whereas the overall level of ER gene expression was higher in DT⁺ cells (Extended Data Fig. 9d). This relationship was similar in c72 VT⁻ and VT⁺ cells, although the correlation was not as clear (Extended Data Fig. 9c,d), which is presumably because *DDX4* is a late ER gene and most ER genes were demethylated when *DDX4* was upregulated. Thus, ER gene expression may occur, at least in part, in a coordinated manner in response to promoter demethylation, although single-cell analysis would be necessary for any definitive conclusion.

Next, we examined XCR. We classified informative X-linked genes into four classes on the basis of their promoter 5mC levels in human iPS cells: high 5mC (≥50%) on both Xa and Xi (class 1); low (<50%) on Xa and high on Xi (class 2); high on Xa and low on Xi (class 3: *XIST*); low on both Xa and Xi (class 4) (Extended Data Fig. 9e and Supplementary Table 8). In human iPS cells, most class 1 and class 2 genes were expressed exclusively from Xa, whereas class 4 genes were biallelic and X-chromosome inactivation escapees[26]. Although *XIST* (class 3) was most likely to be expressed from Xi, we could not find informative single nucleotide polymorphisms (SNPs) that discriminate parental alleles (Extended Data Fig. 9e). During BMP-driven hPGCLC differentiation, XCR was limited, with most class 1 and class 2 genes remaining expressed mainly from Xa; however, the genes that erased their promoter 5mC on Xi became biallelically expressed (Extended Data Fig. 9e,f). Accordingly, allele-usage analysis with scRNA-seq revealed that genes expressed predominantly from Xa in VEM cells exhibited modest XCR in EM, mitotic and PLL cells, with about 25% of the transcripts derived from Xi (Extended Data Fig. 9g). We then analysed X-chromosome dosage compensation. In both male and female human iPS cells, the X chromosome to autosome (X:A) ratio of gene expression levels was about 0.9, which indicated that Xa is upregulated[27] (Extended Data Fig. 9h). During BMP-driven hPGCLC differentiation, the X:A ratio in male cells progressively decreased, reaching around 0.7 in mitotic pro-spermatogonia-like cells, whereas that in female cells also decreased, but plateaued at around 0.8 in oogonia-like cells (Extended Data Fig. 9h). These distinct dynamics are reminiscent of those observed during mouse, monkey and human germ-cell development[21,22]. Collectively, XCR, which correlates with promoter demethylation, partially proceeds under the current culture conditions, whereas X-chromosome dosage compensation operates in a broadly normal manner.

## BMP signalling and DNMT activities

During BMP-driven hPGCLC specification and differentiation, hPGCLCs and their progeny downregulated de novo DNA methyltransferases (DNMTs) (Fig. 2a and Extended Data Fig. 9i). The genome-wide CpA methylation, a readout for de novo DNMT activity[28], was reduced in c32 cells cultured with BMP2 compared with their precursors and counterparts cultured without BMP2 (effect sizes (Cohen's *d* values) > 0.2), and the reduced level persisted thereafter (Extended Data Figs. 7b and 9j). Furthermore, hPGCLCs and their progeny downregulated *UHRF1*, an essential co-factor for maintenance DNMT[29,30], and with BMP2, UHRF1 partially translocated to the cytoplasm (Fig. 2a and Extended Data Fig. 9i,k). Compared with c32 cells without BMP2 (about 63%), those with BMP2 had substantially lower autosome-wide 5mC levels (around 45%) (Fig. 3a). hPGCLC-derived cells cultured without BMP2 were estimated to reduce their 5mC levels by only about 1.0% per cell cycle by c32. By contrast, those cultured with BMP2 did so by around 4.9% (Extended Data Fig. 9l). hPGCLC-derived cells from other iPS cell lines cultured with BMP2 reduced their 5mC levels at a similar rate, with those from F2-*AGVT* exhibiting an approximate 7.6% 5mC reduction per cell cycle (Extended Data Fig. 9l).

We analysed the characteristics of DNA demethylation dynamics across genomic bins (10-kb bins) bearing different 5mC levels. The DNA demethylation ratios during the differentiation from human iPS cells to hPGCLCs and from hPGCLCs to c32 cells without BMP2 were relatively small and similar among the genomic bins that had different 5mC levels in the original cells (Extended Data Fig. 9m). By contrast, the DNA demethylation ratios from hPGCLCs to c32 cells with BMP2 and subsequent differentiation were larger and varied across genomic bins, with the bins that had higher 5mC levels in the original cells showing lower demethylation ratios (Extended Data Fig. 9m). The bins showing DNA demethylation resistance corresponded to DNA demethylation escapees (Extended Data Fig. 9n). These findings suggest that BMP-driven hPGCLC specification and differentiation involve

attenuation of both de novo and maintenance DNMT activities. This promotes replication-coupled passive genome-wide DNA demethylation, which occurs heterogeneously depending on the properties of the genomic regions (see Supplementary Discussion 2 for further details).

## TET1 prevents aberrant differentiation

TET1, TET2 and TET3 are active DNA demethylases and transcriptional repressors[2,31]. *TET1* is abundantly expressed in human PGCs and hPGCLCs[3,12] (Fig. 2a). We generated *TET1* knockout (KO) human iPS cells (M1-*BTAG TET1*[−/−] KO1 and KO2) (Extended Data Fig. 10a–e). Both KO lines were induced into BT+AG+ cells apparently normally, and after culture with BMP2, they showed a strong expansion. However, after c12, their expansion was attenuated, with a substantial reduction in the enrichment score, which indicated aberrant differentiation. *TET1* KO BT+AG+ cells could be cultured at least until c42 (Extended Data Fig. 10f,g).

RNA-seq analysis revealed that wild-type cells and *TET1* KO cells underwent similar transcriptome changes until c12. Subsequently, *TET1* KO cells failed in proper maturation and displayed an aberrant trajectory; they did not upregulate *TET2* or *TET3* (Fig. 4a, Extended Data Fig. 10h and Supplementary Table 2). The number of DEGs increased with culture progression, with higher numbers of genes upregulated in *TET1* KO cells (Extended Data Fig. 10i). The DEGs from *TET1* KO cells were classified into distinct clusters, with upregulated genes including 'positive regulation of transcription from RNA polymerase II promoter' (for example, *HAND1*, *HAND2*, *CEBPA*, *CEBPD*, *TBXT* and *TBX3*) and downregulated genes including those for 'fertilization/male meiosis' (for example, *SPATA22*, *TDRD12* and *MEIOB*). Notably, *TET1* KO cells failed to properly upregulate ER genes (Extended Data Fig. 10j–l and Supplementary Table 9).

We performed scRNA-seq of the whole culture at c18, when the *TET1* KO cells showed substantial degrees of aberrant differentiation (Extended Data Fig. 10g). Analysed cells were classified into two major groups: group 1 cells represented propagating hPGCLCs (clusters 1–3: PGC marker+; predominantly wild-type cells), whereas group 2 cells represented aberrantly differentiating cells (clusters 4–11: PGC marker[low/−]; predominantly *TET1* KO cells) (Fig. 4b and Extended Data Fig. 11a,b). Clusters 3, 8 and 7 had a relatively high HVG correlation between group 1 and group 2 cells (Extended Data Fig. 11c). Partition-based graph abstraction analysis indicated a trajectory from cluster 3 to cluster 8 between group 1 and group 2 cells (Extended Data Fig. 11d). Cluster 3 cells ectopically upregulated transcription factors (TFs) such as *GATA3* and *HAND1*, with a subset of them repressing PGC and pluripotency genes. By contrast, cluster 8 and cluster 7 cells more fully repressed PGC and pluripotency genes, with cluster 7 cells strongly upregulating amnion and trophectoderm (AM/TE) markers[32,33] (Fig. 4b and Extended Data Fig. 11e). Cells of clusters 4–6 continued to express AM/TE markers, whereas cluster 9 cells acquired an endothelium-like profile, and cluster 10 and cluster 11 cells were of unclear identity (Extended Data Fig. 11e). We next performed an integrated analysis of our datasets against those of PS-cell-based human development models and a human gastrula in vivo[34,35]. Cells of clusters 7 and cluster 8 and clusters 4–6 were co-segregated with or predicted to be AM/TE-like and (extra-embryonic) mesoderm-like cells, respectively (Extended Data Fig. 11f–k). Thus, during hPGCLC specification and differentiation, TET1 functions as a transcriptional repressor, including for TFs for AM/TE development, which safeguards hPGCLCs from aberrant differentiation into extraembryonic-cell fates.

## Bivalent gene derepression and ER gene deactivation

We evaluated the impact of *TET1* KO on the DNA methylome. At c12, before which *TET1* KO cells exhibited a relatively normal expansion and transcriptome (Fig. 4a and Extended Data Fig. 10g), they showed higher

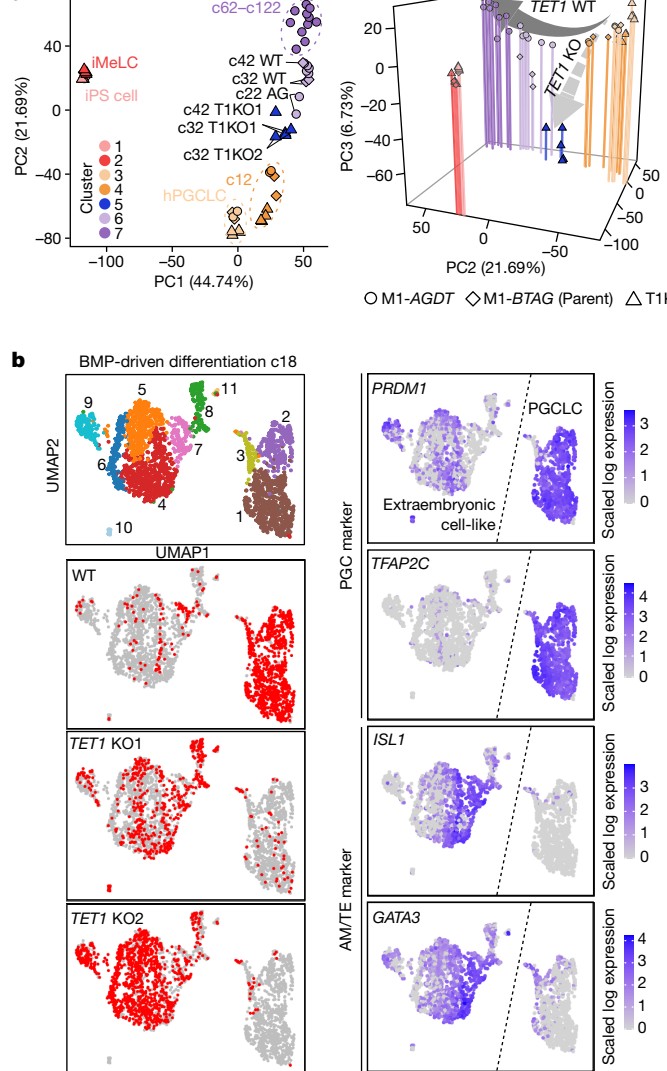

**a**

**b**

**Fig. 4 | TET1 protects hPGCLCs from differentiation into extraembryonic cells. a**, PCA (left: PC1 and PC2; right: PC1, PC2 and PC3) of the transcriptomes of BMP-driven wild-type (WT) cell and *TET1* KO (T1KO) hPGCLC differentiation (see Supplementary Table 2 for full sample information and Extended Data Fig. 10h for cluster information). Colour coding is as indicated. **b**, UMAP and Louvain clustering of scRNA-seq data of wild-type and *TET1* KO hPGCLC culture at c18 (left column, top), with the annotation of the genotype (left column) or with the expression levels of the indicated genes (right column). Colour code is as indicated.

5mC levels in all genomic elements (about 4% higher genome-wide), and DMRs were enriched in promoters, CGIs and coding sequences (CDSs) (Fig. 5a–c and Supplementary Table 5). We classified all open sites (promoters and enhancers (non-promoter open sites)) during hPGCLC induction into three major categories (active, bivalent and poised) on the basis of their combinatorial epigenetic states[36] (Extended Data Fig. 12a). We then examined 5mC levels of each category as well as of silent regions (unopen promoters and enhancers not categorized into active, bivalent or poised). In *TET1* KO cells, all promoters and enhancers showed elevated 5mC levels, with poised promoters and bivalent/poised enhancers exhibiting high increases (about >10% higher) (Fig. 5d). ER genes were enriched in poised promoters, and they (particularly early ER genes) exhibited highly increased promoter 5mC levels, whereas imprint DMRs showed increases in 5mC levels that were slightly higher than the genome-wide average (Fig. 5e). The differences in 5mC levels on the intergenic regions from which all

open sites were subtracted, that is, the 'background' genome (about 55% of the genome), were small (around 2% difference) (Fig. 5f). This result suggests that the impact of TET1 on the background genome might be relatively minor.

Genes with bivalent promoters and enhancers showed a general trend of upregulation in *TET1* KO cells (Fig. 5g). Accordingly, such genes, including *MSX1*, *HAND1* and *CDX2*, were over-represented among upregulated DEGs, whereas ER genes displayed a trend of downregulation and were over-represented in downregulated DEGs (Fig. 5h and Extended Data Fig. 10l). The hypermethylation of bivalent promoters associated with transcriptional upregulation might be due to impaired recruitment of Polycomb repressive complexes, as shown in mouse embryonic stem (ES) cells[37,38]. The upregulated genes with bivalent (the largest set) and poised promoters were highly enriched with the targets of TET1 in human ES cells[39] (Fig. 5i), which indicated that key targets of TET1 may be shared between human ES cells and hPGCLCs.

By c42, *TET1* KO cells showed impaired expansion, and their transcriptomes were highly aberrant (Fig. 4a and Extended Data Fig. 10g–l), which precluded a direct evaluation of the primary function of TET1. Nevertheless, they failed to properly promote DNA demethylation in all genomic elements, and the extent of demethylation seemed to reflect their expansion rates (Fig. 5a and Extended Data Fig. 10g). Furthermore, DMRs remained enriched in promoters, CGIs and CDSs (Extended Data Fig. 12b). Notably, the differences in 5mC levels in promoters and enhancers of bivalent genes were relatively small. By contrast, those of poised/ER genes were large, and those of imprint DMRs were less than the genome-wide average (Extended Data Fig. 12c,d).

Genes with bivalent promoters and enhancers showed an overall trend of upregulation, and such genes were highly over-represented in upregulated DEGs (Extended Data Fig. 12e,f). Most ER genes were downregulated, with a significant enrichment in downregulated DEGs (Extended Data Figs. 10l and 12f). By contrast, genes with poised promoters and enhancers, although they exhibited failure of demethylation, did not show a general trend of downregulation (Extended Data Fig. 12c,e,f). The upregulated DEGs with bivalent promoters were large in number and highly enriched with the targets of TET1 in human ES cells[39] (Extended Data Fig. 12g). Collectively, these findings indicate that TET1 functions as a transcriptional repressor for bivalent genes, including key TFs for the extraembryonic cell fates. TET1 also contributes primarily to the demethylation of the regulatory elements, especially the poised and ER gene promoters, and such demethylation is correlated with the upregulation of ER genes, but not genes with poised promoters in general.

## Discussion

We showed that hPGCLCs cultured with BMP2 propagate stably with reduced levels of MAPK (ERK) signalling and both de novo and maintenance DNMTs. These conditions probably promote replication-coupled, passive genome-wide DNA demethylation to differentiate into mitotic pro-spermatogonia or oogonia-like cells (Extended Data Fig. 12h). The finding that BMP signalling stabilizes germ-cell fate is reminiscent of that role that BMP signalling has in sustaining the self-renewal of mouse ES cells by blocking their differentiation[40]. Given that repression of MAPK (ERK) signalling has a key role in inducing naive pluripotency that accompanies repression of DNMTs and DNA methylome reprogramming[41–43], a similar mechanism might operate for hPGCLC differentiation. The precise mechanism of BMP signalling and its relevance in epigenetic reprogramming of PGCs in other species warrant investigation. A recent study[20] induced DAZL+ PGCLCs using RA, activin A and overexpression of *SOX17* and *DMRT1*. However, these cells do not express *DDX4* and remain highly methylated (about 76–79%) (Extended Data Fig. 8f), and are distinct from the mitotic pro-spermatogonia and oogonia-like cells in this study.

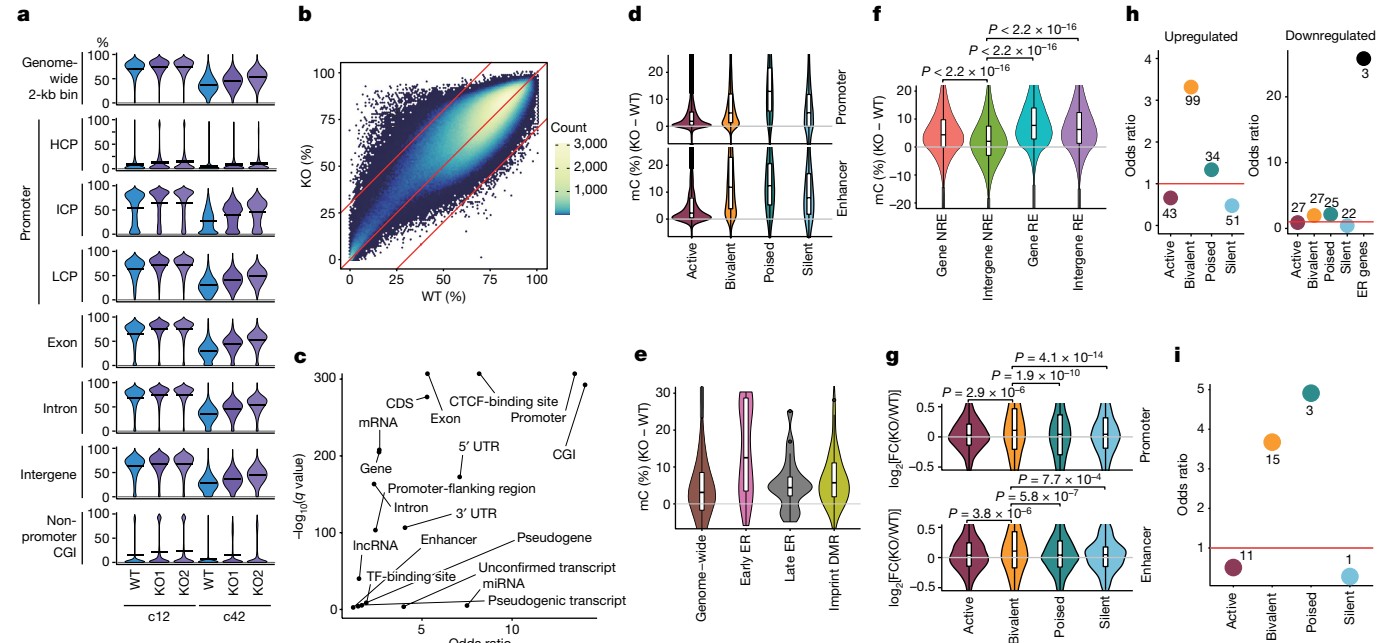

**Fig. 5 | *TET1* KO cells hypermethylate regulatory elements and de-repress bivalent genes. a**, Violin plots of the average 5mC levels on the indicated genomic loci in wild-type and *TET1* KO hPGCLC-derived cells at c12 and c42. Bars represent the average values. **b**, Scatter plot of 5mC levels across all 2-kb bins in wild-type and *TET1* KO hPGCLC-derived cells at c12. The numbers of bins with higher 5mC levels (≥30%) in wild-type (*n* = 10,920) and *TET1* KO (*n* = 238) hPGCLC-derived cells are indicated. For the KO lines, the average values of KO1 and KO2 were used. **c**, Odds ratio and *q* value of the enrichment of the 2-kb bins with higher 5mC levels in wild-type cells than *TET1* KO cells at c12 in the Ensembl Regulatory Build annotations. **d**–**g**, Violin plots for the 5mC level (%) (**d**–**f**) and the expression level (log₂ fold change) (**g**) differences between wild-type and *TET1* KO hPGCLC-derived cells at c12 on the indicated elements. Promoters, enhancers (non-promoter open sites) and their labels are based on the data for day 4 hPGCLCs[36] (Extended Data Fig. 12a). Silent promoters are promoters that did not overlap with open sites; silent enhancers are enhancers categorized neither into active, bivalent nor poised. In **f**, all open sites during hPGCLC induction[36] were defined as regulatory elements (REs). NRE, non-RE regions. Intergenic NREs are defined as the 'background' genome. The upper hinges, lower hinges and middle lines indicate 75 percentiles, 25 percentiles and median values, respectively. The whiskers were drawn in length equal to the inter-quartile range multiplied by 1.5. Data beyond the upper and lower whiskers are shown as dots. Numbers of each element are as follows: **d**, 11,256, 5,014, 4,249 and 17,401 for active, bivalent, poised and silent promoters, respectively; 3,085, 11,654, 111,315, and 275 for active, bivalent, poised, silent enhancers, respectively; in **e**, 1,392,085, 37, 20 and 50 for genome-wide, early ER gene, late ER gene and imprint DMR, respectively; in **f**, 647,564, 770,128, 220,403 and 209,585 for genic NRE, intergenic NRE, genic RE and intergenic RE, respectively; in **g**, 8,562, 1,991, 695 and 2,219 for active, bivalent, poised and silent promoters, respectively; 1,682, 1,560, 6,964 and 101 for active, bivalent, poised and silent enhancers, respectively. *P* values calculated using two-sided Wilcoxon rank-sum test (**f**) or two-sided *t*-test adjusted by Bonferroni correction (**g**). **h**, Odds ratio of the enrichment of genes with indicated promoters and ER genes (for downregulated genes) in genes upregulated (left) or downregulated (right) in *TET1* KO hPGCLC-derived cells at c12. Number of each gene class is indicated. **i**, Odds ratio of the c12 upregulated genes bound by TET in human ES cells[39] in each category of promoters. The odds ratio was calculated relative to the background ratio of all genes bound by TET in each respective promoter category. Number of each gene class is indicated.

In line with a previous report[44], we showed that *TET1*-deficient human iPS cells differentiate into BT⁺AG⁺ cells relatively normally. However, after BMP-driven differentiation, *TET1*-deficient BT⁺AG⁺ cells differentiated aberrantly into extraembryonic cells, with de-repression of crucial TFs with bivalent promoters (Figs. 4 and 5). The role of TET proteins in bivalent gene repression in cooperation with Polycomb repressive complexes is consistent with previous findings[37–39]. *TET1*-deficient hPGCLC-derived cells failed to upregulate ER genes, with their promoters remaining methylated (Fig. 5). In mice, *Tet1* is dispensable for genome-wide DNA demethylation per se, but is crucial in maintaining demethylation of key genes for spermatogenesis and oogenesis and of imprint DMRs[45–47]. To better understand the function of TET proteins in epigenetic reprogramming in humans, it would be necessary to evaluate catalytic mutants while leaving their transcriptional repressor activity intact.

During BMP-driven and xrOvary-based hPGCLC differentiation, unlike autosomes and Xa, Xi—especially Xi on promoters and non-promoter CGIs—exhibited a resistance to reprogramming with limited XCR (Fig. 3a). This effect is probably due to an aberrant Xi epigenetic state, including Xi-wide H3K9me3-based repression, in human iPS cells[48]. In line with this idea, using cynomolgus monkey cells, PGCLCs induced from ES cells bearing stable Xi with H3K27me3-based repression underwent more comprehensive Xi reprogramming and XCR in xrOvaries[25]. These cells also differentiated up to fetal oocyte-like cells at the zygotene stage of the meiotic prophase[25]. It will be important to perform comprehensive characterizations of the epigenetic states, including the histone-modification states, for BMP-driven hPGCLC differentiation and explore their significance in the properties and functions of hPGCLC-derived cells. A recent study showed that hPGCLCs induced from epigenetically reset human PS cells have an accelerated differentiation potential[15]. This finding warrants further investigation, including using the BMP-driven hPGCLC differentiation system.

In late culture phases, about 5% of hPGCLC-derived cells differentiate into AG⁻DT⁺ or AG⁻VT⁺ cells with gene expression suggestive of meiotic entry (Figs. 1c and 2b–d). However, they failed to acquire sufficient levels of meiotic genes and upregulated genes for ectopic developmental and signalling pathways (Extended Data Fig. 6d–h). BMP signalling induces competent mouse PGCLCs into meiotic prophase[49,50], and the mechanism for oocyte differentiation may be conserved in humans. However, compared to mice, humans and monkeys exhibit distinct transcriptome dynamics after oogonia-to-oocyte transition[8,22] and may use a distinct mechanism for oocyte differentiation. BMP-driven hPGCLC differentiation should serve as a system to identify conditions for inducing human fetal oocytes and to investigate the underlying

mechanism. In a broader context, with its capacity to provide abundant numbers (>$10^{10}$ cells) of both mitotic pro-spermatogonia and oogonia, this system will be important for future directions of human IVG research.

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

# Methods

## Human iPS cell culture

All experiments using human iPS cells were approved by the Institutional Review Board of Kyoto University and were performed according to the guidelines of the Ministry of Education, Culture, Sports, Science, and Technology of Japan. The iPS cell lines used were 585B1, 1383D6, 1390G3 and NCLCN, and their derivatives[11,52,53]. NCLCN (derived from cord blood cells) was purchased from XCell Science and licensed for academic use. All human iPS cell cultures were performed in a humidified incubator with 5% $CO_2$ at 37 °C. All human iPS cell lines used in this study were maintained in StemFit AK02N or AK03N (Ajinomoto) on a plate coated with iMATRIX-511 (Nippi, 892014). For passage, human iPS cell colonies were dissociated into single cells by treating with 0.5× Tryple select (1:1 mixture of Tryple select (Gibco, 12563-011) and PBS(−) (phosphate buffered saline, Nacalai Tesque, 11480-35) containing 0.5 mM EDTA (Nacalai Tesque, 06894-14)). Dissociated cells were plated with culture medium containing 10 μM of Y27632 (Tocris, 1254), and the medium was replaced with fresh medium without Y27632 the next day. Pictures of human iPS cell colonies were taken using a CKX41 inverted microscope (Olympus) equipped with a DS-Fi2 camera (Nikon).

## hPGCLC induction

All iMeLC and hPGCLC induction experiments were performed in a humidified incubator with 5% $CO_2$ at 37 °C. Human iPS cells were induced to iMeLCs and then into hPGCLCs as previously described[11]. To induce iMeLCs, $1.0–2.0 \times 10^5$ cells per well of human iPS cells were plated with iMeLC induction medium (15% knockout serum replacement (KSR) (Gibco, 10828-028), 1% MEM non-essential amino acids solution (NEAA) (Gibco, 11140-050), 1% penicillin–streptomycin, 2 mM L-glutamine (Gibco, 25030-081), 2 mM sodium pyruvate (Gibco, 11360-070) and 0.1 mM 2-mercaptoethanol in GMEM (Gibco, 11710-035)) supplemented with 3 μM CHIR99021 (Tocris, 4423), 50 ng ml$^{-1}$ activin A (PeproTech, AF-120-14) and 10 μM Y27632 on a fibronectin (Millipore, FC010)-coated 12-well plate (the induction time was 44 h for M1-*BTAG* and M1-*AGDT* human iPS cells, 48 h for M1-*AGVT*, M2 and F2-*AGVT* human iPS cells and 54 h for F1-*AGVT* human iPS cells). Note that 50 nM PD173074 was added for iMeLC induction from female human iPS cell lines[11,53]. To induce hPGCLCs, $3.0–6.0 \times 10^3$ cells per well of iMeLCs were plated with PGCLC induction medium (15% KSR, 1% NEAA, 1% penicillin–streptomycin, 2 mM L-glutamine, 2 mM sodium pyruvate and 0.1 mM 2-mercaptoethanol in GMEM or advanced RPMI (Gibco, 12633-012)) supplemented with 200 ng ml$^{-1}$ BMP4 (R&D Systems, 314-BP), 100 ng ml$^{-1}$ SCF (R&D SYSTEMS, 255-SC), 10 ng ml$^{-1}$ LIF (Merck Millipore, LIF1010), 50 ng ml$^{-1}$ EGF (PeproTech, AF-100-15) and 10 μM Y27632 in a v-bottom 96-well plate (Greiner, 651970). At 6 or 8 days after hPGCLC induction, iMeLC aggregates were subjected to FACS (see 'FACS analysis') to isolate hPGCLCs. Pictures of iMeLC aggregates under the hPGCLC induction condition were taken using a M205C microscope (Leica) equipped with a DP72 camera (Olympus).

## hPGCLC expansion and differentiation culture

All hPGCLC expansion and differentiation cultures were performed in a humidified incubator with 5% $CO_2$ at 37 °C and maintained in the expansion culture medium (aRK15: advanced RPMI (Gibco, 12633-012) supplemented with 15% KSR (Gibco, 10828-128), 1× GlutaMAX (Gibco, 35050-061), 1% penicillin–streptomycin (Gibco, 15140-122) and 0.1 mM 2-mercaptoethanol (Gibco, 21985-023)), 2.5% FBS (Nichirei, 175012), 100 ng ml$^{-1}$ hSCF (R&D Systems, 255-SC), 10 μM forskolin (Sigma, F3917), 20 ng ml$^{-1}$ hbFGF (Wako, 064-04541) and 1.5 μM IWR1 (Sigma, I0161)) unless otherwise specified. For differentiation of male hPG-CLCs, 25 ng ml$^{-1}$ BMP2 was supplemented until the first passage. For differentiation of female hPGCLCs, 50 ng ml$^{-1}$ BMP2 was supplemented until the first passage. After the first passage, both male and female hPGCLCs were cultured with 100 ng ml$^{-1}$ BMP2. Only when cells were plated after passage, 10 μM Y27632 was added. For the cytokine and chemical screens (Fig. 1d), the expansion culture medium was supplemented with the following cytokines or chemicals in the indicated amounts: DMSO (Sigma, D8418), VO-OHpic (10 μM; Selleckchem, S8174), PS48 (10 μM; Sigma, P0022), MHY1485 (5 μM; Sigma, SML0810), PD0325901 (1 μM; StemGent, 04-0006), PD173074 (1 μM; StemGent, 04-0008), CHIR99021 (1 μM; Tocris, 4423), RA (1 μM; Sigma, R2625), BMS 493 (1 μM; Sigma, B6688), WIN18,499 (1 μM; Cayman Chemical, 14018), SAG (1 μM; Tocris, 4366), PGD2 (100 ng ml$^{-1}$; Cayman Chemical, 12010), FGF4 (100 ng ml$^{-1}$; R&D Systems, 235-F4-025), FGF9 (100 ng ml$^{-1}$; R&D Systems, 7399-F9), FGF18 (100 ng ml$^{-1}$; R&D Systems, 8988-F18), WNT5A (100 ng ml$^{-1}$; R&D Systems, 645-WN), BMP2 (25 ng ml$^{-1}$; R&D Systems, 355-BM-01M/CF), BMP4 (25 ng ml$^{-1}$; R&D Systems, 314BP01M), BMP7 (100 ng ml$^{-1}$; R&D Systems, 5666-BP-010), Nodal (50 ng ml$^{-1}$; R&D Systems, 1315-ND), Inhibin A (10 ng ml$^{-1}$; R&D Systems, 8346-IN), NOGGIN (50 ng ml$^{-1}$; Proteintech, HZ-1118), pleiotrophin (100 ng ml$^{-1}$; R&D SYSTEMS, 252-PL-050), Midkine (100 ng ml$^{-1}$; R&D Systems, 9760-MD), osteopontin (100 ng ml$^{-1}$; R&D Systems, 441-OP), SDF-1a (100 ng ml$^{-1}$; R&D Systems, 460-SD), IGF2 (100 ng ml$^{-1}$; R&D Systems, 792-MG), Desert Hedgehog (50 ng ml$^{-1}$; R&D Systems, 733-DH) or Jagged1 (100 ng ml$^{-1}$; R&D Systems, 599-JG). Details are summarized in Supplementary Table 10. For comparison with basal medium (Extended Data Fig. 1d), aRK15 was replaced with the indicated basal medium listed in Supplementary Table 10. At the beginning of the expansion culture, hPGCLCs were directly plated on the mitomycin-c-treated m220-5 feeder cells by FACS using the automated cell deposition unit sorting mode. Alternatively, hPGCLCs were sorted into FACS buffer (0.1% BSA–PBS(−) containing 10 μM Y27632). Then, the sorted hPGCLCs were collected by centrifugation at 200*g* for 7 min and were plated with the expansion culture medium. hPGCLCs were identified as described in 'FACS analysis'.

A step-by-step protocol for the passage of hPGCLC expansion and differentiation culture (modified from ref. 12) is as follows.

For preparation of the culture plate and m220-5 feeder cells: (1) coat a flat-bottom cell culture plate (for example, Falcon, 353072) with 0.1% gelatin solution at room temperature for at least 1 h. (2) Plate MMC-treated (4 μg ml$^{-1}$, 2 h) m220-5 feeder cells on a gelatin-coated cell culture plate. Typically, $6.25 \times 10^4$ cells are plated in one well of a 96-well plate with 1 ml of DMEM containing 10% FBS, 1% penicillin–streptomycin and 1× GlutaMAX. (3) Incubate m220-5 feeder cells in a humidified incubator maintained at 5% $CO_2$ and 37 °C until use.

For cell collection and re-plating: (1) wash cultured cells with PBS(−) once. (2) Treat cultured cells with a 1:4 mixture of 0.5% trypsin–EDTA and PBS(−) at 37 °C for 5 min. Alternatively, Accumax (AM105, Innovative Cell Technologies) can be used to dissociate the cultured cells. (3) Neutralize the trypsin by adding advanced RPMI containing 10% FBS, 0.1 mg ml$^{-1}$ DNase1 and 10 μM Y27632. Alternatively, the expansion culture medium can be used for the neutralization. (4) Repeat the pipetting of the cell suspension to dissociate it into single cells. (5) If needed, collected cells were subjected to immunolabelling (see 'FACS analysis'). (6) Plate the desired volume of cell suspension with culture medium supplemented with 10 μM Y27632 on the m220-5 feeder cells. Typically, the collected cell suspension is passaged at a dilution ratio of around 1:5. (7) Incubate cells in a humidified incubator maintained at 5% $CO_2$ and 37 °C. (8) The remaining cell suspension was subjected to FACS after adding DAPI at a final concentration of 0.1 μg ml$^{-1}$. To measure the growth of AG$^+$ cells, DAPI$^-$ live cells were used for analysis.

For maintenance of the expansion culture: (1) replace the medium with fresh medium one on the second day. (2) Replace half the medium with fresh medium every other day until the next passage on around day 10 after cell plating.

## Establishment of the reporter knock-in iPS cell lines

To generate iPS cell lines bearing *TFAP2C-p2A–eGFP*, *DAZL-p2A–tdTomato* and *DDX4-p2A–tdTomato* reporters, the *p2A-eGFP* and

*p2A-tdTomato* sequences were replaced with the termination codon of the targeted gene by TALEN-assisted homologous recombination. First, homologous regions surrounding the termination codon of human *DAZL* and *DDX4* were cloned into the pCR2.1 vector (pCR2.1-DAZL and pCR2.1-DDX4, respectively) (TOPO TA Cloning; Thermo Fisher Scientific, 450641). Next, the *p2A-tdTomato* sequences concatenated with the PGK promoter-driven neomycin resistance gene flanked by *loxP* sites were subcloned into the pCR2.1-DAZL and pCR2.1-DDX4 vectors by using a GeneArt seamless cloning and assembly kit (Thermo Fisher Scientific, A13288). Finally, the MC1 promoter-driven DT-A gene was isolated by XbaI/ApaI or XhoI/ApaI digestion and then subcloned into the region downstream of the homologous regions by enzymatic ligation.

To assemble TALEN constructs targeting the human *DAZL* and *DDX4* gene loci, a GoldenGate TALEN and TAL Effector kit (Addgene, 1000000016) was used. The amino acid sequences of repeat variable di-residues in each TALEN were designed as follows: *DAZL* 5′ end [NG NG NN NI NG HD HD NG HD HD NG NN NN HD NG NG NI NG HD]; *DAZL* 3′ end [NI NG NG HD NI NI NI NI HD HD NI NN HD NI NI HD NG NG]; DDX4 5′ end [HD HD HD NI NI NG HD HD NI NN NG NI NN NI NG NN NI NG]; and *DDX4* 3′ end [NN NI NI NN NN NI NG NN NG NG NG NG NN NN HD NG].

Next, 5 µg of the donor vector and 2.5 µg of each TALEN plasmid were introduced into $1.0 \times 10^6$ human iPS cells (585B1#17, *TFAP2C-p2A–eGFP* with the PGK promoter-driven puromycin resistance gene flanked by *loxP* sites)[11] by electroporation using a NEPA21 type II electroporator (Nepagene). Then, 150 µg ml$^{-1}$ of G418 (2–4 days after the electroporation) and 0.2 µg ml$^{-1}$ puromycin (4 days after the electroporation) were added to eliminate untargeted cells. About 2 weeks after the electroporation step, single iPS cell colonies were isolated by manually picking them up using disposable plastic pipette tips. The zygosities of the targeted locus and random integrations of isolated clones were assessed by PCR using genomic DNA samples. Isolated clones bearing reporter genes knocked-in in a heterozygous manner were subsequently transfected with Cre recombinase expression plasmids. Finally, the excision of *loxP*-flanked drug-resistance genes in isolated clones was verified by PCR using genomic DNA samples. DNA sequences of the oligonucleotides used are listed in Supplementary Table 11.

M1-*AGDT*, M1-*AGVT* and F1-*AGVT* iPS cell lines were confirmed to be karyotypically normal (46, XY for M1 and 46, XX for F1, respectively) by G-banding analysis performed by Nihon Gene Research Laboratories.

## Establishment of *TET1* KO human iPS cell lines

*TET1* KO human iPS cell lines were established as previously described[54]. M1-*BTAG* was used as a parental line. The DNA oligonucleotides used for vector construction are listed in Supplementary Table 11. One million BTAG iPS cells were transfected with 3.0 µg each of two vectors respectively encoding guide RNAs targeting the human *TET1* gene and mCherry-tagged Cas9 nickase in 100 µl OptiMEM (Gibco, 31985-062) using a NEPA21 type II electroporator. Two days after transfection, the transfected BTAG iPS cells highly expressing mCherry were subjected to single cell sorting by FACS using the automated cell deposition unit sorting mode. The sorted cells were directly plated on 96-well plates pre-filled with AK02N supplemented with 10 µM Y27632 and 0.25 µg cm$^{-2}$ iMATRIX. Three days after single-cell sorting, the culture medium was replaced with AK02N. At 12–14 days after single-cell sorting, the viable cells were cryopreserved in Stem-Cellbanker cryopreservation solution (Zenoaq, CB045) or snap-frozen in liquid nitrogen for genotyping PCR.

## Genotype analysis of *TET1* KO human iPS cell lines

Each cell pellet was resuspended in lysis buffer (3 µl of 10× KOD Plus Neo Buffer (Toyobo, KOD-4B), 3 µl of 5% Nonidet P40 (Nacalai Tesque, 23640-94), 1 µl of 20 mg ml$^{-1}$ proteinase K and 23 µl of water). All cell suspensions were incubated at 55 °C for 2 h, followed by 95 °C for 5 min. Alternatively, genomic DNA was extracted from the iPS cell pellet by using NucleoSpin Tissue (Macherey-Nagel, U0952S)

according to the manufacturer's instructions. A total of 1 µl of the cell lysate or genomic DNA was subjected to PCR amplification using KOD-Plus-Neo (Toyobo, KOD-401) with the following primers: forward, TTTTTAAAGAGTGATCTTTGAGGATAAGTGA; reverse, ATCACTCTGGCTATTGTAAATGTAT. After purification of the PCR product, amplicons were cloned into a pGEM-T Easy Vector (Promega, A1360). The vectors were transformed into homemade competent cells of the DH5α strain. Transformed cells were plated on a LB plate coated with X-gal and IPTG for blue–white selection and were incubated at 37 °C until colony formation. White colonies were directly used as a template for the PCR amplification using KOD FX Neo (Toyobo, KFX-201) with the following primers: forward, CAGGAAACAGCTATGAC; reverse, GTAAAACGACGGCCAG. To identify the insertions and deletions of each clone, PCR products were subjected to Sanger sequence analysis by Eurofins Genomics.

## Cell number count of hPGCLC-derived cells

To calculate the expansion of hPGCLC-derived cells, we used the sum of reporter/surface marker$^+$ cells. That is, the sum of BT$^+$AG$^+$ cells for M1-*BTAG*, the sum of AG$^+$DT$^-$/VT$^-$, AG$^+$DT$^+$/VT$^+$ and AG$^-$DT$^-$/VT$^+$ cells for M1-*AGDT*/*VT* and F1/F2-*AGVT*, and the sum of EpCAM$^+$/ITGB6$^+$ cells for M2.

## Preparing samples for mass spectrometry by in-gel digestion

About $2 \times 10^6$ M1-*BTAG* iPS cells or *TET1* KO iPS cells were lysed in RIPA buffer (Santa Cruz, SC-24948). After 30 min of incubation at 4 °C on a rotator, the cell lysate was subjected to sonication. Then, the cell lysate was centrifuged at 14,000 r.p.m. at 4 °C for 10 min, and the supernatant was used for the experiment. Total protein concentration was measured using a Pierce BCA Protein Assay kit (Thermo Fisher Scientific, 23227). Cell lysate containing 12 µg protein was mixed with 4× Laemmli sample buffer (Bio-Rad, 1610747) supplemented with 2-mercaptoethanol and boiled at 95 °C for 5 min. After incubation on ice, the cell lysate was separated by 8% SDS–PAGE in MOPS buffer at 100 V for 20 min. Gel pieces containing separated proteins of around 260 kDa and 160 kDa were collected into microtubes independently. The gel pieces were sequentially treated as follows: destaining with 200 µl of 50 mM ammonium bicarbonate (ABC) (Fujifilm Wako, 017-02875) and 50% acetonitrile (ACN) for 30 min; dehydration with 100% ACN; reduction with 500 µl of 10 mM dithiothreitol (DTT) (Fujifilm Wako, 045-08974) and 50 mM ABC for 30 min; alkylation with 50 mM iodoacetamide (IAA) (Fujifilm Wako, 095-02151) and 50 mM ABC for 30 min in the dark; washing with 200 µl of 0.5% acetic acid and 50% methanol twice; equilibration with 50 mM ABC; dehydration with 100% ACN; protein digestion with 10 µl of trypsin solution (10 ng µl$^{-1}$ in 50 mM ABC) for 5 min; and additional incubation with 50 µl of 50 mM ABC buffer at 37 °C overnight. Trypsinization was terminated by adding 5 µl of 10% trifluoroacetic acid (TFA). The supernatant was subjected to additional extraction with 50% ACN and 0.1% TFA and 80% ACN and 0.1% TFA. Finally, the supernatant was dried and suspended in 0.1% TFA. The resulting samples were desalted using SDB-XC StageTips[55,56].

## LC–MS/MS

LC–MS/MS for in-gel digested samples was performed using an Orbitrap Fusion Lumos (Thermo Fisher Scientific) linked with an Ultimate 3000 pump (Thermo Fisher Scientific) and HTC-PAL autosampler (CTC Analytics). Samples were subjected to a needle column pulled in-house (150 mm length, 100 µm inner diameter, 6 µm needle opening) house-packed with Reprosil-Pur 120 C18-AQ 1.9 µm reversed-phase material (Dr. Maisch), using a 60 min gradient of 5–40% B at a flow rate of 500 nl min$^{-1}$ (mobile phase A: 0.5% acetic acid; mobile phase B: 0.5% acetic acid and 80% ACN). Orbitrap MS1 spectra were collected at a resolution of 120,000. Data-dependent ion trap MS2 scans were collected in the Top Speed mode using a cycle time of 1.5 s between full MS scans.

## LC−MS/MS data analysis

Acquired data files were processed using FragPipe (v.20.0) (MSFragger (v.3.8), Philosopher (v.5.0.0) and IonQuant (v.1.9.8)) for protein identification and quantitfication[57–59]. Peptides and proteins were identified by automated database searching against the human UniprotKB/Swissprot database (accessed on 15 July 2021) with settings of strict trypsin (carboxy-terminal of K and R) specificity and the allowance for up to two missed cleavages. Carbamidomethylation of cysteine (C, +57.021465) was set as a fixed modification. Oxidation of methionine (M, +15.9949) and acetylation of protein amino terminus (Protein N-term, +42.0106) were set as variable modifications. False discovery rates of peptide and protein levels were controlled to 1% using PeptideProphet and ProteinProphet[60]. Peptide quantification was performed using IonQuant.

## FACS analysis

hPGCLCs were isolated from the aggregates of iMeLCs as previously described[12]. In brief, iMeLC aggregates were treated with a 1:1 mixture of PBS(−) and 0.5% Trypsin-EDTA (Gibco, 15400-054). After the neutralization with DMEM (Gibco, 10313-021) containing 10% FBS, 1% penicillin−streptomycin, 2 mM L-glutamine, 10 μM Y27632, and 0.1 mg ml$^{-1}$. Samples were subjected to a needle column pulled in-housel DNase I (Sigma, DN25), the cell suspensions were subjected to centrifugation at 200$g$ for 5 min. The cells were then resuspended in FACS buffer and sorted using a FACS Aria III. hPGCLCs were gated as BT$^+$AG$^+$ cells (M1-$BTAG$ line) or AG$^+$ cells (M1-$AGDT$, M1-$AGVT$, F1-$AGVT$ and F2-$AGVT$ lines) or EpCAM$^+$ITGA6$^+$ cells (M2) on hPGCLC induction day 6 or day 8.

To analyse the surface antigen expression, trypsinized cells were resuspended in FACS buffer containing antibodies (BV421-conjugated anti-TRA-1-85 antibody (BD Bioscience, 563302) or BV421-conjugated isotype control antibody (BD Bioscience, 562438), PE-conjugated or BV421-conjugated anti-ITGA6/CD49f antibody (BioLegend, 313611/313624), APC-conjugated anti-EpCAM/CD326 antibody (BioLegend, 324208)) and were incubated on ice for 30 min in the dark. After washing with PBS(−) and pelleting, the cells were resuspended in FACS buffer and were subjected to FACS analysis. For FACS gating details, see Supplementary Fig. 2.

## Immunofluorescence analysis

Immunofluorescence analysis was performed as previously described[12]. The expanded PGCLCs were cultured in a μ-Slide 8-Well chamber slide (ibidi, 80826) for several days. After washing with PBS(−) once, the cells were fixed in 4% paraformaldehyde for 15 min at room temperature and washed with PBS(−) three times. The fixed cells were incubated in blocking buffer (PBS(−) with 10% normal donkey serum (Jackson ImmunoResearch, 017-000-121), 3% BSA (Sigma, A4503) and 0.1% Triton-X 100 (Nacalai Tesque, 35501-02)). Then, the cells were incubated in 0.5× blocking buffer (a 1:1 mixture of blocking buffer and PBS (−)) containing primary antibodies for 30 min at room temperature. The dilution rates for the primary antibody reactions were as follows: rat anti-GFP antibody (Nacalai Tesque, 04404-84; 1:250); goat anti-tdTomato antibody (Origene, AB8181-200; 1:200); mouse anti-UHRF1 antibody (Millipore, MABE308, 1:200); mouse anti-AP2γ/TFAP2C (1:100; Santa Cruz, sc-12762); mouse anti-DAZL antibody (Santa Cruz, sc-390929; 1:100); and rabbit anti-DDX4 antibody (Abcam, ab13840; 1:250). Then the cells were washed with PBS(−) three times and incubated in 0.5× blocking buffer containing 1 μg ml$^{-1}$ of DAPI and secondary antibodies (Alexa Fluor 488 donkey anti-rat IgG (Invitrogen, A21208; 1:800); Alexa Fluor 568 donkey anti-goat IgG (Invitrogen, A11057; 1:800); Alexa Fluor 647 donkey anti-mouse IgG (Invitrogen, A31571; 1:800); and Alexa Fluor 647 donkey anti-rabbit IgG (Invitrogen, A31573; 1:800)) for 1 h at room temperature. The cells were washed with PBS(−) three times and were mounted in ibidi mounting medium (ibidi, 50001). Images were acquired using a confocal microscopy system (IX81-FV1000, Olympus) or fluorescence microscope (BZX810, Keyence). For the line-plot

analysis (Extended Data Fig. 9k), the signal intensities of the lines placed across randomly picked ten cells were measured with the 'plot profile' function of Fiji software[61]. The outlines of the nucleus and cytoplasm were manually determined based on the visual inspection of the DAPI and AG signals, respectively. The UHRF1 signals were normalized to set the maximum value as 1. To plot the signals from the various lengths of the nucleus and cytoplasm in an integrated manner, the lengths of the nucleus and cytoplasm were normalized to a range from 0 to 1. The curve fitting analysis was performed using the geom_smooth function with "method = "gam"" option from the ggplot2 package in R.

## Automated image analysis

Immunofluorescence images were quantified using CellProfiler (v.4.2.1) automated image-processing software[62]. For analysing UHRF1 subcellular localization (Extended Data Fig. 9k), first, nuclei were identified based on the DAPI channel signal using the IdentifyPrimaryObjects module. Second, hPGCLC-derived cells were identified based on the GFP channel signal using the IdentifySecondaryObjects module. Third, cytoplasms were identified by subtracting the nuclei from the cells with the IdentifyTertiaryObjects module. Finally, the intensity of the nuclei and cytoplasms was calculated using the MeasureObjectIntensity module. Mean intensity values of each cell were used for calculating the ratio (cytoplasm over nucleus). For the validation of fluorescence reporter expression, nuclei and cells were identified as described above. The intensity from the nuclei (TFAP2C and GFP) or cells (DAZL, DDX4, GFP and tdTomato) was calculated using the MeasureObjectIntensity module. Mean intensity values were used for analysis.

## Chromosome counting

Human iPS cells were incubated with 0.1 μg ml$^{-1}$ demecolcine (Wako, 045-18761) for 4 h. After incubation, iPS cells were dissociated into single cells with 0.5× Tryple select and then centrifuged at 160$g$ for 5 min.

Expanding hPGCLCs were incubated with 0.1 μg ml$^{-1}$ demecolcine overnight. To isolate expanded hPGCLCs, AG$^+$ cells were sorted using a FACS Aria III (BD) from the whole cell suspension collected from the expansion culture. To prevent the resumption of cell division, cells were sorted into Cellotion solution (Zenoaq, CB051) containing 0.1 μg ml$^{-1}$ demecolcine and centrifuged at 200$g$ for 7 min.

After centrifugation, cells were treated with hypotonic buffer (Genial Helix, GGSJL006B) for 30 min at 37 °C and then treated cells were placed on ice for 3 min. Treated cells were then fixed in a fixative solution (methanol and acetic acid at 3:1) and spread on glass slides. Chromosome spreads were stained with 0.1 μg ml$^{-1}$ DAPI for 10 min at room temperature. Images were taken using a confocal microscopy system (IX81-FV1000, Olympus).

## DNA dot blotting

Genomic DNA was extracted from cell pellets using NucleoSpin Tissue (Macherey-Nagel, U0952S) following the manufacturer's instructions. Eluted DNA was diluted to 5 ng μl$^{-1}$ (for 5mC detection) or 25 ng μl$^{-1}$ (for 5hmC detection). Next, 5 μl of 0.5 N NaOH was added to 20 μl of the DNA. Then, the mixture was incubated at 99 °C for 5 min. After the incubation, 2.5 μl of 6.6 M ammonium acetate was added to the mixture. Finally, 2.75 μl of the mixture, which is equivalent to 50 ng of genomic DNA, was spotted onto a nitrocellulose membrane pre-soaked in 20× SSC. The membrane was baked for 2 h at 80 °C and then incubated in blocking buffer (PBST (PBS containing 0.1% Tween20) containing 10% skim milk (BD, 232100) and 1% BSA (Sigma, A4503)) for 1 h at room temperature. The membrane was incubated with rabbit anti-5hmC antibody (Active Motif, 39069) at 1:5,000 dilution or rabbit anti-5mC antibody (Sigma, SAB5600040) at 1:1,000 dilution in blocking buffer overnight at 4 °C. The membrane was washed in PBST for 5 min 3 times, and then incubated with anti-rabbit IgG goat IgG conjugated with HRP (Sigma, A6154) at 1:8,000 dilution in blocking buffer for 1 h at room temperature. The membrane was washed in PBST for 5 min 3 times, and

then the membrane was subjected to chemiluminescence detection with Chemi-Lumi One Super (Nacalai Tesque, 02230-30) or Chemi-Lumi One Ultra (Nacalai Tesque, 11644-24). The chemiluminescence signal was detected by Fusion solo S (Vilber Lourmat).

## Western blot analysis

About $2.0 \times 10^4$ BT$^+$AG$^+$ cells were collected into a tube containing Cellotion by FACS at c33. After centrifugation, cells were lysed in 4× Laemmli sample buffer supplemented with 2-mercaptoethanol and boiled at 95 °C for 5 min. After incubation on ice, the cell lysates were separated by 10% SDS–PAGE (about $6.0 \times 10^3$ cells per lane) in MOPS buffer at 200 V for 25 min. Separated proteins were transferred onto a PVDF membrane, and the membrane was treated with 5% skim milk in TBS containing 0.1% Tween20 (TBST) for 1 h at room temperature. After blocking, the membrane was incubated with anti-p44/42 MAPK (ERK1/2) antibody (1:1,000, CST, 4695) or anti-phospho p44/42 MAPK (ERK1/2) antibody (1:2,000, CST, 4730) overnight at 4 °C in 5% BSA–TBST. The membrane was washed three times in TBST and subjected to secondary antibody reaction with anti-rabbit IgG goat IgG conjugated with HRP. Immunoblotting for α-tubulin was performed on the same membrane after antibody stripping by using WB stripping solution (Nacalai Tesque, 05364-55). Then the membrane was treated as described above except for a 1-h primary antibody reaction in 5% skim milk in TBST (anti-α-tubulin antibody, 1:8,000, Sigma, T9026). Secondary antibody reaction was performed with anti-mouse IgG sheep IgG conjugated with HRP (1:5,000, Sigma, A5906). After washing with TBST 3 times, chemiluminescence was detected using Chemi-Lumi One Super (for ERK1/2 and α-tubulin, Nacalai Tesque, 02230-30) or Immunostar LD (for phosho ERK1/2, Fujifilm, 296–69901). Images were taken using a Fusion Solo imaging system. The signal intensity was quantified by CaptAdvanced software (Vilber Lourmat). To determine the relative phosphorylation levels of ERK1/2, first, the signal intensities of ERKs or pERKs were divided by the signal intensities of α-tubulin. Then, the normalized pERK intensity was divided by the normalized ERK intensity. For gel source data, see Supplementary Fig. 3.

## cDNA amplification and qPCR

For the screening of cytokines and chemicals, the whole cell lysate of the hPGCLC expansion culture was subjected to reverse transcription and subsequent qPCR using a CellAmp Direct TB Green RT–qPCR kit (Takara, 3735A) following the manufacturer's instructions. For other experiments, total RNA was extracted from cells using a NucleoSpin RNA XS kit (Macherey-Nagel, U0902A) according to the manufacturer's instructions. A total of 1 ng of extracted RNA was used for reverse transcription and cDNA amplification as described in a previous report[63]. Amplified cDNA was used for qPCR with Power SYBR Green PCR master mix (Applied Biosystems, 4367659) on a CFX384 Touch Real-Time PCR detection system (Bio-Rad Laboratories). For the cytokines and chemicals screen, the whole cell lysate of the hPGCLC expansion culture was subjected to reverse transcription and subsequent qPCR using a CellAmp Direct TB Green RT–qPCR kit (Takara, 3735A) following the manufacturer's instructions. DNA sequences of the primers used for qPCR are listed in Supplementary Table 11.

## Reference genomes and annotations

Genome sequences and transcript annotations for the human genome (GRCh38.p12) were downloaded from the NCBI ftp site. Repeatmasker information for GRCh38 was downloaded from the UCSC table browser. CpG islands were from previously defined data[64] in hg18 format, which were converted into GRCh38 using the binary version of the LiftOver program (https://genome.ucsc.edu/cgi-bin/hgLiftOver). DMRs of imprinted genes were obtained from a previous study[65]. Promoter regions of genes were defined as sequences between 900 bp upstream and 400 bp downstream of the transcription start sites (TSSs). Intergenes were defined as regions between genes and excluding retrotransposons. Exons, introns and intergenes with a length of between 500 bp and 10 kb were used for the analysis.

## Bulk RNA-seq library preparation, data processing and analysis

Sequencing libraries were generated from amplified cDNA as previously described[12,63,66]. Mapping and processing were performed as previously described[63] and on the Gene Expression Omnibus (GEO) depository site (see Data availability statement). In brief, all reads were processed using Cutadapt (v.1.18)[67] for trimming of adaptor and poly-A sequences, then mapped onto GRCh38.p12 transcript references using TopHat2 (v.2.1.1)[68]. The resulting BAM files were processed with HTseq (v.0.9.1)[69] to calculate read counts per gene. Read counts per gene were divided by total read counts in all chromosomal genes to estimate the RPM values for genes. For chromosomal expression ratio analysis, the autosome (chromosome 10): total autosome ratio and the chromosome X: total autosome ratio (X:A ratio) was calculated according to a previously described method[70]. In brief, 75th percentile $\log_2(\text{RPM} + 1)$ values of the expressed genes (the maximum $\log_2(\text{RPM} + 1)$ values > 3, among the cells analysed) on the autosomes and chromosome X except for the pseudo-autosomal regions (https://www.ncbi.nlm.nih.gov/grc/human) in individual samples or cells were calculated and used as representative expression values in the samples and cells. A total of 12,880, 532 and 509 genes on total autosome, chromosome 10 and chromosome X, respectively, were used. GO analysis was performed using DAVID (v.2021)[71].

## scRNA-seq library preparation, data processing and analysis

Whole-cell suspensions collected from the expansion culture were pelleted and were subjected to Cell Multiplexing Oligo labelling. Alternatively, samples for analysis were cryopreserved as follows: whole-cell suspensions collected from the expansion culture were pelleted and were resuspended in Cell Banker Type 1 Plus for cryopreservation at −80 °C. For Cell Multiplexing Oligo labelling, after thawing in a water bath at 37 °C if needed, collected cells were resuspended in Cell Multiplexing Oligo solution and incubated for 5 min at room temperature. After incubation, cells were washed with 1% BSA (Sigma, A1519) and PBS(−) and were collected by centrifugation at 200$g$ for 10 min. Collected cells were resuspended in 1% BSA–PBS(−) containing anti-TRA-1-85 antibody (BD Bioscience, 563302, 1:20) and placed on ice for 30 min in the dark. Then, the cells were washed with PBS once and were resuspended in 1% BSA–PBS(−) containing 3 µM DRAQ7 (Abcam, ab109202). To isolate live human cells, TRA-1-85$^+$DRAQ7$^-$ cells were sorted into DMEM/F12 (Gibco, 10565-018) + 1% BSA by using a FACS Aria III. Sorted cells were used for the subsequent library construction with a Chromium Single Cell 3′ Reagent kit (v.3.1 Chemistry Dual Index), followed by sequencing on an Illumina NovaSeq 6000 platform. All steps were performed in accordance with the manufacturer's instructions. Public 10x scRNA-seq data[21,22,34] were downloaded and were subjected to subsequent processing steps. Obtained read data were processed using CellRanger (v.6.0.1) with default settings using the transcript reference as described above.

**Comparison of hPGCLC-derived cells in vitro and human germ cells in vivo.** The analysis of gene–barcode matrices was performed using the Seurat R package (v.4.2.1)[72] following online tutorials (https://satijalab.org/seurat/index.html). Cells with low-quality, putative doublets or multiplets and putative stripped nuclei were excluded from further analysis. Doublet cells were removed using the Scrublet Python package (v.0.2.3) according to previously published instructions[73]. The threshold for doublet estimation was determined based on the bimodal distribution of simulated doublet scores. Detailed information for parameters used in the quality filtering are described in Supplementary Table 12. For further analysis, in vitro datasets were merged with cells in the DDX4$^+$ germ-cell cluster extracted from human in vivo 10x datasets[21,22]. The merged count matrix was applied to noise reduction using

the RECODE.fit_transform function from the screcode Python package (v.0.1.2)[74]. The denoised data were normalized and log-transformed ($\log_e$(size-scaled (ss) UMI + 1)) using the NormalizeData (scale.factor = 100,000) function in the Seurat R package. To identify HVGs, we applied the FindVariableFeatures (selection.method = "vst,") function in the Seurat R package and detected 5,000 HVGs, which were expressed ($\log_e$(ssUMI + 1) > 2) in at least 5 cells. Using the HVGs, batch correction was performed using the RunFastMNN function in the Seurat R package. Cell clusters were characterized and visualized using the Seurat RunUMAP, FindNeighbors and FindClusters functions with the parameter "dims=1:50". The FindClusters function identified 12 clusters. After removing three small clusters consisting of non-germ cells or putative doublets or multiplets that co-expressed the markers of other cell types, ten clusters were defined by the expression pattern of key germ cell markers as VEM1, VEM2, EM1, EM2, M1, M2, M3, PLL1, PLL2 and ZPD. To select variably expressed genes during human germ-cell development, we defined 5,000 HVGs again using human in vivo datasets. For GO enrichment analysis and heatmap visualization, genes with low expression levels (cluster average of $\log_e$(ssUMI + 1) $\leq$ 1.0: 3,745 genes) were excluded. Cell-cycle scoring and annotation were performed using the CellCycleScoring function of the Seurat package based on a previously described scoring strategy[75]. The human Quick GO term 'Meiotic cell cycle' was used to score the meiotic prophase I. Cell-cycle scores for the S, G2/M and meiotic prophase I phases were calculated using the AddModuleScore function, and cells without any of these annotations were assigned to the G1 phase. RNA velocity analysis was processed using the scvelo.pp.moments(n_pcs = 30, n_neighbors = 30) and scvelo.tl.velocity(mode = 'stochastic') functions in the scVelo Python package (v.0.2.5)[76]. To analyse the allelic gene expression, the 10x transcriptome analysis data were remapped to the masked GRCh38.p12 reference using the cellranger multi pipeline in Cell Ranger (v.6.0.1) with the following parameters: "include-introns"=true, and modified star parameters "--alignEndsType EndToEnd" and "--outSAMattributes NH HI NM MD". The BAM files resulting from the mapping were processed with SNPsplit (v.0.5.0) using the "--paired" option to split the BAM files into phased alleles. The BAM files were then converted to FASTQ files using the bamtofastq command in Cell Ranger (v.6.0.1). The cellranger count pipeline was used to calculate UMI counts with default parameters except for "--include-introns" for both sets of FASTQ files, separately. UMI counts for Xa and Xi were generated using the phasing blocks defined below. Genes with low expression (total UMI counts from Xa and Xi <3 in datasets) and those overlapping with multiple phasing blocks were excluded. Subsequently, Xa allele usage was calculated as [UMI counts from Xa] over [total UMI counts from Xa and Xi] ([total UMI counts from Xa and Xi > 2]) for each gene and cell.

**Analysis of BMP-driven differentiation of *TET1* KO hPGCLCs.** Cell-Ranger output files were loaded by the Read10X and CreateSeuratObject functions of Seurat package (v.4.1.1) in R software. RECODE was applied to mitigate the course of dimensionality of scRNA-seq data through the desktop application (v.1.1.1). Doublet cells were detected using Scrublet[73] with default settings and were excluded from the data. Low-quality cells were excluded using the subset function of the Seurat package based on the following criteria: nCount_RNA > 1,0000 & nCount_RNA < 100,000 & percent.mt <10. UMI counts of the cells passing the filters described above were normalized by the NormalizeData function of Seurat with the "scale.factor = 100000" option. Highly variable genes were identified by the FindVariableFeatures function with the following options: "selection.method = "vst"", "nfeatures = 2000". Data scaling, PCA, construction of a shared nearest neighbour graph, identification of clusters and dimensional reduction by UMAP were done using the ScaleData, RunPCA, FindNeighbors, FindClusters and runUMAP functions, respectively. For the data of *TET1* KO hPGCLC-derived cells, the specific parameters in the FindNeighbors and FindClusters were as follows, respectively: "dims = 1:22" and

"resolution = 0.5". Partition-based graph abstraction (PAGA) analysis[77] was performed using the Python package scanpy (v1.9.1). For the data from ref. 34, the specific parameters in FindNeighbors and FindClusters were as follows: "dims = 1:10" and "resolution = 0.4", respectively. Data integration was performed using the IntegrateData function[78].

## Public scRNA-seq data processing and analysis
Processed scRNA-seq data of human blastocysts[16] were a gift from T. Coorens. The cell clusters were annotated in the original manuscript[16]. For the scRNA-seq data of a human gastrula[17], read data were processed using Trim_Galore! (v.0.6.3) for quality check and mapped on the human GRCh38.p12 genome with TopHat2 (v.2.1.1). Mapped bam files were converted to FPKM using Cufflinks (v.2.2.1)[79] with the "--compatible-hits-norm" option and human GRCh38.p12 transcript reference. Processed scRNA-seq data of human embryonic gut[18] were downloaded from CZ CELLxGENE[80].

## EM-seq library preparation, data processing and analysis
EM-seq libraries were generated and sequenced on an Illumina NovaSeq 6000 platform as previously described[25]. EM-seq data for the F2-*AGVT* iPS cells were previously reported[25] and were used for the subsequent processing steps. Paired-end reads were processed using Trim_Galore! (v.0.6.3) and mapped on the GRCh38.p12 genome using Bismark (v.0.22.1)[81] and Bowtie2 (v.2.3.4.1) with the "-X 2000" option. BAM files were then processed with the deduplicate_bismark script to remove PCR duplicates, and with bismark_methylation_extractor script to count methyl- and unmethyl-cytosines per CpG or CpA sequences. Post-bisulfite adaptor tagging sequence (PBAT) data for human sperm, oocytes, blastocysts[51], human fetal germ cells (female and male germ cells at 9 w.p.f.)[3] and hPGCLC-derived cells in xrOvaries (ag120 AG and ag120 AGVT)[14] were processed using Trim_Galore! and Bismark with the "--pbat" option, followed by processing with the bismark_methylation_extractor script as previously described[14]. Data from biological replicates were averaged using the following procedures and were used for the presentation unless otherwise stated. Average CpG levels in binned loci were calculated as the average of per cent mC in CpG (depth $\geq$ 4), and only bins with $\geq$4 CpG were used for analysis. Average CpA levels in binned loci were calculated as [sum of mC calls] over [sum of mC and C calls] (sum of mC and C calls$\geq$10). To obtain the average methylation level of replicates, bins with a read depth greater than three in all replicates were used for the calculation.

## Methylome analysis
**Demethylation escapees.** Mapped bam files for M1-*AGDT* c122 AGDT, M1-*AGVT* c82 AGVT, M2 c76, F1-*AGVT* c127 AGVT, F2-*AGVT* c68 and ag120 AGVT, hGC Wk9M, and hGC Wk9F were processed using MethPipe (v.3.4.3)[82] to calculate hypermethylated regions with the default setting. Overlap of the regions was calculated using the intersect command of BEDTools[83]. Annotations of the regions were calculated using Homer (v.4.11.1)[84].

**Calculating demethylation rates.** Demethylation rates of bins were calculated by dividing the decrease in the DNA methylation level of each bin, which was one of the DNA methylation intervals defined as (0, 20], (20, 40], (40, 60], (60, 80], and (80, 100], by the initial DNA methylation level of that bin.

## Oxford Nanopore Technology library preparation, mapping and processing
Genomic DNA from more than $6 \times 10^6$ iPS cells was purified using a Monarch HMW DNA Extraction kit for Tissue. Library generation was performed using an Ultra-Long Sequencing kit (Oxford Nanopore Technology (ONT), SQK-ULK001) according to the manufacturer's instructions. The sequencing was done using R9.4.1 flow cells. ONT data in fast5 format were processed using megalodon (v.2.5.0) on the GRCh38.p12

genome with the options "--guppy-config dna_r9.4.1_450bps_sup_prom. cfg", "--remora-modified-bases dna_r9.4.1_e8 sup 0.0.0 5mc CG 0", and "--outputs basecalls mappings mod_mappings mods". The mapping. bam"files from two runs were sorted and merged using samtools (v.1.15.1)[85]. SNP calling was performed using Clair3 (v.0.1.12) with the model of "r941_prom_sup_g5014" and "--platform=ont" option. The 'PASS' marked SNPs with genotypes of '1/1' and '1/2' were extracted, and the reference fasta files for GRCh38.p12 at these positions were replaced with ALT bases using the "bcftools consensus" command of bcftools (v.1.15.1) to make a F1/F2-*AGVT* custom genomic reference. Initial fast5 files were reprocessed using megalodon on the GRCh38. p12- F1/F2-*AGVT* genome, followed by Clair3 as described above.

### Haplotype phasing for ONT data and assignment of phasing blocks to Xa and Xi

'PASS' marked SNPs with phasing block marks in chromosome X were used for haplotype phasing ONT, EM-seq and transcriptome data. The bam files were subgrouped into two alleles using the "whatshap haplotag" and "whatshap split" commands of Whatshap (v.1.4). Methylation calls at the CG sequence were calculated using modbam2bed (v.0.5.3) with the "-e -m 5mC --cpg" option. From the careful observation of 5mC levels in the promoter, CGI and genome-wide 10-kb-binned regions, we found that 5mC levels of the promoters and the CGIs in one allele of the chromosome Xs tended to be more highly methylated than those in the other allele. Moreover, the allele with higher 5mC levels in the promoters and the CGIs showed relatively lower 5mC levels in the genome-wide 10-kb bin data than the other allele. Therefore, among the phasing blocks longer than 500 kb, we assigned Xa as the allele that had 5mC levels in the promoters and the CGIs lower by 50% than those in the other allele, and 5mC levels in the genome-wide 10-kb-bins higher than those in the other allele for the blocks that did not contain any promoter or CGI (Supplementary Table 6).

### Mapping and calculation of allelic 5mC levels for EM-seq data

Human, GRCh38.p12- F1/F2-*AGVT* custom reference, as described above, was masked with N at the allelic SNP positions using the bedtools maskfasta command of BEDTools. EMseq reads from F1/F2-*AGVT* samples were mapped on this genome as described above. Bam files were split by allele using SNPsplit (v.0.3.2) with the "--paired" and "--bisulfite" options and masked GRCh38.p12- F1/F2-*AGVT* custom reference. Resulting bam files for two alleles were processed with deduplicate_bismark and bismark_methylation_extractor to calculate 5mC levels per CpG or CpA as described above. Average 5mC levels for both CpG and CpA for allelic 5mC levels within bins were calculated as [sum of mC calls] over [sum of mC and C calls] (sum of mC and C calls≥10) to detect limited signals.

### Mapping and calculation of allelic 5mC levels for public ONT, EM-seq and PBAT data

ONT data in fast5 format for F2-AGVT (R9.4.1 flow cell, Ultra-Long Sequencing kit) were processed for base calling, mapping, SNP calling and haplotype phasing as described above. Allelic 5mC levels for the F2-*AGVT* iPS cells (EM-seq) and ag120 AGVT cells in xrOvaries (PBAT) were as described above and previously[25].

### ChIP–seq data processing

Public ChIP–seq data[36] were processed as previously described[86]. In brief, reads were processed using Trim Galore (v.0.4.1) or Cutadapt (v.1.9.1) to remove adaptor sequences. The truncated reads were then aligned to (GRCh38p2) using Bowtie2 (v.2.3.4.1) with the "-very-sensitive" option. Reads aligned to chromosomes 1 to 22, X, and Y were converted to the BAM format using samtools (v.1.7). BED files were obtained from the BAM files using the bamtobed command of BEDTools (v.2.29.2). BigWig files were generated from the BAM files using bamcoverage for raw count with the "--normalizeUsing CPM -bs

25" or bamcompare for IP/Input command with the "--pseudocount 1 -bs 1000" option of deepTools (v.3.5.0)[87]. In both cases, the blacklist regions[88] (https://www.encodeproject.org/files/ENCFF419RSJ) were excluded. TET1 ChIP–seq read data[39] were processed using TrimGalore to remove adapter and low-quality bases and mapped onto genome reference GRCh38.p12 with bowtie2. Mapped bam files were sorted by coordinates using samtools. Peak calling was performed using MACS2 (v.2.2.7.1) with the default settings[89]. Peaks in the promoter regions were annotated to the nearest TSSs by the annotatePeakInBatch function in the ChIPpeakAnno package in R.

### ATAC–seq data processing

Public ATAC–seq data[36] were processed as previously described[86]. In brief, adaptor sequences were trimmed from the reads using Trim-Galore (v.0.4.1) or Cutadapt (v.1.9.1). These reads were aligned using Bowtie2 (v.2.3.4.1) to GRCm38p2 with the "--very-sensitive -X 2000" option. The properly mapped reads with the flag (99, 147, 83 or 163) were extracted by awk, and mitochondrial reads were excluded. Duplicated reads were removed using the MarkDuplicates command of Picard Tools (v.2.18.23; https://broadinstitute.github.io/picard/). These de-duplicated reads were then filtered for high quality (MAPQ ≥ 30). The reads with an insert size of less than 100 bp were extracted as nucleosome-free region reads. Bed files for downstream analysis were generated by the bamtobed command of BEDTools (v.2.29.2) with the "-bedpe" option. BigWig files were generated from the BAM files using bamcoverage for raw count with the "--normalizeUsing CPM -bs 25" option of deepTools (v.3.5.0). The blacklist regions were excluded as well as ChIP–seq data processing.

Peak calling was performed using MACS (v.2.1.1) with the "--nomodel --shift -100 --extsize 200 --keep-dup all" option after shifting nucleosome-free region reads with the offset by +4 bp in the positive strand and by -5 bp in the negative strand.

### Overlap enrichment analysis

The overlap between genomic regions and annotated intervals was examined using Fisher's exact tests as implemented in the fisher.test() from R library stats (v.3.6.1). Ensembl Regulatory build annotations (v.20180516) were sourced directly from Ensembl.

### Epigenome-based clustering of *cis*-regulatory elements

ATAC–seq peaks identified using MACS2 were pooled across cell types using DiffBind (v.3.8.4)[90] to identify consensus peaks with re-centred summits. Twenty 200 bp bins surrounding peak summits were selected to capture the chromatin state ±2 kb from accessible regions, in which the $\log_2$(enrichment over input) values of ChIP–seq signals were extracted as input for dimension reduction through UMAP[91] and subsequently clustered through HDBSCAN[92] as implemented in cuML (v.23.02.00). For UMAP, correlation distances were used together with a grid search over min_dist of [0.0, 0.01, 0.1], n_neighbors of [15, 30, 50, 70, 100] and n_components of 2–10, and the resultant embeddings were all subjected to HDBSCAN clustering to identify epigenetically distinct clusters through visual inspection. For HDBSCAN, a grid search over min_cluster_size and min_samples over [50, 100, 200, 500, 1,000] were tested. In a semi-supervised fashion, individual clusters were isolated and subjected to further subclustering until the embedding no longer exhibited distinct segregation of data points for any individual epigenetic signal. We assessed the epigenetic profiles within each cluster and categorized the major clusters (that is, clusters 1, 2 and 4) as follows: active clusters were characterized by high levels of H3K4me3 and H3K27ac marks. Bivalent clusters exhibited high levels of H3K4me3 and H3K27me3 marks. Poised clusters were specifically marked by high H3K4me1 levels. Promoters that overlapped with each category of open sites were considered active, bivalent or poised promoters, whereas promoters that did not overlap with open sites were considered silent promoters. Active, bivalent and poised enhancers were defined

as the open sites within each cluster that did not overlap with promoters. Open sites that did not fall into any of these categories (active, bivalent or poised) were classified as silent enhancers. To investigate enhancer-mediated gene regulation, we focused on genes with TSSs located within a proximity of 10 kb both upstream and downstream of the enhancers.

## Statistics and reproducibility

PCA was performed using the prcomp function in R. Unsupervised hierarchical clustering was performed using the hclust function with the "method = "ward.D2"" option based on the Euclidean distance calculated by the Dist function in the amap package in R. Welch's $t$-test was performed using the t.test function with the "paired = TRUE" option in R (Fig. 1d) or the T.TEST function in Excel (Extended Data Fig. 5g). Tukey–Kramer test (Extended Data Fig. 9k) was performed using the TukeyHSD function in R. Two-sided Dunnet's test (Extended Data Fig. 10l) was performed using the SimTestDiff function with "type = "Dunnet"" option from the SimComp package in R. Paired two-sided $t$-test (Extended Data Fig. 10l) was performed using the t.test function with the "paired = TRUE" option in R. t-test with Bonferroni correction (Fig. 5g and Extended Data Fig. 12d) and Wilcoxon rank-sum test (Fig. 5f) were performed by the geom_signif function in the ggsignifr package in R. Cohen's $d$ values were calculated using the cohens_d function from the rstatix package in R (Extended Data Fig. 9m). Data from single experiments are shown in Fig. 1g, Extended Data Figs. 1b,d, 2h, 4j,k,m,n,s, 9k and 10c,d and Supplementary Fig. 1d.

## Reporting summary

Further information on research design is available in the Nature Portfolio Reporting Summary linked to this article.

## Data availability

All sequencing data generated in this study are available from the GEO database with accession number GSE232078. The raw MS data and analysis files have been deposited into the ProteomeXchange Consortium (http://proteomecentral.proteomexchange.org) through the jPOST partner repository[93] (https://jpostdb.org) and can be accessed using the dataset identifier PXD048118 (https://repository.jpostdb.org/entry/JPST002432.3). Human genome reference GRCh38.p12 (https://www.ncbi.nlm.nih.gov/datasets/genome/GCF_000001405.38/) and GRCh38.p2 (https://www.ncbi.nlm.nih.gov/datasets/genome/GCF_000001405.28/) were used in this study. The accession number for the sequencing data (bulk RNA-seq, scRNA-seq and EM-seq) generated in this study is GSE232078 (GEO database). The accession number for the ONT long-read sequence data is JGAS000690 (NBDC human database). The following public data were used in this study: processed data of human embryonic gut 10X scRNA-seq[18] are from CZ CELLxGENE (https://cellxgene.cziscience.com/collections/17481d16-ee44-49e5-bcf0-28c0780d8c4a); human gastrula Smart-Seq2 scRNA-seq data[17] are from ArrayExpress (identifier E-MTAB-9388); human fetal ovary 10x scRNA-seq data (week 11)[22] are from the GEO (GSE194266); human fetal ovary 10x scRNA-seq data (weeks 7, 9, 10, 13 and 16)[21] are from the GEO (GSE143380); human embryonic sac model 10x scRNA-seq data[34] are from the GEO (GSE134571); ChIP–seq data for human ES cell, pre-mesoderm cells, mesoderm cells, definitive endoderm cells, PGCLCs and PGCs[36] are from the GEO (GSE159654); DNA methylome data for hPGCLC induction[23] and for hPGCLC culture[24] are from the GEO (GSE86586 and GSE174485, respectively). Source data are provided with this paper.

## Code availability

Computational codes and scripts used in this study are available at GitHub (https://github.com/KU-SaitouLab/Murase2024, https://github.com/masahiro-nagano/Murase_2024_Nature) and Zenodo at (https://doi.org/10.5281/zenodo.11077386)[94].

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

**Acknowledgements** We thank the members of our laboratory for their helpful input on this study; Y. Nagai, N. Konishi, E. Tsutsumi and M. Kawasaki of the Saitou Laboratory for their technical assistance; S. Goulas for the critical review of the manuscript; S.Tarumoto and staff at the Single-Cell Genome Information Analysis Core (SignAC) in ASHBi for the RNA, EM and long-read sequence analyses; staff at XCell Science and Healios for the NCLCN human iPS cells. We dedicate this work to T. Mori. This work was supported by Grants-in-Aid for Specially Promoted Research from JSPS (17H06098, 22H04920), a Grant from HFSP (RGP0057/2018), and Grants from the Open Philanthropy Project (2018-193685) to M.S., and a Grant for Research Support Project for Life Science and Drug Discovery (Basis for Supporting Innovative Drug Discovery and Life Science Research (BINDS)) from AMED (JP22ama121021) to T.T., T.Y. and M.S.

**Author contributions** Y.M. and M.S. conceived the project, and Y.M., R.Y. and M.S. designed the experiments. Y.M. and R.Y. conducted BMP-driven hPGCLC differentiation and analysed the data. M.M., A.K., P.P., C.Y., K.M., K.O. and Y.I. assisted with the experiments, and Y.Y., Y.K., B.H., M.N., T.T. and T.Y. assisted with the data analyses. Y.M., R.Y., Y.Y., Y.K., B.H., M.N. and M.S. wrote the manuscript.

**Competing interests** M.S., Y.M. and R.Y., together with Kyoto University, have filed a provisional patent application (2023-133928) covering the propagation and differentiation of germ cells induced from human PS cells. All other authors declare no competing interests.

**Additional information**
**Correspondence and requests for materials** should be addressed to Mitinori Saitou.

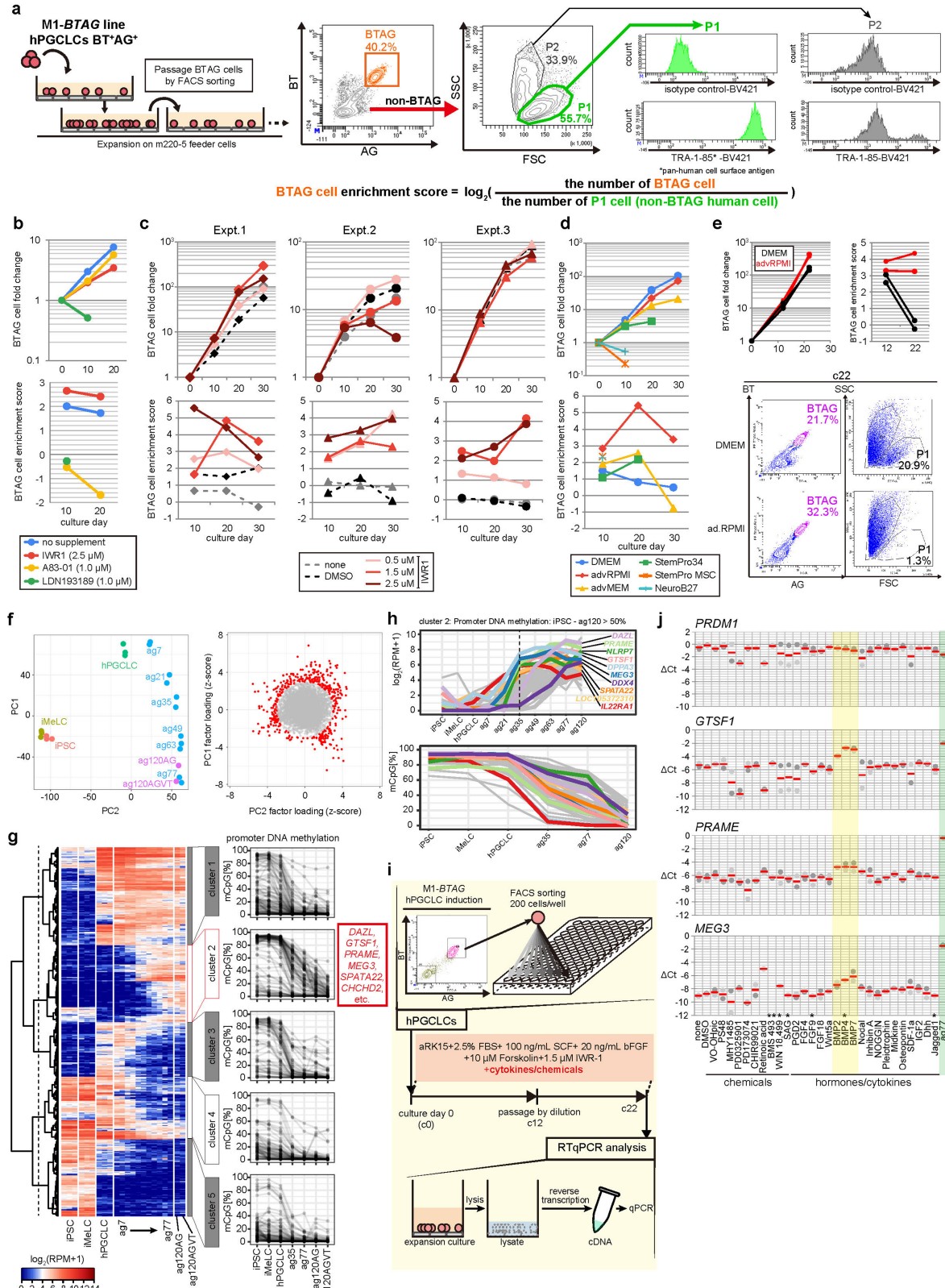

**Extended Data Fig. 1** | See next page for caption.

**Extended Data Fig. 1 | Exploration of the signaling for hPGCLC differentiation. a**, Scheme for hPGCLC expansion culture (left)[12] and flow cytometric plot for BTAG expression of the hPGCLC culture and for forward and side scatter (FSC and SSC) of the non-BTAG cells (middle). The P1 cells in the middle panel are TRA-1-85$^+$ (a human-specific antigen)[95], i.e., de-differentiated hPGCLC-derived cells, whereas a majority of the P2 cells are TRA-1-85$^-$, i.e., m220 feeders (right). Accordingly, the enrichment score is defined as $\log_2$ (the number of BT$^+$AG$^+$ cells/the number of cells in the P1 gate) (right). hPGCLCs were cultured as in[12]. See Fig. 1a for the summary of acronyms used in this study. **b−d**, hPGCLC expansion and the enrichment score of the hPGCLC culture with IWR1[96], A83-01, and LDN193189 at culture day (c) 10 and 20 (**b**), with different doses of IWR1 at c10, 20, and 30 (**c**), and with different basal media (**d**). The passages were performed using flow cytometry. The color coding is as indicated. hPGCLCs were cultured as in[12] with or without indicated chemicals. 1 biological replicate for (**b**) and (**d**), and 3 biological replicates for (**c**). **e**, hPGCLC expansion and the enrichment score of the hPGCLC culture with IWR1 (1.5 μm) in DMEM or advanced RPMI at c12 and 22 (top), and FACS plots for BTAG expression and FSC/SSC of the non-BTAG cells of the hPGCLC culture with IWR1 (1.5 μm) in DMEM or advanced RPMI at c22 (bottom). The passages were performed with dilution. The color coding is as indicated. Note that there were nearly no de-differentiated cells in the P1 gate in the culture with advanced RPMI. The data show (top)/represent (bottom) 2 biological replicates. **f**, Principal component analysis (PCA) of transcriptomes of key cell types during hPGCLC induction and hPGCLC differentiation in xrOvaries[14] (top) and the identification of genes making significant contributions [radius of standard deviations (SDs) ≥ 3] to scaled PC1 and PC2 loadings (bottom). Genes expressed in at least one sample [$\log_2$(RPM + 1) ≥ 4] were used for PCA. **g**, (left) Unsupervised hierarchical clustering (UHC) of the genes selected in (**f**) based on their expression dynamics, and (right) promoter methylation dynamics of the genes in the five clusters in (left) during hPGCLC induction and hPGCLC differentiation in xrOvaries[14]. Among the cluster 2 genes, those showing promoter 5mC-level reduction from human iPS cells (hiPSCs) to oogonia-like cells by ≥ 50% are defined as epigenetic reprogramming-activated genes (ER genes). **h**, Expression (top) and promoter methylation (bottom) dynamics of epigenetic reprogramming-activated genes (ER genes) during hPGCLC differentiation in xrOvaries. Top eight ER genes in the expression level at ag35, and *DAZL* and *DDX4* are annotated. **i**, Scheme for the screening of cytokines/chemicals that induce ER gene up-regulation. **j**, Expression of *PRDM1*, *GTSF1*, *PRAME*, and *MEG3* measured by qRT-PCR at culture day (c) 22 with the indicated cytokines/chemicals. For each gene, ΔCt from the average Ct values of two housekeeping genes, *RPLP0* and *PPIA* (set as 0), were calculated and plotted for 2 biological replicates. Mean values are shown as a red bar. *, **: Not detected or ΔCt <−10 in one or two replicates, respectively. ag77: expression values in hPGCLC-derived cell at ag77 in xrOvaries[14].

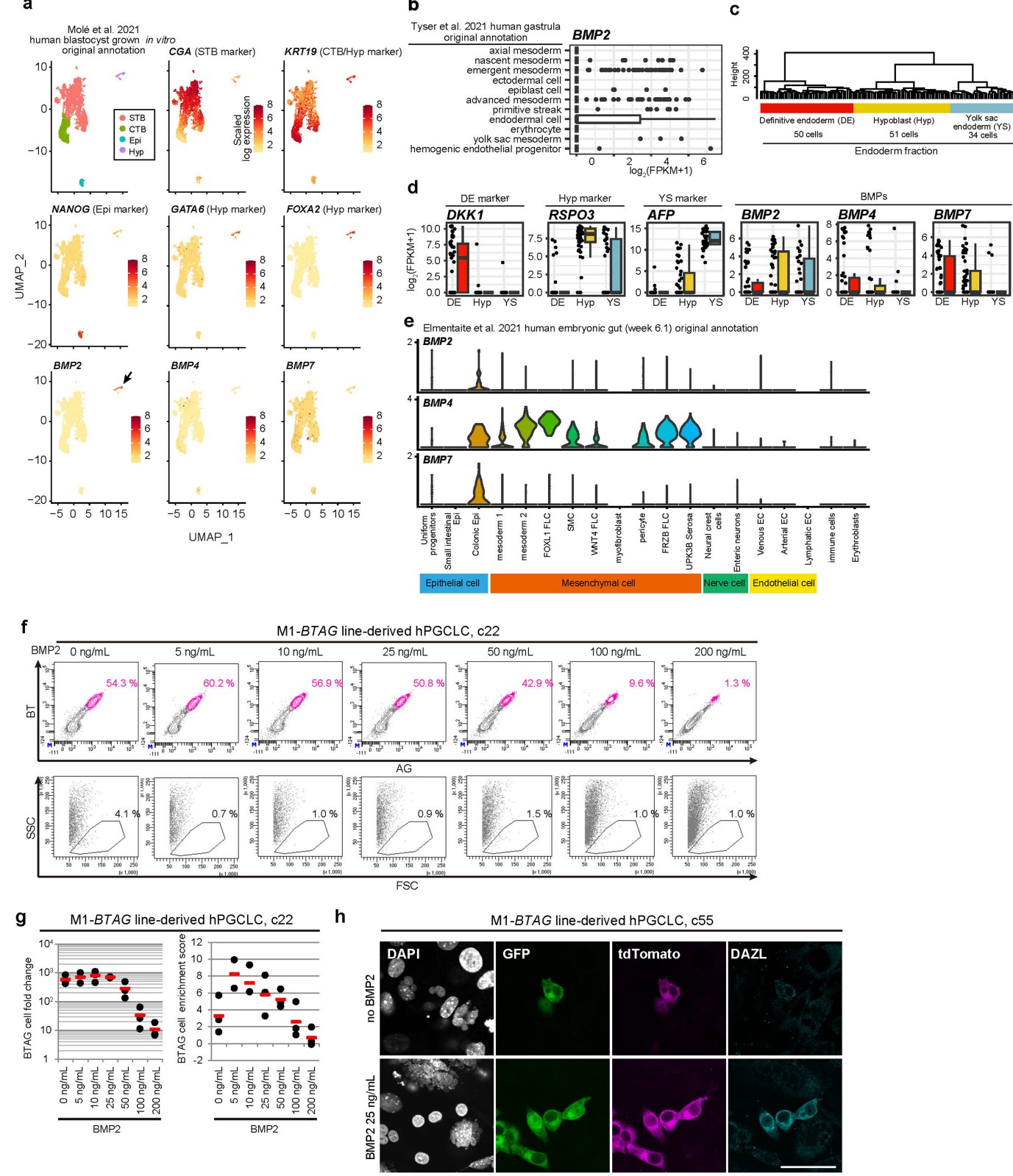

**Extended Data Fig. 2** | See next page for caption.

**Extended Data Fig. 2 | BMP signaling and hPGCLC differentiation.**
**a**, Expression of key lineage markers and BMP ligands in single cells of cultured human embryos (-E11)[16] visualized by Uniform manifold approximation and projection (UMAP) and Louvain clustering. Color coding is as indicated. STB: syncytiotrophoblast; CTB: cytotrophoblast; Epi: epiblast; Hyp: hypoblast. **b**, Expression of *BMP2* in various cell types in a gastrulating human embryo at -E16[17]. **c**, **d**, Unsupervised hierarchical clustering (UHC) (**c**) and cell-type annotation (**c**, **d**) based on key marker expression of endoderm cells in a gastrulating human embryo at -E16 in (**b**)[17] and expression of BMP ligands in each cell type (**d**). Numbers of the cells in each cluster are: n = 50 for DE; n = 51 for Hyp; n = 34 for YS. In (**b**, **d**), the upper hinges, lower hinges, and middle lines indicate 75 percentiles, 25 percentiles, and median values, respectively. The whiskers are drawn in length equal to the inter-quartile range (IQR) multiplied by 1.5. Data beyond the upper/lower whiskers are shown as dots. **e**, Expression of BMP ligands in cells composing human embryonic gut at week 6.1[18]. Note that BMP ligands are expressed at a high level in colonic (i.e., hindgut) epithelium and mucosal mesoderm. FLC: fibroblasts; SMC: smooth muscle cells. **f**, **g**, Representative FACS plots for BTAG expression (**f**, top) and for FSC/SSC fluorescence of the BT⁻AG⁻ cells (**f**, bottom), and hPGCLC fold-change (**g**, left) and the enrichment scores (**g**, right) of the hPGCLC culture with various concentrations of BMP2 with IWR1 (1.5 μM) in advanced RPMI at c22. The data represent (**f**)/show (**g**) 2 (BMP2 5 ng/mL) and 3 (BMP2 0, 10–200 ng/mL) biological replicates. The passages were performed with dilution. Note that there were nearly no de-differentiated cells in the P1 gate under all conditions. **h**, Immunofluorescence (IF) analysis of the expression of GFP (*TFAP2C-EGFP*: *AG*), tdTomato (*BLIMP1-tdTomato*: *BT*), and DAZL in hPGCLC-derived cells cultured without (top) or with (bottom) BMP2 (25 ng/ml) at c55. -19% (5/26) of BT⁺AG⁺ cells were DAZL⁺ in the culture with BMP2, whereas no DAZL⁺ cells were found in the culture without BMP2 (1 biological replicate). Bar, 50 μm.

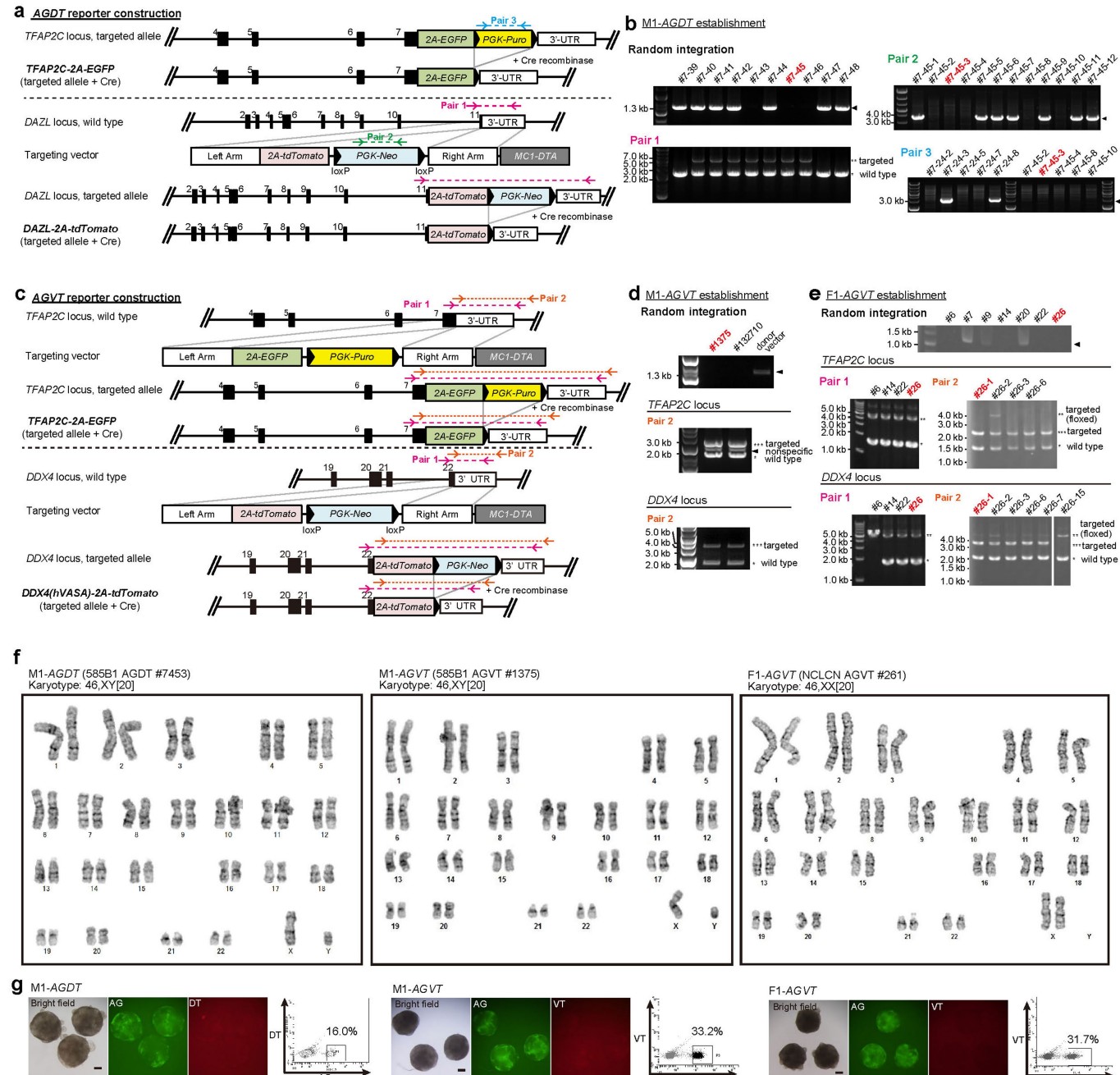

**Extended Data Fig. 3 | Generation of fluorescent reporters for hPGCLC differentiation. a**, (top) Schematic illustrations of the human *TFAP2C* locus with knock-in of the *2A-EGFP* and *PGK-Puro* cassette[11] and the same locus with the excision of *PGK-Puro* by Cre-recombinase. (bottom) Schematic illustrations of the human *DAZL* locus, the *DAZL* targeting vector for knocking in the *2A-tdTomato* and *PGK-Neo* cassette, the knocked-in locus, and the knocked-in locus with the excision of *PGK-Neo* by Cre-recombinase. Positions of the primer pairs for the screening by PCR of the genotypes are shown. Black boxes indicate the exons. **b**, Screening by PCR of the targeted alleles for *DAZL-2A-tdTomato* (*DT*) and *TFAP2C-2A-EGFP* (*AG*), and of random integration of the targeting vectors. Targeted: bands for the targeted allele; wild-type: bands for the wild-type allele; arrowheads: random integration of the targeting vectors. The 585B1-AGDT #7453 line (M1-*AGDT*) was selected for subsequent experiments. **c**, (top) Schematic illustrations of the human *TFAP2C* locus, the *TFAP2C*-targeting vector for knocking in the *2A-EGFP* and *PGK-Puro* cassette, the knocked-in locus, and the knocked-in locus with the excision of *PGK-Puro* by Cre-recombinase. (bottom) Schematic illustrations of the human *DDX4* (human VASA homolog) locus, the *DDX4* targeting vector for knocking in the

*2A-tdTomato* and *PGK-Neo* cassette, the knocked-in locus, and the knocked-in locus with the excision of *PGK-Puro* by Cre-recombinase. Positions of the primer pairs for the screening by PCR of the genotypes are shown. Black boxes indicate the exons. **d**, **e**, Screening by PCR of the targeted alleles for *TFAP2C-2A-EGFP* (*AG*) and *DDX4* (human VASA homolog)-*2A-tdTomato* (*VT*), and of random integration of the targeting vectors. Targeted: bands for the targeted allele; wild-type: bands for the wild-type allele; arrowheads: random integration of the targeting vectors. The 585B1-AGVT #1375 line (M1-*AGVT*) (**d**) and the NCLCN-AGVT #26-1 line (F1-*AGVT*) (**e**) were selected for subsequent experiments. **f**, Representative result for the G-band analysis of M1-*AGDT*, M1-*AGVT*, and F1-*AGVT* bearing normal karyotypes (46, XY or 46, XX). For each line, 20 cells in 1 biological replicate were analyzed, and all showed normal karyotypes. **g**, Bright-field and fluorescence [AG and DT or VT] images and flow cytometric plots for AGDT or AGVT expression of the iMeLC aggregates induced for hPGCLCs for 6 days from the M1-*AGDT* (left), M1-*AGVT* (middle), and F1-*AGVT* (right) lines. Bar, 200 μm. The images represent 4 (M1-*AGDT*), 2 (M1-*AGVT*), and 8 (F1-*AGVT*) biological replicates.

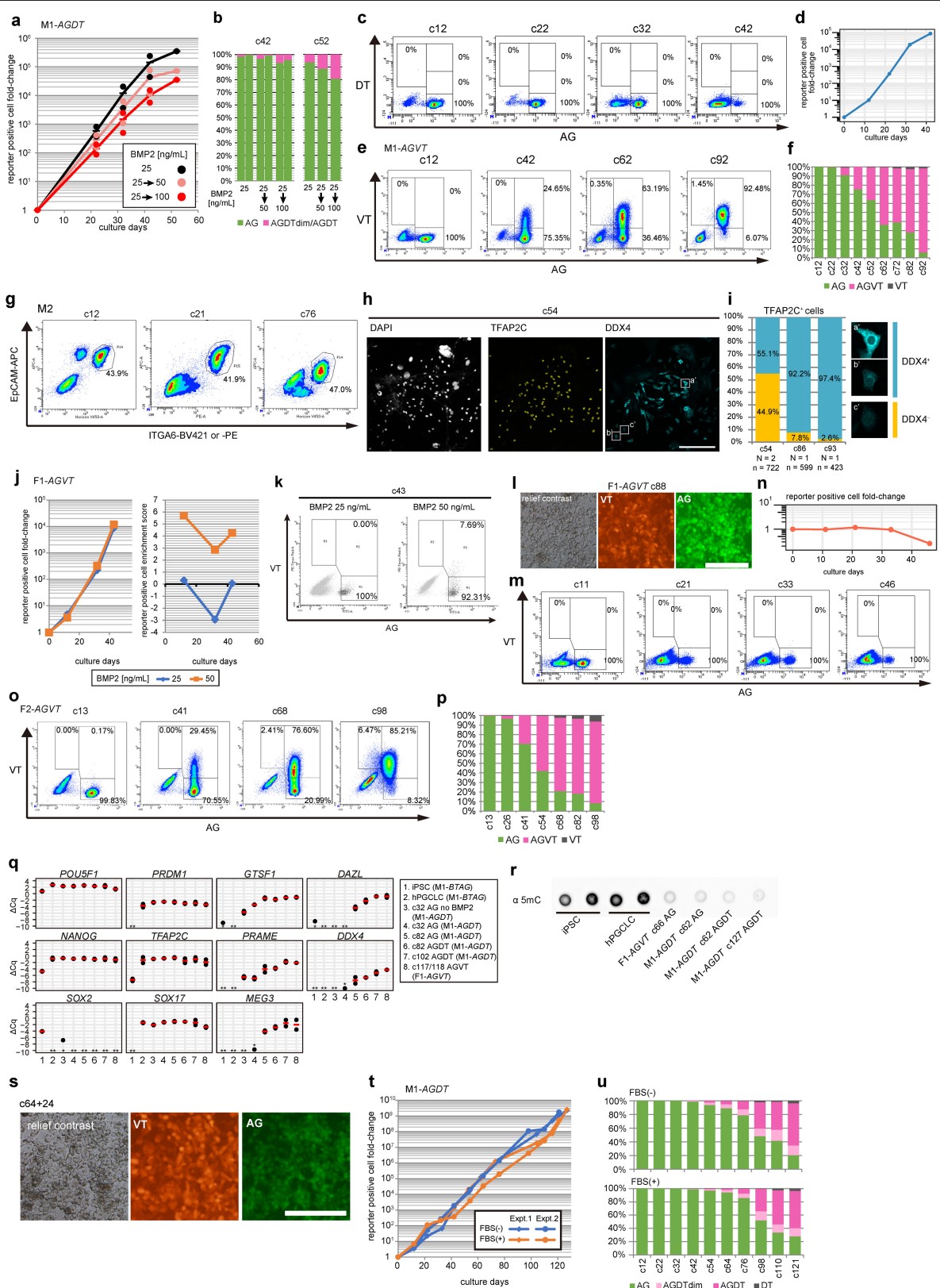

**Extended Data Fig. 4** | See next page for caption.

**Extended Data Fig. 4 | BMP signaling promotes hPGCLC differentiation.**
**a**, **b**, Growth curve (**a**) and proportion of cells with the indicated fluorescence-marker expression at c42 and c52 (**b**) during BMP-driven M1-*AGDT* hPGCLC differentiation with varying concentrations of BMP2 as indicated (2 biological replicates). **c**, **d**, Flow cytometric plots for AGDT expression at the indicated culture days (**c**) and growth curve (**d**) of M1-*AGDT* hPGCLC-derived cells cultured without BMP2 (2 biological replicates). **e**, **f**, Flow cytometric plots for AGVT expression (**e**) and proportion of cells with the indicated fluorescence-marker expression (**f**) at the indicated culture days during BMP-driven M1-*AGVT* hPGCLC differentiation (2 biological replicates). **g**–**i**, Flow cytometric plots for EpCAM and ITGA6 expression (**g**), IF analysis of TFAP2C and DDX4 expression (**h**), and proportion of DDX4$^+$ cells (**i**) at the indicated culture days during BMP-driven M2 hPGCLC differentiation (2 biological replicates). In (**i**), the numbers of experiments (N) and of cells analyzed (n), and typical images for the positivity of DDX4 staining are shown. Bar, 200 μm. **j**, **k**, Growth curve and enrichment scores (**j**) and flow cytometric plots for AGVT expression at c43 (**k**) during BMP-driven F1-*AGVT* hPGCLC differentiation with 25 ng/ml or 50 ng/ml of BMP2 (1 biological replicate). **l**, Relief contrast and fluorescence (VT and AG) images of F1-*AGVT* hPGCLC-derived cells at c88 (8 biological replicates). Bar, 200 μm. **m**, **n**, Flow cytometric plots for AGVT expression at the indicated culture days (**m**) and growth curve (**n**) of F1-*AGVT* hPGCLC-derived cells cultured without BMP2 (1 biological replicate). **o**, **p**, Flow cytometric plots for AGVT expression (**o**) and proportion of cells with the indicated fluorescence-marker expression (**p**) at the indicated culture days during BMP-driven F2-*AGVT* hPGCLC differentiation (2 biological replicates). **q**, Expression of the indicated genes in the indicated cells measured by qRT-PCR (2 biological replicates). Quantification was as in Extended Data Fig. 1j. **r**, Dot blot analysis of the genomic 5mC level in the indicated cells (2 biological replicates for hiPSCs and hPGCLCs, and 1 biological replicate the other cells). **s**, Relief contrast and fluorescence (VT and AG) images of F1-*AGVT* hPGCLC-derived cells frozen at c64 and thawed and cultured for an additional 24 days (1 biological replicate). Bar, 200 μm. **t**, **u**, Growth curve (**t**) and proportion of cells with the indicated fluorescence-marker expression (**u**) during BMP-driven M1-*AGDT* hPGCLC differentiation with or without FBS (2 biological replicates).

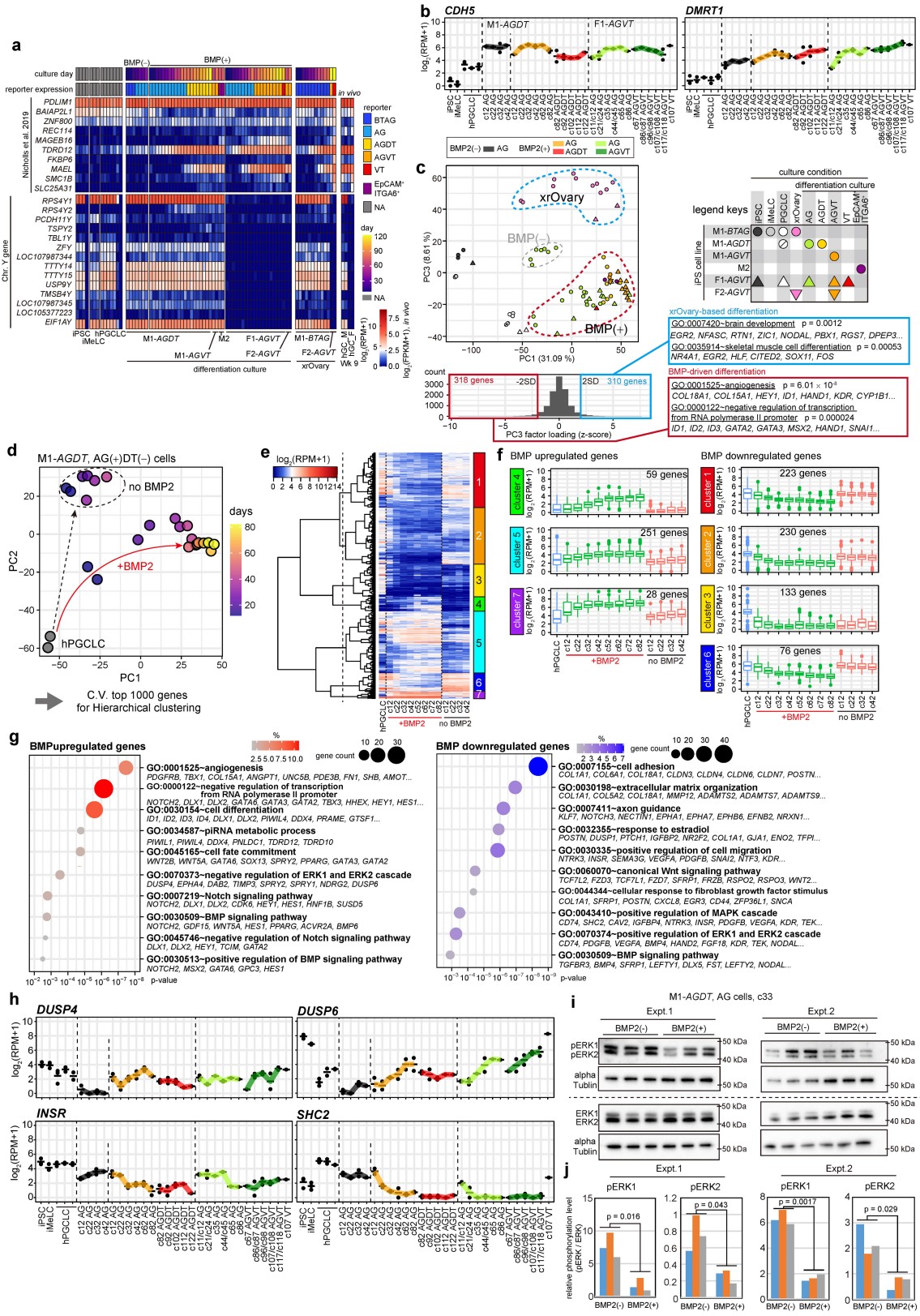

**Extended Data Fig. 5 |** See next page for caption.

**Extended Data Fig. 5 | Identification of distinctive transcriptional processes driven by BMP signaling. a**, Heatmap showing the expression levels of 13 previously reported genes that show up-regulation in gonadal germ cells (*DAZL* and *DDX4* are excluded)[19] (top), and the unique genes on the Y chromosome (bottom) in the indicated cell types (see Supplementary Table 2 for full sample information). Color coding is as indicated. NA: not applicable. **b**, Expression dynamics of *CDH5* and *DMRT1*, the genes used as markers for human germ cells from the migration stage onward[20], during hPGCLC induction and BMP-driven hPGCLC differentiation. The average (bar) and replicate (circles) values are shown (see Supplementary Table 2 for full sample information). The data for iPSCs, iMeLCs were with M1-*BTAG*, and the data for hPGCLCs were with the M1-*BTAG*, M1-*AGDT*, and F1-*AGVT* lines. Color coding is as indicated. **c**, PC1–PC3 plane of the PCA of transcriptomes during hPGCLC induction and BMP-driven and xrOvary-based hPGCLC differentiation in Fig. 3b (left, top), and the GO enrichments with *p* values of genes contributing to the negative [standard deviation (SD) < − 2: BMP-up-regulated genes] and positive [SD > 2: xrOvary-up-regulated genes] scores of PC3 (left, bottom; right). Color coding is as indicated. **d**, PCA of M1-*AGDT* hPGCLC-derived AD+DT− cells cultured with or without BMP2. The color coding is as indicated. Genes expressed in at least one sample [$\log_2(RPM + 1) \geq 4$] were used for PCA. **e**, UHC of highly variable genes (top 1,000 genes with high coefficient of variance) in (**d**) based on their expression dynamics. **f**, Box plots of the expression dynamics of the 7 gene clusters in (**e**) during hPGCLC culture with or without BMP2. The 7 gene clusters are classified into those showing progressive up- (clusters 4, 5, 7) or down- (clusters 1, 2, 3, 6) regulation during BMP-driven hPGCLC differentiation. Numbers of genes in each cluster are: n = 223 for cluster 1; n = 230 for cluster 2; n = 133 for cluster 3;

n = 59 for cluster 4; n = 251 for cluster 5; n = 76 for cluster 6; n = 28 for cluster 7. The upper hinges, lower hinges, and middle lines indicate 75 percentiles, 25 percentiles, and median values, respectively. The whiskers are drawn in length equal to the inter-quartile range (IQR) multiplied by 1.5. Data beyond the upper/lower whiskers are shown as dots. **g**, Gene ontology (GO) enrichments and representative genes in up- (clusters 4, 5, 7) (left) and down- (clusters 1, 2, 3, 6) (right) regulated genes. *p*-values are provided by Fisher's exact test. The color coding is as indicated. **h**, Expression dynamics of *DUSP4* and *DUSP6* (GO:0070373-negative regulation of ERK1 and ERK2 cascade), and *INSR* and *SHC2* (GO:0043410-positive regulation of MAPK cascade) during hPGCLC induction and BMP-driven hPGCLC differentiation. The average (bar) and replicate (circles) values are shown (see Supplementary Table 2 for full sample information). The data for iPSCs, iMeLCs were with M1-*BTAG*, and the data for hPGCLCs were with the M1-*BTAG*, M1-*AGDT*, and F1-*AGVT* lines. **i**, Western blot analysis of the levels of phosphorylated or total ERK1 and 2 in M1-*AGDT* hPGCLC-derived cells at c33 cultured with or without BMP2. 3 independent cultures were analyzed for 2 biological replicates. αTUBLIN were used for the loading control. For the gel source data, see Supplementary Figure 3. pERK: phosphorylated ERK. **j**, Quantification of pERK1 and 2 levels normalized by αTUBLIN in M1-*AGDT* hPGCLC-derived cells at c33 cultured with or without BMP2 in (**h**). The average fold-differences of the Western blot signals for pERK1 and pERK2 were -4.5-fold and -2.9-fold (Expt. 1) and -4.3-fold and -3.9-fold (Expt. 2), respectively. *p* values with two-sided Welch's *t*-test are shown. Data from 2 independent experiments with 3 biological replicates were shown in (**i**) and (**j**).

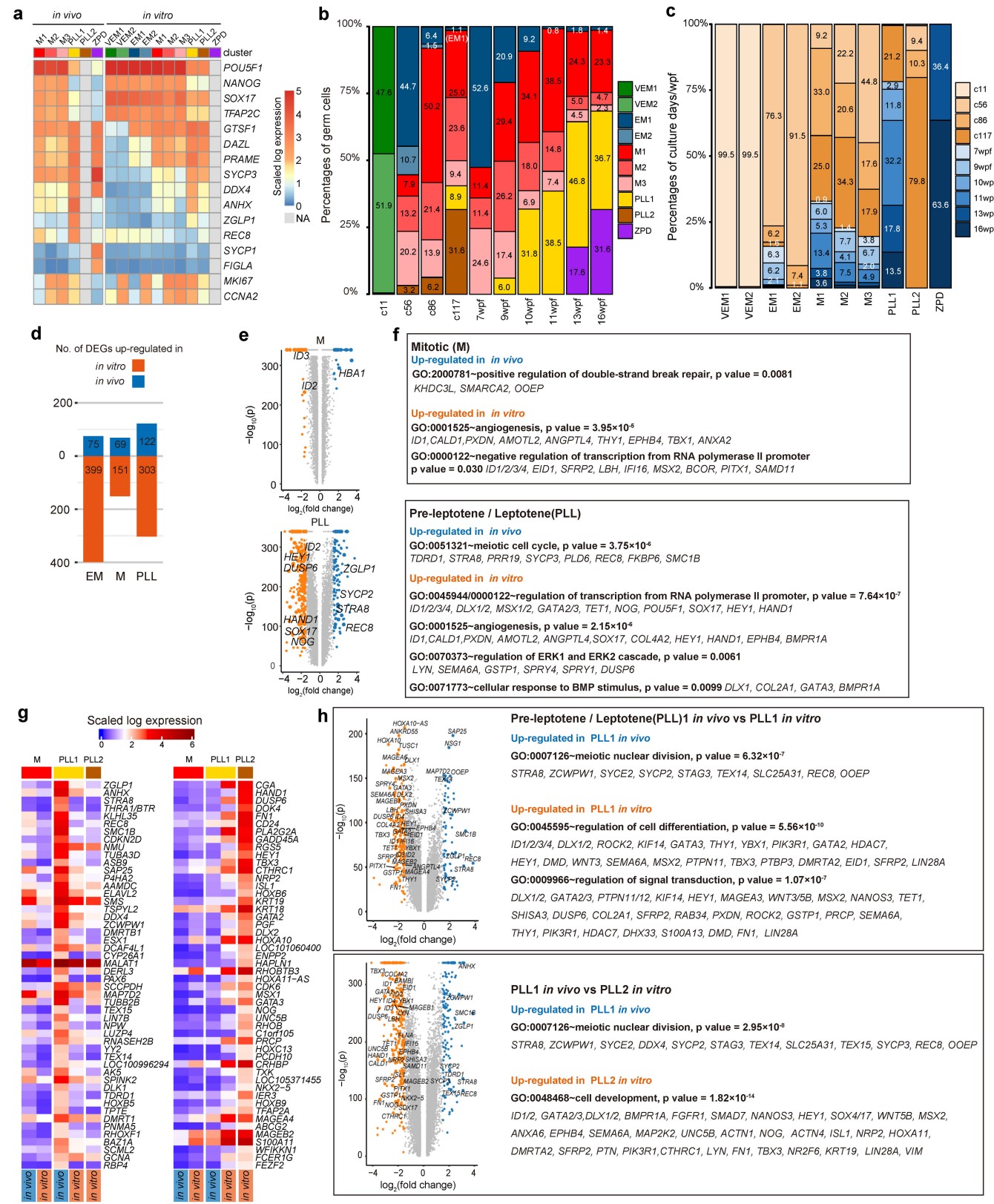

**Extended Data Fig. 6** | See next page for caption.

**Extended Data Fig. 6 | scRNA-seq analysis of BMP-driven female hPGCLC differentiation. a**, Heatmap showing the expression levels of key genes in oogonia/fetal oocytes in vivo (left) and F1-*AGVT* hPGCLC-derived cells in vitro (right) classified into 10 clusters in Fig. 3c. The actual expression levels [log2(normalized read counts+1)] are provided in Source Data Extended Data Fig. 6. The color coding is as indicated. **b**, **c**, Proportion of the 10 clusters in Fig. 3c in the indicated samples (**b**) and of the indicated samples in each cluster (**c**). The actual percentages of major clusters (**b**)/culture days/weeks post-fertilization (wpf) (**c**) are shown within the histogram. The full information is provided in Source Data Extended Data Fig. 6. The color coding is as indicated.

**d**–**f**, The numbers of differentially expressed genes (DEGs) between in vivo and in vitro cell types in the EM, M, and PLL clusters (**d**), volcano plots for the comparisons in the M and PLL clusters (**e**), and the GO enrichments with $p$ values of DEGs in the M and PLL clusters (**f**). In (**e**, **f**), $p$-values are provided by Fisher's exact test. **g**, Heatmap showing the expression levels of PLL1 (left) or PLL2 (right) signature genes (top 50 genes highly expressed in PLL1 or 2 relative to all other clusters) in the indicated samples. The color coding is as indicated. **h**, GO enrichments with $p$ values of DEGs between PLL1 in vivo and in vitro cells (top) and between PLL1 in vivo and PLL2 in vitro cells (bottom). $p$-values are provided by Fisher's exact test.

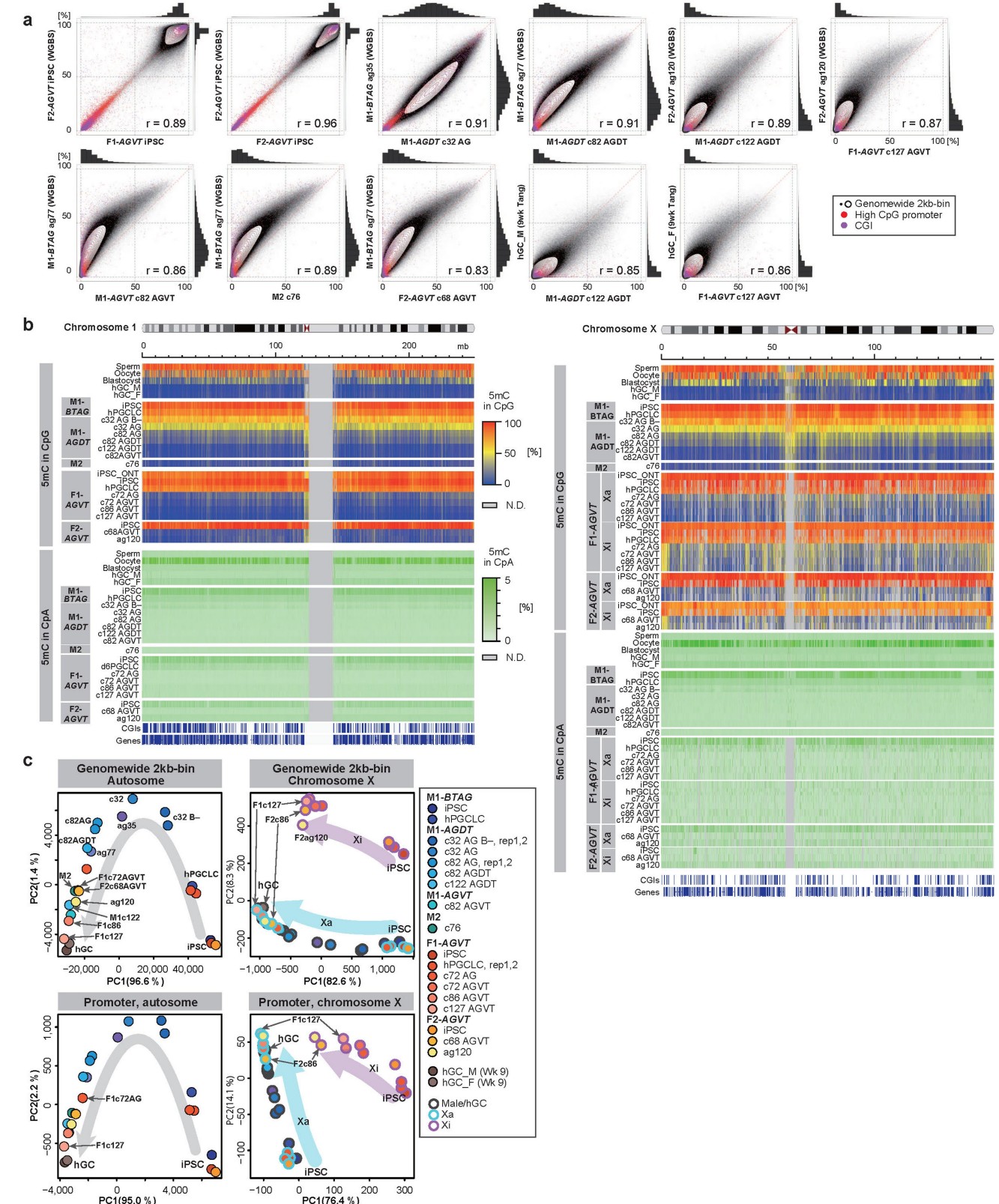

**Extended Data Fig. 7 | DNA methylome reprogramming during BMP-driven hPGCLC differentiation. a**, Scatter-plot comparisons (contour representation) of the 5mC levels (genome-wide 2-kb bins), combined with histogram representation (top and right of the scatter plots), between the indicated cell types. Note that genome-wide 5mC profiles of F1 and F2-*AGVT* hiPSCs measured by EM-seq are highly similar to those of F2-*AGVT* hiPSCs measured by whole genome bisulfite sequence (WGBS)[25]. **b**, Heatmap of the 5mC [CpG (top) or CpA

(bottom)] levels on chromosome 1 (left) and chromosome X (right) in the indicated samples. For chromosome X (right), data were generated using the reads overlapping with allelic SNPs. mb, megabases. The color coding is as indicated. N.D.: bins without enough CpGs (4) with read depth ≥4 in CpG or bins without enough mC + C calls (≥10) in CpA. **c**, PCA of the indicated samples using the 5mC levels on the autosome-wide (left) or Xa- and Xi-wide (right) 2-kb bins (top) and promoters (bottom). The color coding is as indicated.

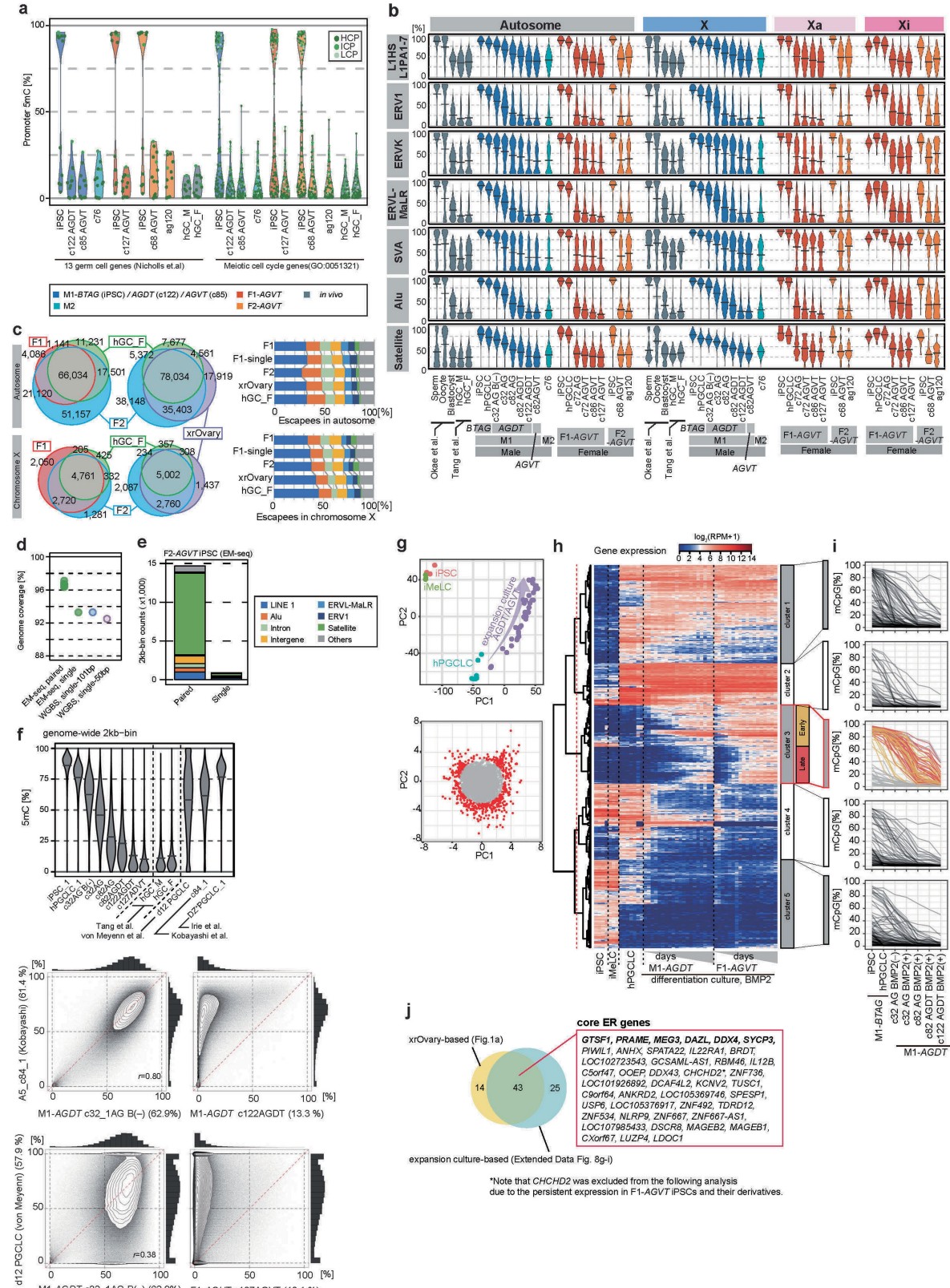

**Extended Data Fig. 8** | See next page for caption.

**Extended Data Fig. 8 | DNA methylome reprogramming and identification of core ER genes during BMP-driven hPGCLC differentiation. a**, Violin plots of the promoter 5mC-level dynamics of 13 previously reported genes that show up-regulation in gonadal germ cells[19] (left) and genes included in the GO term "meiotic cell cycle" (GO: 0051321) in the indicated cells during BMP-driven hPGCLC differentiation and in in vivo germ cells[3]. All relevant promoters are classified into H/I/LCP (high/intermediate/low CpG promoter) and plotted. **b**, Violin plots of the average 5mC levels on the indicated repeat elements in the indicated cell types (see Supplementary Table 2 for full sample information). Bars represent the average values. The DNA methylome data for human spermatozoa, oocytes, and blastocytes are from[51] and those for human male and female germ cells at 9 wpf are from[3]. **c**, Venn diagram showing the relationships of the DNA demethylation escapees among the indicated samples, and composition of the escapees in the indicated samples (**d**: male samples; **e**; female samples, with autosomes and X chromosomes separately indicated). Color coding is as indicated. **d**, Genome coverage (%) by EM-seq with paired-end sequencing (this study), EM-seq with computationally simulated single-end sequencing, whole genome bisulfite sequence (WGBS) with single-end 101 bp sequencing[14], and WGBS with single-end 50 bp sequencing[3]. **e**, Annotation of differentially covered regions between paired-end and single-end sequencing in (**d**). Color coding is as indicated. **f**, (top) Violin plots of the average 5mC levels (genome-wide 2 kb bins) in the indicated cell types. Bars represent the average values. (bottom) Scatter-plot comparisons (contour representation) of the 5mC levels (genome-wide 2-kb bins), combined with histogram representation (top and right of the scatter plots), between the indicated cell types. Note that DAZL+ PGCLCs by Irie et al.[20] are highly methylated (~76%) and that hPGCLCs by von Meyenn et al.[23] and cultured hPGCLCs by Kobayashi et al.[24] remain methylated (the average 5mC levels of 57.9% and 61.4%, respectively) and show a methylome similar to that in M1-*AGDT* hPGCLC-derived cells at c32 cultured without BMP2 (AG B−). **g**, PCA of transcriptomes of key cell types during hPGCLC induction and BMP-driven hPGCLC differentiation (top) and the identification of the genes with significant contributions [radius of standard deviations (SDs) ≥ 3] to scaled PC1 and PC2 loadings (bottom). Genes expressed in at least one sample [$\log_2$(RPM + 1) ≥ 4] were used for PCA. **h**, **i**, UHC of the genes selected in (**g**) based on their expression dynamics (**h**), and promoter methylation dynamics of the genes in the five clusters in (**h**) (**i**) during hPGCLC induction and BMP-driven hPGCLC differentiation. Among the cluster 3 genes, those showing promoter 5mC-level reduction from hiPSCs to oogonia-like cells by ≥ 50% are defined as epigenetic reprogramming-activated genes (ER genes), which are classified into early and late ER genes based on their expression dynamics. **j**, Venn diagram showing the overlap of ER genes defined for xrOvaried-based (Extended Data Fig. 1f–h) and BMP-driven (Extended Data Fig. 8g–i) hPGCLC differentiation.

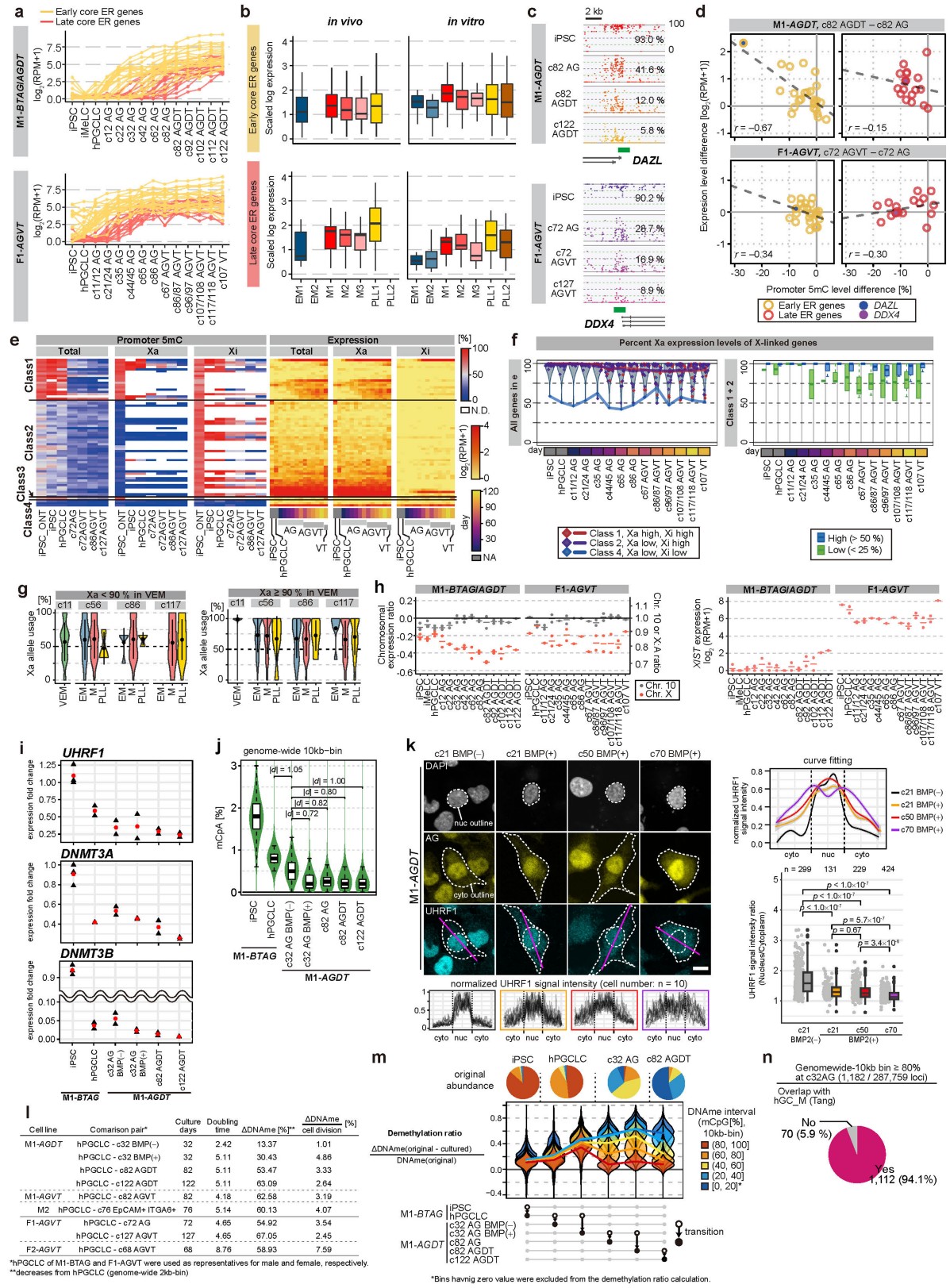

**Extended Data Fig. 9** | See next page for caption.

**Extended Data Fig. 9 | ER gene regulation and XCR. a**, Expression dynamics of core ER genes (early: yellow; late: red) (Extended Data Fig. 8f–i) during BMP-driven M1-*AGDT* (top) and F1-*AGVT* (bottom) hPGCLC differentiation. **b**, Box plots showing the expression of core ER genes (n = 42) in in vitro and in vivo EM, M, and PLL cells in Fig. 3c. The upper hinges, lower hinges, and middle lines indicate 75 percentiles, 25 percentiles, and median values, respectively. The whiskers are drawn in length equal to the inter-quartile range (IQR) multiplied by 1.5. Data beyond the upper/lower whiskers are not shown. **c**, 5mC-level tracks of *DAZL* (top) and *DDX4* (bottom) loci in the indicated cell types. Green bars represent the promoters [+400 bp and −900 bp of the transcription start sites (TSSs)], and their 5mC levels are indicated. **d**, Scatter-plot representations of the relationship between promoter-5mC-level differences and expression-level differences for early (yellow, left) and late (red, right) ER genes between c82 AG$^+$DT$^-$ and DT$^+$ cells (top) and between c72 AG$^+$VT$^-$ and VT$^+$ cells (bottom). Regression lines are indicated. **e**, Heatmap of the promoter 5mC (left) and expression (right) level dynamics of the X-linked genes during BMP-driven F1-*AGVT* hPGCLC differentiation. The Xa and Xi allelic data were generated using the reads overlapping allelic SNPs. The genes were classified according to their promoter 5mC levels on the Xa and Xi alleles in hiPSCs: class 1 genes with high (≥ 50%) 5mC on both Xa and Xi (16 genes), class 2 genes with low (<50%) 5mC on Xa and high 5mC on Xi (40 genes), a class 3 gene with high 5mC on Xa and low 5mC on Xi (*XIST*), and class 4 genes with low 5mC on both Xa and Xi (3 genes) (Supplementary Table 8). The color coding is as indicated. N.D., promoters with insufficient read depths. Note that there were no informative single nucleotide polymorphisms (SNPs) that discriminate *XIST* expression from parental alleles with the 3-prime RNA-seq analysis[63]. **f**, Expression dynamics from the Xa and Xi alleles during BMP-driven F1-*AGVT* hPGCLC differentiation. (left) Proportions of the expression from the Xa allele in the three gene classes in (**e**) are plotted, with individual values plotted as diamonds and their averages shown as colored lines. The distributions of the Xa ratio of all genes are shown as violin plots. Data points at 100% are dispersed within the range of 5% for better visualization. Raw data are available in (Supplementary Table 8). (right) Proportions of the expression from the Xa allele of the class 1 and 2 genes are box-plotted, with genes retaining high (≥ 50%) and low (≤ 25%) 5mC levels in c117/118 AG$^+$VT$^+$ cells colored blue and green, respectively. **g**, Xa allele usage of genes expressed similarly from Xa and Xi in VEM cells at c11 (% Xa usage <90%; 8 genes) (top) or those expressed predominantly from Xa in VEM cells at c11 (% Xa usage ≥ 90%; 34 genes) (bottom) in the indicated cell types. Xa: active X chromosome. VEM, M, and PLL are defined in Fig. 3c. **h**, Dynamics of the X chromosome:autosome ratio (X:A ratio) of gene-expression levels (top) and *XIST* expression (bottom)

during BMP-driven M1-*AGDT* (left) and F1-*AGVT* (right) hPGCLC differentiation, based on the bulk RNA-seq data. The ratios of the 75%-tile expression values of the genes from the chromosome X or chromosome 10, relative to those of all genes are plotted in the log$_2$ (left) or linear (right) scale. **i**, Absolute expression-level fold-changes of *UHRF1*, *DNMT3A*, and *DNMT3B* during BMP-driven hPGCLC specification and differentiation. The data in Fig. 3a are used and the value in one replicate in hiPSCs is set as one. 3 biological replicates for hiPSCs and 2 biological replicates for the other cells. The red circles present the average. **j**, Violin plots for the methylated CpA levels (genome-wide 10-kb bin, n = 290,409) in the indicated cell types. Absolute values of effect sizes (Cohen's d-values) are as follows: 1.05 for hPGCLC, 0.72 for c32 AG BMP (+), 0.82 for c82 AG, 0.80 for c82 AGDT, 1.00 for c122. The upper hinges, lower hinges, and middle lines indicate 75 percentiles, 25 percentiles, and median values, respectively. The whiskers are drawn in length equal to the inter-quartile range (IQR) multiplied by 1.5. Data beyond the upper/lower whiskers are not shown. **k**, IF analysis of the expression and subcellular localization of UHRF1 co-stained with GFP (*TFAP2C-EGFP*: AG) and DAPI in the indicated cell types (left, top) (1 replicate for each culture), and normalized UHRF1 signal intensities across the nucleus and cytoplasm (magenta lines) of randomly chosen 10 cells (left, bottom) and their curve fitting representation by Generalized additive model with grey error bands indicating 95% confidence intervals (right, top). The outlines of the nucleus (nuc) and cytoplasm (cyto) were determined based on the visual inspection of the DAPI and AG signals (dotted lines), respectively. In (left, top), AG appeared to be enriched in the nucleus, but the reason was unclear. (right, bottom) Quantification of the nuclear/cytoplasmic ratio of UHRF1 by an automated image analysis was shown in. The numbers of cells measured in each sample (n) were indicated. $p$ values provided by Tukey-Kramer test are as follows: <1.0 × 10$^{-7}$ for comparison using c21 BMP2 (−), 0.67 for c21 BMP2 (+) vs c50 BMP2 (+), 5.7 × 10$^{-7}$ for c21 BMP2 (+) vs c70 BMP2 (+), 3.4×10$^{-6}$ for c50 BMP2 (+) vs c70 BMP2 (+). The upper hinges, lower hinges, and middle lines indicate 75 percentiles, 25 percentiles, and median values, respectively. The whiskers are drawn in length equal to the inter-quartile range (IQR) multiplied by 1.5. Data beyond the upper/lower whiskers are shown as dots. Bar, 10 μm. **l**, Doubling times, 5mC demethylation levels, and 5mC demethylation rates per cell division in the indicated culture periods. **m**, 5mC demethylation ratios of genomic bins bearing different 5mC levels in the originated cell types during the indicated cell-type transitions. Pie charts indicate the proportion of each bin in the originated cell types. The color coding is as indicated. **n**, A pie chart showing overlap of the bins bearing ≥ 80% 5mC levels in c32 cells cultured with BMP2 with the DNA demethylation escapees in human germ cells in vivo.

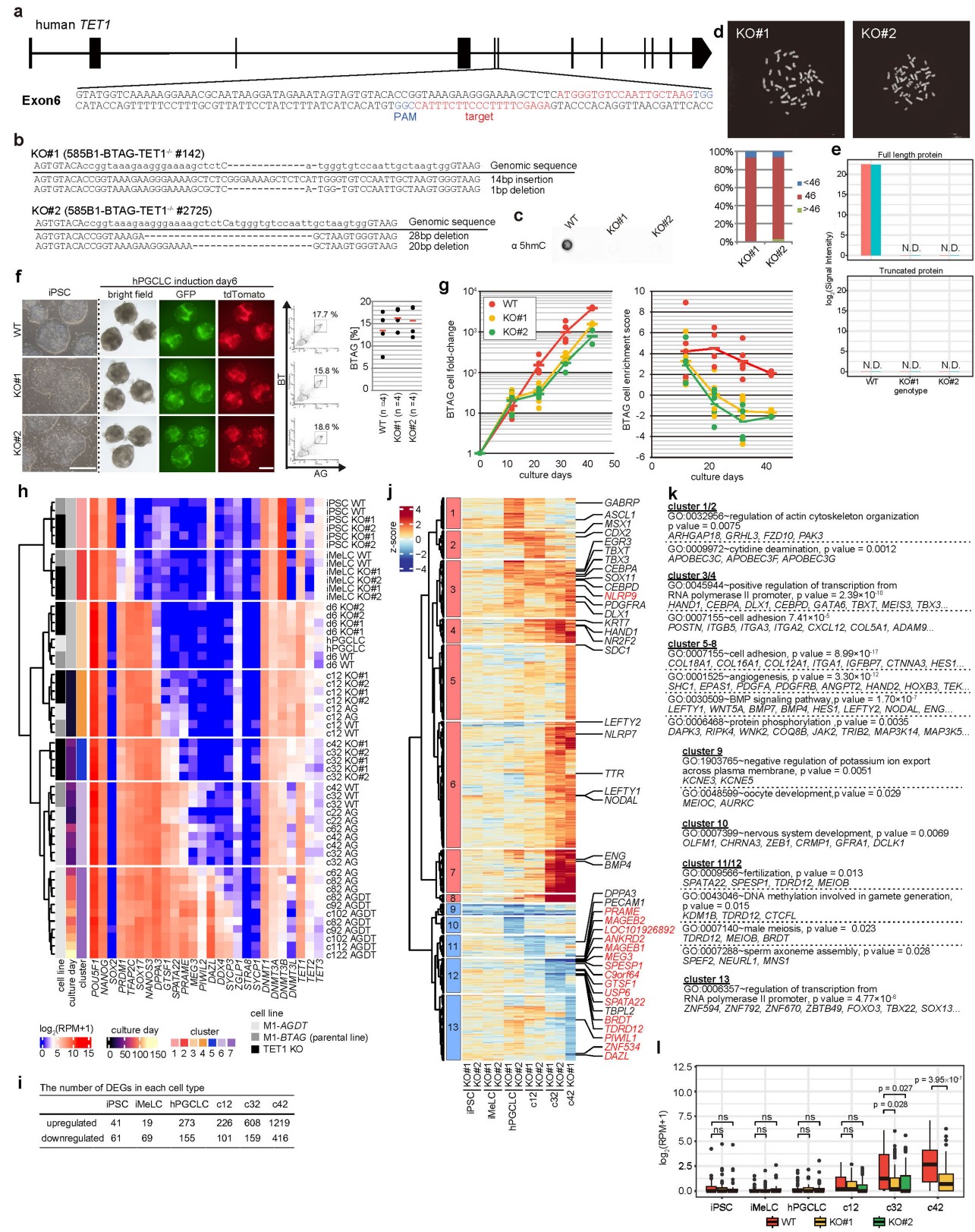

**Extended Data Fig. 10** | See next page for caption.

**Extended Data Fig. 10 | Generation of *TET1* knockout hiPSCs and analysis of BMP-driven *TET1*-knockout hPGCLC differentiation. a**, Scheme of the human *TET1* locus, with the illustration of PAM (protospacer adjacent motif) and guide RNA sequences in the exon 6. Black boxes indicate the exons. **b**, Sequences of the targeted loci in two *TET1* knockout (KO) cell lines [*TET1* KO#1 and #2 (M1-*BTAG TET1*$^{-/-}$ #142 and #2725)]. **c**, Dot-blot analysis of genomic 5hmC levels in wild-type and *TET1* KO hiPSCs (1 replicate for each line). **d**, Karyotype of *TET1* KO#1 and #2 hiPSCs (top: chromosome spreads; bottom: percentage of cells with 46 or other chromosome numbers) (1 biological replicate for each line). Bar, 10 μm. **e**, Mass spectrometric analysis [log$_2$(signal intensities)] for TET1 and its truncated protein potentially derived from the *TET1* KO allele in wild-type and *TET1* KO cells. Peptides from the full-length (top), but not the truncated (bottom), TET1 were detected from the wild-type cells (red and blue bars), whereas neither form was detected from the *TET1* KO cells (2 biological replicates). **f**, Induction of hPGCLCs from wild-type (M1-*BTAG*) and *TET1* KO#1 and #2 hiPSCs. Photomicrographs of hiPSCs and iMeLC aggregates induced for hPGCLCs for 6 days (bright-field and fluorescence images for AG and BT) (left), their flow cytometric plots for AG and BT expression (middle), and percentages of BT$^+$AG$^+$ cells (right) from each genotype are shown (4 biological replicates). Bar, 500 μm. **g**, Growth curves of BT$^+$AG$^+$ cells and enrichment scores during BMP-driven (-c12: 25 ng/ml; c12-: 100 ng/ml) wild-type and *TET1* KO#1 and #2 hPGCLC differentiation. 5 and 2 biological replicates for c12–c32 and for c42, respectively. The color coding is as indicated. **h**, UHC of the transcriptomes during hPGCLC induction and BMP-driven hPGCLC differentiation from wild-type and *TET1* KO hiPSCs, with the expression levels of key genes indicated. The color coding is as indicated. **i**–**k**, The numbers of the differentially expressed genes (DEGs) [log$_2$(RPM + 1) ≥ 3, fold change ≥ 2] between wild-type and *TET1* KO cells (up- or down-regulated in *TET1* KO cells) (**i**), UHC of the DEGs (**j**), and the GO enrichments and representative genes in the indicated DEG clusters (**k**). DEGs were defined using average expression values of biological replicates. The DEG numbers were unions of two comparisons (i.e., wild-type vs KO#1 and vs KO#2). Core ER genes were highlighted in red in (**j**). **l**, Box plots for the expression dynamics of ER genes (n = 42) during hPGCLC induction and BMP-driven hPGCLC differentiation from wild-type and *TET1* KO hiPSCs. *p*-values of Two-sided Dunnet's test (except c42) or paired two-sided t-test (c42) were shown. The upper hinges, lower hinges, and middle lines indicate 75 percentiles, 25 percentiles, and median values, respectively. The whiskers are drawn in length equal to the inter-quartile range (IQR) multiplied by 1.5. Data beyond the upper/lower whiskers are not shown.

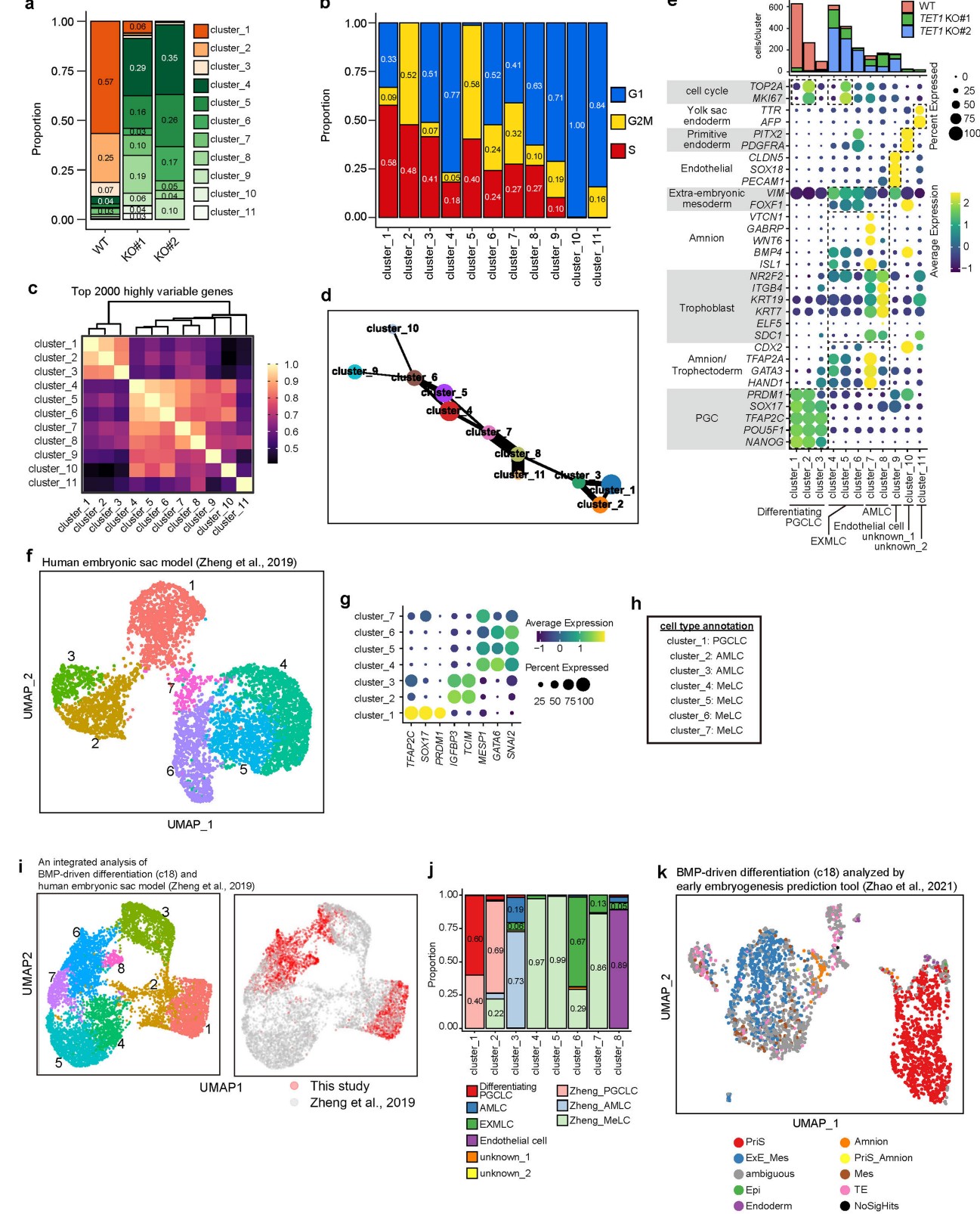

**Extended Data Fig. 11 |** See next page for caption.

**Extended Data Fig. 11 | TET1 protects hPGCLCs from differentiation into extraembryonic cells. a**, **b**, Proportion of wild-type and *TET1 KO* cells (**a**) and cell-cycle phases (**b**) in the 11 clusters in Fig. 4b. The actual proportions of major clusters (**a**)/cell-cycle phases (**b**) are shown within the histogram. The full information is provided in Source Data Extended Data Fig. 11. The color coding is as indicated. **c**, Correlations among clusters in Fig. 4b based on expression of the top 2,000 highly variable genes (HVGs). Spearman's rank correlation coefficient was calculated for analysis. UHC of the clusters is indicated on the heatmap. **d**, Partition-based graph abstraction (PAGA) analysis[77] of the relationships of the clusters in Fig. 4b. **e**, Genotype composition (top) and expression profiles of key lineage markers and cell-type annotation (bottom) of each cluster. AMLC: amnion-like cells; EXMLC: extra-embryonic mesoderm-like cells. Color coding is as indicated. **f** – **h**, UMAP plots and Louvain clustering of scRNA-seq data of a PSC-based model of early human post-implantation development[34] (**f**), the expression of key lineage markers in the 7 clusters in (**f**) (**g**), and the annotation of the 7 clusters based on their gene expression (**h**). AMLC: amnion-like cells; MeLC: mesoderm-like cells. The color coding is as indicated. **i**, (left) Integrated UMAP plots and Louvain clustering of scRNA-seq data in Fig. 4b with those of (**f**). (right) Distributions of data in this study and the study of Zheng et al. **j**, Cell-type composition of each cluster. Annotations are based on Zheng et al. and the results in panel (**h**). The actual proportions of major cell types are shown within the histogram. The full information is provided in Source Data Fig. 5. **k**, Prediction of the cell types of the clusters in Fig. 4b using the prediction tool by Zhao et al. [35] The color coding is as indicated. PriS: primitive streak; ExE_Mes: extra-embryonic mesoderm; Epi: epiblast; PriS_Amnion; primitive streak_amnion; Mes: mesoderm; TE: trophectoderm; NoSigHts: no significant hits.

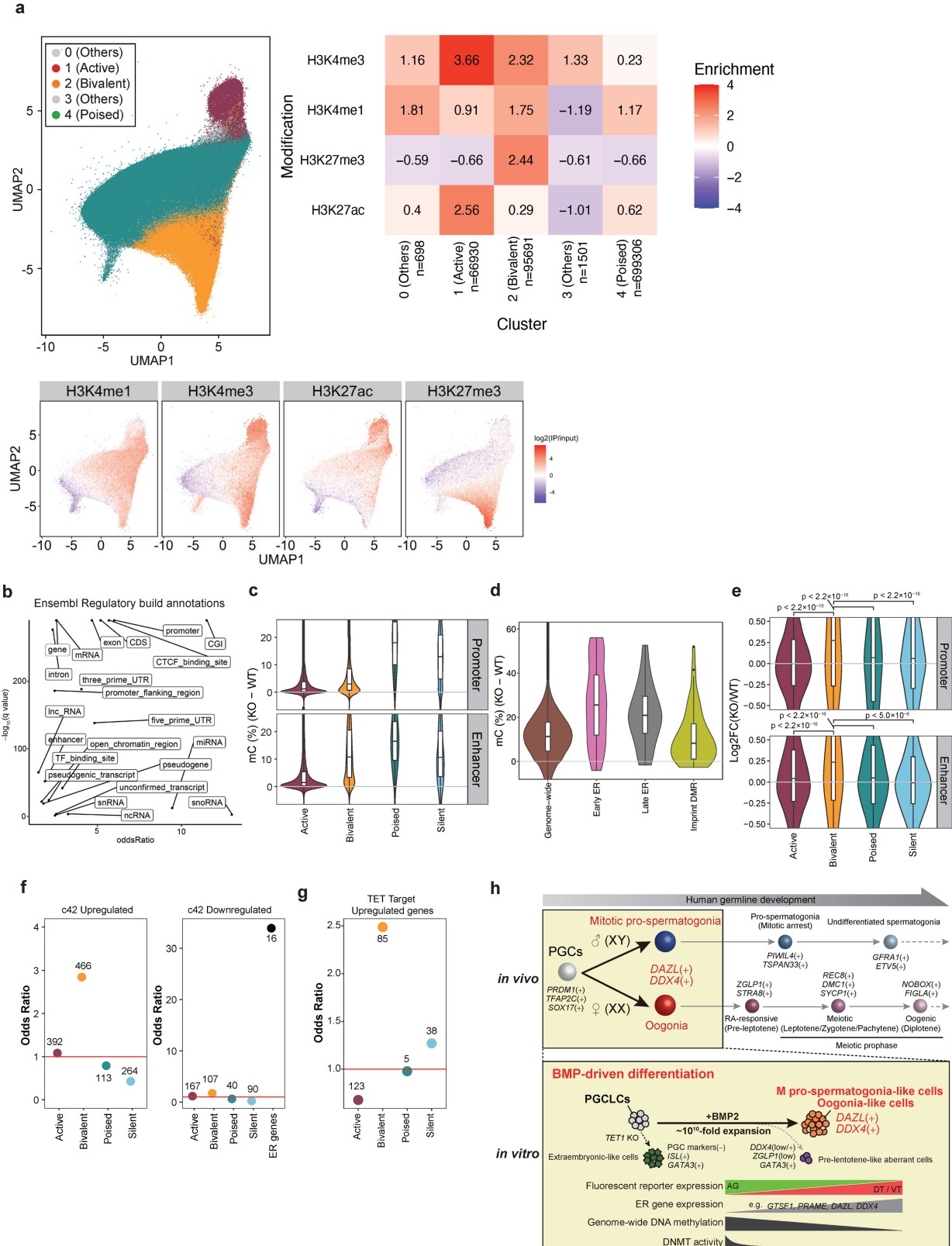

**Extended Data Fig. 12** | See next page for caption.

Extended Data Fig. 12 | *TET1* KO cells hyper-methylate regulatory elements and de-repress bivalent genes. a, Two dimensional UMAP embedding of all open sites (ATAC-seq peaks) during hPGCLC induction based on epigenetic signals of relevant cell types using public data[36], with labels derived from semi-supervised HDBSCAN (hierarchical density-based spatial clustering of applications with noise). The open sites were colored according to the labels (top, left) or signal intensities of relevant histone modifications (bottom). The averaged signal intensities of relevant histone modifications in each label (cluster) are shown (top, left). b, Odds ratio and *q*-value of the enrichment of the 2-kb bins with higher 5mC levels in *TET1* KO cells compared to wild-type cells at c42 in the Ensembl Regulatory Build annotations. c–e, Violin plots for the 5mC-level (%) (c, d) and the expression-level (log$_2$ fold-change) (e) differences between wild-type and *TET1* KO hPGCLC-derived cells at c42 on the indicated elements. Annotations and the numbers of each element are same as Fig. 5d,e,g. In (e), *p*-values of each comparison are as follows: <2.2 × 10$^{-16}$ for active promoter, poised promoter, silent promoter, active enhancer, and poised enhancer, *p* = 5.0 × 10$^{-6}$ for silent enhancer (two-sided *t*-test adjusted by Bonferroni correction). Promoters, enhancers (non-promoter open sites), and their labels are based on the data for d4 hPGCLCs[36]. Silent promoters: promoters that did not overlap with open sites; silent enhancers: enhancers categorized neither into active, bivalent, nor poised. In (c–e), the upper hinges, lower hinges, and middle lines indicate 75 percentiles, 25 percentiles, and median values, respectively. The whiskers are drawn in length equal to the inter-quartile range (IQR) multiplied by 1.5. Data beyond the upper/lower whiskers are shown as dots. f, Odds ratio of the enrichment of genes with indicated promoters defined in d4 hPGCLCs[36] and ER genes (for down-regulated genes) in genes up- (left) or down- (right) regulated in *TET1* KO hPGCLC-derived cells at c42. Number of each gene class is indicated. g, Odds ratio of the c42 up-regulated genes bound by TET in hESCs[39] in each category of promoters. The odds ratio was calculated relative to the background ratio of all genes bound by TET in each respective promoter category. Number of each gene class is indicated. h, A summery scheme of the present work.

# Reporting Summary

## Statistics

For all statistical analyses, confirm that the following items are present in the figure legend, table legend, main text, or Methods section.

| n/a | Confirmed | |
|---|---|---|
| ☐ | ☒ | The exact sample size (*n*) for each experimental group/condition, given as a discrete number and unit of measurement |
| ☐ | ☒ | A statement on whether measurements were taken from distinct samples or whether the same sample was measured repeatedly |
| ☐ | ☒ | The statistical test(s) used AND whether they are one- or two-sided<br>*Only common tests should be described solely by name; describe more complex techniques in the Methods section.* |
| ☒ | ☐ | A description of all covariates tested |
| ☐ | ☒ | A description of any assumptions or corrections, such as tests of normality and adjustment for multiple comparisons |
| ☐ | ☒ | A full description of the statistical parameters including central tendency (e.g. means) or other basic estimates (e.g. regression coefficient) AND variation (e.g. standard deviation) or associated estimates of uncertainty (e.g. confidence intervals) |
| ☐ | ☒ | For null hypothesis testing, the test statistic (e.g. *F*, *t*, *r*) with confidence intervals, effect sizes, degrees of freedom and *P* value noted<br>*Give P values as exact values whenever suitable.* |
| ☒ | ☐ | For Bayesian analysis, information on the choice of priors and Markov chain Monte Carlo settings |
| ☒ | ☐ | For hierarchical and complex designs, identification of the appropriate level for tests and full reporting of outcomes |
| ☐ | ☒ | Estimates of effect sizes (e.g. Cohen's *d*, Pearson's *r*), indicating how they were calculated |

*Our web collection on statistics for biologists contains articles on many of the points above.*

## Software and code

Policy information about availability of computer code

| Data collection | CFX384 Touch Real-Time PCR detection system (Bio-Rad Laboratories)<br>PromethION Flow Cells R9.4.1 (Oxford Nanopore Technologies)<br>PromethION 24 sequencer (Oxford Nanopore Technologies)<br>Novaseq 6000 sequencer (Illumina)<br>Nextseq 500/550 sequencer (Illumina)<br>CKX41 inverted microscope (Olympus)<br>DS-Fi2 microscopic camera (Nikon)<br>M205C microscope (Leica)<br>DP72 microscopic camera (Olympus)<br>FV1000-IX81 confocal microscope system (Olympus)<br>BZX810 (Keyence)<br>Orbitrap Fusion Lumos (Thermo Fisher Scientific)<br>Ultimate 3000 pump (Thermo Fisher Scientific)<br>HTC-PAL autosampler (CTC analytics)<br> Fusion solo S (VILBER LOURMAT) |
|---|---|
| Data analysis | FV10-ASW v4.1 (Olympus)<br>FACSDiva v8.0.2 (BD)<br>LiftOver program (https://genome.ucsc.edu/cgi-bin/hgLiftOver)<br>Fiji (ImageJ v1.52i)<br>CaptAdvance v16.13b (VILBER LOURMAT) |

FragPipe v20.0 (MSFragger v3.8, Philosopher v5.0.0, IonQuant v1.9.8)
Excel 2010 (Microsoft)
R version v4.0.3
DAVID 2021 (https://davidbioinformatics.nih.gov)
RECODE v1.1.1
Cellranger v6.0.1
cutadapt v1.18 / v1.9.1
Tophat v2.1.1
Bowtie2 v2.3.4.1
HTSeq v0.9.1
cuffinks v.2.2.1
Trim galore! v0.4.1 / v0.6.3
Bismark v0.22.1
Samtools v1.15.1 / v1.7
Methpipe v3.4.3
bedtools v2.29.2
Homer v4.11.1
deepTools v3.5.0
PicardTools v2.18.23
MACS2 v2.1.1
Megalodon v2.5.0
Clair3 v0.1.12
bcftools v1.15.1
Whatshap v1.4
modbam2bed v0.5.3
SNPsplit v0.3.2
scVelo v0.2.5
scanpy v1.9.1
Scrublet v0.2.3
screcode v0.1.2
DiffBind v3.8.4
cuML v23.02.00
Seurat v4.2.1 / v4.1.1
stats v3.6.1
ggplot2 v3.4.1
rstatix v0.7.2
SimComp v3.3
ggsignifr v0.6.0
ChIPpeakAnno v3.24.2

For manuscripts utilizing custom algorithms or software that are central to the research but not yet described in published literature, software must be made available to editors and reviewers. We strongly encourage code deposition in a community repository (e.g. GitHub). See the Nature Portfolio guidelines for submitting code & software for further information.

# Data

Policy information about availability of data

All manuscripts must include a data availability statement. This statement should provide the following information, where applicable:

- Accession codes, unique identifiers, or web links for publicly available datasets
- A description of any restrictions on data availability
- For clinical datasets or third party data, please ensure that the statement adheres to our policy

Human genome reference GRCh38.p12(https://www.ncbi.nlm.nih.gov/datasets/genome/GCF_000001405.38/) and GRCh38.p2(https://www.ncbi.nlm.nih.gov/datasets/genome/GCF_000001405.28/) are used in this study. The accession number for the sequencing data (bulk RNA-seq, scRNA-seq, EM-seq) generated in this study is GSE232078 (the GEO database). The R script is available on request. ONT long read sequence data have been deposited to NBDC Human Database with accession number JGAS000690. The raw MS data and analysis files have been deposited in the ProteomeXchange Consortium (http://proteomecentral.proteomexchange.org) via the jPOST partner repository (https://jpostdb.org) and can be accessed using the dataset identifier PXD048118. Public data used in this study were listed below.

Processed data of human embryonic gut 10X scRNA-seq is from https://cellxgene.cziscience.com/collections/17481d16-ee44-49e5-bcf0-28c0780d8c4a.
Human gastrula Smart-Seq2 scRNAseq data is from E-MTAB-9388 (https://www.ebi.ac.uk/biostudies/arrayexpress/studies/E-MTAB-9388).
Human gonadal ovary 10X scRNA-seq data (wk11) is from GSE194266 (https://www.ncbi.nlm.nih.gov/geo/query/acc.cgi?acc=GSE194266).
Human gonadal ovary 10X scRNA-seq data (wks 7/9/10/13/16) is from GSE143380 (https://www.ncbi.nlm.nih.gov/geo/query/acc.cgi?acc=GSE143380).
Human embryonic sac model is from GSE134571 (https://www.ncbi.nlm.nih.gov/geo/query/acc.cgi?acc=GSE134571).
Human ESCs, pre-mesoderm cells, mesoderm cells, definitive endoderm cells, human PGCLCs, and human PGCs ChIP-seq data is from GSE159654 (https://www.ncbi.nlm.nih.gov/geo/query/acc.cgi?acc=GSE159654).
hPGCLC induction and hPGCLC culture methylome data are from GSE86586 (https://www.ncbi.nlm.nih.gov/geo/query/acc.cgi?acc=GSE86586) and GSE174485 (https://www.ncbi.nlm.nih.gov/geo/query/acc.cgi?acc=GSE174485), respectively.

# Research involving human participants, their data, or biological material

Policy information about studies with human participants or human data. See also policy information about sex, gender (identity/presentation), and sexual orientation and race, ethnicity and racism.

| | |
|---|---|
| Reporting on sex and gender | NA |
| Reporting on race, ethnicity, or other socially relevant groupings | NA |
| Population characteristics | NA |
| Recruitment | NA |
| Ethics oversight | NA |

Note that full information on the approval of the study protocol must also be provided in the manuscript.

# Field-specific reporting

Please select the one below that is the best fit for your research. If you are not sure, read the appropriate sections before making your selection.

☒ Life sciences  ☐ Behavioural & social sciences  ☐ Ecological, evolutionary & environmental sciences

For a reference copy of the document with all sections, see nature.com/documents/nr-reporting-summary-flat.pdf

# Life sciences study design

All studies must disclose on these points even when the disclosure is negative.

| | |
|---|---|
| Sample size | Sample size was not predetermined. The number of cells used for a statistical test was enough when comparing to previous studies reporting quantification of immunofluorescence signals(Ohta et al., 2017; Nagano et al., 2022). The number of genes used for statistical tests was determined by a biological criteria. |
| Data exclusions | For bulk RNA-seq, a sample derived from cells collected by inadequate FACS sorting gate was excluded retrospectively. For scRNA-seq by 10x chromium, low quality cells were excluded based on criteria described Methods section. |
| Replication | The exact number of replicates were indicated in the main text, figure legend or Methods section. |
| Randomization | For hPGCLC expansion culture experiments, the initial population of the expansion culture was randomly assigned by FACS sorting from gated cells. For immunofluorescence quantification analysis, images used for analysis were taken from randomly chosen locations. |
| Blinding | All experiments were not blinded. Investigators who performed experiments planned the sample allocation and culture conditions. |

# Reporting for specific materials, systems and methods

We require information from authors about some types of materials, experimental systems and methods used in many studies. Here, indicate whether each material, system or method listed is relevant to your study. If you are not sure if a list item applies to your research, read the appropriate section before selecting a response.

### Materials & experimental systems

| n/a | Involved in the study |
|---|---|
| ☐ | ☒ Antibodies |
| ☐ | ☒ Eukaryotic cell lines |
| ☒ | ☐ Palaeontology and archaeology |
| ☒ | ☐ Animals and other organisms |
| ☒ | ☐ Clinical data |
| ☒ | ☐ Dual use research of concern |
| ☒ | ☐ Plants |

### Methods

| n/a | Involved in the study |
|---|---|
| ☒ | ☐ ChIP-seq |
| ☐ | ☒ Flow cytometry |
| ☒ | ☐ MRI-based neuroimaging |

# Antibodies

| | |
|---|---|
| Antibodies used | rat anti-GFP, monoclonal Nacalai Tesque 04404-84 |

| Antibodies used | mouse anti-UHRF1, monoclonal Millipore MABE308 |
| --- | --- |
| | goat anti-tdTomato, polyclonal Origene AB8181-200 |
| | mouse anti-AP2γ/TFAP2C, Santa Cruz sc-12762 |
| | mouse anti-DAZL, monoclonal Santa Cruz sc-390929 |
| | rabbit anti-DDX4 antibody, abcam ab13840 |
| | rabbit anti-5hmC antibody  39069 Active Motif |
| | rabbit anti-5mC antibody  SIGMA SAB5600040 |
| | rabbit anti p44/42 MAPK(ERK1/2) monoclonal CST 4695 |
| | rabbit anti phospho p44/42 MAPK(ERK1/2) monoclonal CST 4370 |
| | mouse anti alpha Tublin monoclonal SIGMA T9026 |
| | mouse anti-TRA-1-85 conjugated with Brilliant Violet 421, monoclonal BD Bioscience 563302 |
| | mouse anti EpCAM antibody conjugated with APC monoclonal  BioLegend, 324208 |
| | rat anti CD49f (ITGA6) antibody conjugated with APC monoclonal  BioLegend, 313624 |
| | rat anti CD49f (ITGA6) antibody conjugated with PE monoclonal  BioLegend, 313611 |
| | |
| | Alexa Fluor 488 donkey anti-rat IgG  Invitrogen A21208 |
| | Alexa Fluor 568 donkey anti-goat IgG  Invitrogen A11057 |
| | Alexa Fluor 647 donkey anti-mouse IgG Invitrogen A31571 |
| | anti-rabbit IgG goat IgG conjugated with HRP  SIGMA A6154 |
| | anti-mouse IgG sheep IgG conjugated with HRP  SIGMA A5906 |

| Validation | rat anti-GFP, monoclonal Nacalai Tesque 04404-84 |
| --- | --- |
| | Validation performed by the external collaborator was provided by the manufacturer (https://www.nacalai.co.jp/global/download/pdf/Epitope_Tag_Antibody.pdf). This antibody was used in previous studies(Yamashiro et al., 2018, Murase et al., 2020, Gyobu-Motani et al., 2023) |
| | |
| | mouse anti-UHRF1, monoclonal Millipore MABE308 |
| | Validation was performed by the manufacturer using HeLa cells (https://www.merckmillipore.com/JP/ja/product/Anti-ICBP90-UHRF1-Antibody-clone-1RC1C-10,MM_NF-MABE308). |
| | |
| | goat anti-tdTomato, polyclonal Origene AB8181-200 |
| | Validation was performed by the manufacturer using 293T cells expressing tdTomato (https://www.origene.com/catalog/antibodies/tag-antibodies/ab8181-200/tdtomato-goat-polyclonal-antibody). |
| | |
| | mouse anti-AP2γ/TFAP2C, Santa Cruz sc-12762 |
| | Validation was not described by the manufacturer.(https://www.scbt.com/ja/p/ap-2gamma-antibody-6e4-4) This antibody was used in previous studies (Murase et al., 2020, Mizuta et al., 2022) |
| | |
| | mouse anti-DAZL, monoclonal Santa Cruz sc-390929 |
| | Validation was not described by the manufacturer. This antibody was used in previous studies (Murase et al., 2020, Mizuta et al., 2022) (https://www.scbt.com/ja/p/dazl-antibody-e-6). |
| | |
| | rabbit anti-DDX4 antibody, abcam ab13840 |
| | Validation was not described by the manufacturer.(https://www.abcam.co.jp/products/primary-antibodies/ddx4--mvh-antibody-ab13840.html) This antibody was used in previous studies (Mizuta et al., 2022) |
| | |
| | rabbit anti-5hmC antibody, polyclonal  39069 Active Motif |
| | This antibody was validated for MeDIP and dot blot analysis the by manufacturer (https://www.activemotif.com/catalog/details/39769.html). |
| | |
| | rabbit anti-5mC antibody, monoclonal  SIGMA SAB5600040 |
| | Validation was not described by the manufacturer, but was recommended to be used for ELISA, dot blot, immunocytochemistry, or immunohistchemistry (https://www.sigmaaldrich.com/JP/ja/product/sigma/sab5600040). |
| | |
| | rabbit anti p44/42 MAPK(ERK1/2) monoclonal CST 4695 |
| | Validation was not described by the manufacturer, but can be applicable to western blotting (https://www.cellsignal.jp/products/primary-antibodies/p44-42-mapk-erk1-2-137f5-rabbit-mab/4695). |
| | |
| | rabbit anti phospho p44/42 MAPK(ERK1/2) monoclonal CST 4370 |
| | Validation was not described by the manufacturer, but can be applicable to western blotting (https://www.cellsignal.jp/products/primary-antibodies/phospho-p44-42-mapk-erk1-2-thr202-tyr204-d13-14-4e-xp-rabbit-mab/4370). |
| | |
| | mouse anti alpha Tublin monoclonal SIGMA T9026 |
| | Validation was not described by the manufacturer, but can be applicable to western blotting (https://www.sigmaaldrich.com/JP/en/product/sigma/t9026). |
| | |
| | mouse anti-TRA-1-85 conjugated with Brilliant Violet 421, monoclonal BD Bioscience 563302 |
| | Validation was not described by the manufacturer, but can be applicable to flow cytometry (https://www.bdbiosciences.com/ja-jp/products/reagents/flow-cytometry-reagents/research-reagents/single-color-antibodies-ruo/bv421-mouse-anti-human-tra-1-85-antigen.563302).This antibody was used in previous studies (Murase et al., 2020) |
| | |
| | mouse anti EpCAM antibody conjugated with APC monoclonal  BioLegend, 324208 |
| | Validation was not described by the manufacturer, but can be applicable to flow cytometry (https://www.biolegend.com/de-at/cell-health/apc-anti-human-cd326-epcam-antibody-3758). This antibody was used in previous studies (Murase et al., 2020) |
| | |
| | rat anti CD49f (ITGA6) antibody conjugated with APC monoclonal  BioLegend, 313624 |

Validation was not described by the manufacturer, but can be applicable to flow cytometry (https://www.biolegend.com/de-de/products/brilliant-violet-421-anti-human-mouse-cd49f-antibody-8644?GroupID=BLG10323). This antibody was used in previous studies (Murase et al., 2020)

rat anti CD49f (ITGA6) antibody conjugated with PE monoclonal  BioLegend, 313611
Validation was not described by the manufacturer, but can be applicable to flow cytometry (https://www.biolegend.com/fr-fr/products/pe-anti-human-mouse-cd49f-antibody-4108).

Alexa Fluor 488 donkey anti-rat IgG Invitrogen A21208
According to supplier's description, cross-reactivity to bovine, chicken, goat, guinea pig, hamster, horse, human, mouse, rabbit, and sheep serum proteins was reduced to the minimum level by affinity-based purification. Specificity of the antibody was increased by cross-adsorption. This antibody can be used for immunofluorescence (https://www.thermofisher.com/antibody/product/Donkey-anti-Rat-IgG-H-L-Highly-Cross-Adsorbed-Secondary-Antibody-Polyclonal/A-21208).

Alexa Fluor 568 donkey anti-goat IgG Invitrogen A11057
According to supplier's description, cross-reactivity to bovine, chicken, goat, guinea pig, hamster, horse, human, mouse, rabbit, and sheep serum proteins was reduced to the minimum level by affinity-based purification. Specificity of the antibody was increased by cross-adsorption. This antibody can be used for immunofluorescence (https://www.thermofisher.com/antibody/product/Donkey-anti-Goat-IgG-H-L-Cross-Adsorbed-Secondary-Antibody-Polyclonal/A-11057).

Alexa Fluor 647 donkey anti-mouse IgG Invitrogen A31571
According to supplier's description, cross-reactivity to bovine, chicken, goat, guinea pig, hamster, horse, human, mouse, rabbit, and sheep serum proteins was reduced to the minimum level by affinity-based purification. Specificity of the antibody was increased by cross-adsorption. This antibody can be used for immunofluorescence (https://www.thermofisher.com/antibody/product/Donkey-anti-Mouse-IgG-H-L-Highly-Cross-Adsorbed-Secondary-Antibody-Polyclonal/A-31571).

anti-rabbit IgG goat IgG conjugated with HRP  SIGMA A6154
According to supplier's description, this is affinity-isolated antibody. This antibody can be used for western blot (https://www.sigmaaldrich.com/JP/en/product/sigma/a6154).

anti-mouse IgG sheep IgG conjugated with HRP  SIGMA A5906
According to supplier's description, this is affinity-isolated antibody. This antibody can be used for western blot (https://www.sigmaaldrich.com/JP/en/product/sigma/a5906).

# Eukaryotic cell lines

Policy information about cell lines and Sex and Gender in Research

| | |
|---|---|
| Cell line source(s) | iPSC cell lines 585B1, 1390G3 and 1383D6 was provided by Dr. Keisuke Okita (CiRA, Kyoto, Japan). iPSC cell line NCLCN was purchased from XCell Science and licensed for academic use. M1-BTAG(585B1-BTAG) and F2-AGVT(1390G3-AGVT) lines were generated in previous studies (Sasaki et al., 2015, Yamashiro et al., 2018). M1-AGDT/AGVT(585B1-AGDT/AGVT) and F1-AGVT(NCLCN-AGVT) lines were generated in this study. |
| Authentication | iPSC cells were maintained in AK02N or AK03N which was developed for maintaining iPSC cell lines. iPSCs were authenticated by a typical morphology of human primed pluripotent stem cells. For TET1 KO iPSC cell line, each clone was authenticated by the specific indel pattern in the exon 6 as follows: 14 bp ins/1 bp del for KO#1, 28 bp del/20 bp del  for KO#2. |
| Mycoplasma contamination | Cell lines used in this study were tested and confirmed to be negative for mycoplasma contamination . |
| Commonly misidentified lines (See ICLAC register) | No commonly misidentified lines were used in this study. |

# Flow Cytometry

## Plots

Confirm that:

☒ The axis labels state the marker and fluorochrome used (e.g. CD4-FITC).

☒ The axis scales are clearly visible. Include numbers along axes only for bottom left plot of group (a 'group' is an analysis of identical markers).

☒ All plots are contour plots with outliers or pseudocolor plots.

☒ A numerical value for number of cells or percentage (with statistics) is provided.

## Methodology

| | |
|---|---|
| Sample preparation | Expanded hPGCLCs were treated with a 1:4 mixture of 0.5% trypsin–EDTA and PBS (−) at 37°C for 5 min. Alternatively, Accumax can be used instead. After the incubation, trypsin treatment was neutralized by adding advanced RPMI containing 10 % FBS, 0.1 mg/mL DNase1 and 10 uM Y27632. Alternatively, the expansion culture medium can be used instead. Cell suspension was directly subjected after filtration by a strainer. DAPI was added to the cell suspension if needed. |

| | |
|---|---|
| Instrument | FACS Aria 3 |
| Software | FACS Diva |
| Cell population abundance | Cell sorting by FACS Aria 3 was performed in "purity mode" to ensure the purity of sorted cell population. |
| Gating strategy | 585B1-AGDT, NCLCN-AGVT<br>First, cell debris was excluded by gating on FSC/SSC plot. Second, single cell population was gated on FSC-height/FSC-width plot and SSC-height/SSC-width plot. Third, if needed, DAPI(+) dead cells were excluded by gating on DAPI/FSC plot. Resultant single live cells were used for the reporter expression analysis.<br><br>585B1-BTAG, TET1KO#1, TET1KO#2<br>First, cell debris was excluded by gating on FSC/SSC plot. Second, single cell population was gated on FSC-height/FSC-width plot and SSC-height/SSC-width plot. Resultant single live cells were used for the reporter expression analysis. BT(+) cells and AG(+) cells were gated independently on BT/APC or AG/APC plot, respectively.<br><br>585B1-BTAG (TRA-1-85 expression analysis)<br>First, cell debris was excluded by gating on FSC/SSC plot. Second, single cell population was gated on FSC-height/FSC-width plot and SSC-height/SSC-width plot. Resultant single live cells were used for TRA-1-85 expression analysis. BT(+)AG(+) cells were gated on BT/AG plot. Then, non-BTAG cells were plotted on FSC/SSC plot.<br><br>1383D6 (EpCAM/ITGA6 expression analysis)<br>First, cell debris was excluded by gating on FSC/SSC plot. Second, single cell population was gated on FSC-height/FSC-width plot and SSC-height/SSC-width plot. Resultant single live cells were used for EpCAM/ITGA6 expression analysis. |

☒ Tick this box to confirm that a figure exemplifying the gating strategy is provided in the Supplementary Information.

