## [Peer Review File · Nature]

Manuscript Title: In vitro reconstitution of epigenetic reprogramming in the human germ line

Reviewer Comments & Author Rebuttals

Reviewer Reports on the Initial Version:

Referees' comments:

Referee #1 (Remarks to the Author):

The results presented by Murase et al. are interesting and add information about the process by which human primordial germ cells undergo their fetal differentiation and – crucially - global DNA de-methylation. The authors identify a heretofore unknown role for BMP signalling in proliferation and epigenetic re-programming of PGCLCs (adding to its already known roles in PGC specification and oogenesis). The authors also conclude that human PGC de-methylation occurs largely through a passive, replication-dependent process.

Generally, the work has been performed in a very extensive and careful manner. Nonetheless, this study suffers from a limitation. The main work is performed in cells derived from two iPSC lines, which differ from each other in quite substantial ways. This makes conclusions regarding robustness challenging. The two lines differ in chromosome complement (XX vs. XY), reporter gene complement (AGDT vs. AGVT) and culture requirements (differing levels of BMP2 used per cell line, presumably related to the cellular state of the initial iPSC lines). The work examining TET1 function in PGCs was performed in two sub-cultured lines but uses a third reporter complement (BTAG) further complicating analysis. It is strongly recommended to reproduce the key results (kinetics of reporter gene activation and DNA de-methylation timing) in at least one additional cell line of each sex chromosome complement (with matching reporter gene constellations) to determine if the differences in dynamics relate to XX vs. XY cell lines or are more strongly related to the variation in starting states from different iPSCs.

Other points:

An important point relates to nomenclature. The authors refer to the BMP2 treated cells as being “oogonia-like and mitotic pro-spermatogonia-like,” however given no evidence for sexually dimorphic gene expression in these cells, would not the sex neutral term “gonocyte” be a more appropriate label?

Figure 1: Panel H: what fraction of cells become DAZL positive?

Explain to the reader why you generated reporter strains for DAZL and DDX4/VASA expression.

Figure 2:

Panel A: The mis-match in timepoints between the two cell lines makes this figure hard to read (some timepoints line up, while others do not). The differing culture conditions also make this

Panel B: Why not show the two cell lines on the same graph? It is hard to compare the dynamics as is.

Panel C: Legend for second plot: should the grey bar be labelled VT?

Panel G: At c82 the results for DAZL-Tomato selection seem strange. For DAZL itself the result is clear (AG selected cells express substantially less DAZL mRNA than AGDT cells). However, the results for DDX4 do not agree (AG selected cells express nearly identical levels of DDX4 mRNA as AGDT cells). Given that the authors argue that DDX4 is a later marker of differentiation than DAZL this seems counterintuitive.

Figure 3:

Panel A:

Reporter vs. marker expression:

VT cells are selected to be AG(-), if so, why do they show equivalent levels of TFAP2C mRNA?

Why do certain AG cells show similar levels of DDX4 mRNA as AGVT do? Is the DDX4 protein under post-transcriptional regulation in these cells? Immunofluorescence with DDX4 antibody to examine the level and frequency of expression of endogenous protein in later cultures of AG and AGVT cells would help to clarify this point.

Expression dynamics:

In RNA-Seq it appears that selection for DAZL also selects for DDX4 (low level of DAZL mRNA appears in the Tomato-negative population). Subsequent culture of AGDT cells leads to expression of SYCP3. Selection for DDX4 expression (via AGVT) also selects for high DAZL mRNA levels (though at later timepoints AG cells also express DAZL mRNA). In the AGVT harboring cell line, all DAZL expressing cells also express SYCP3. Is the repression of SCP3 transcription less potent in female cells? What about DNA methylation of its promoter?

Panel B: The relationship between experiments in this PCA is unclear. The labelling from 1390G3-AGVT (from previous experiments cultured with xrOvaries?) is not at all clear from the legend. Also, which cells are selected for which reporters? A more direct comparison between BMP-driven hPGCLC differentiation and differentiation in xrOvaries is needed.

Panel C: Have the authors observed any batch effects between in vivo and in vitro data sets? If yes, how have they been resolved?

Panel I: How comparable are the enrichments and statistical significances of the different GO terms identified under in vitro versus in vivo conditions?

Figure 4:

Panel A: Separating promoters into CGI-rich and CpG-poor classes would be useful to compare the dynamics.

Panel A: The authors identified two ways to reprogram the hPGCLC DNA methylome in vitro, via BMP-supported growth versus induction in xenotransplanted ovaries. It is important to perform a careful genome wide comparison between the temporal kinetics and ultimate methylome states between the two conditions. It is particularly interesting to address/discuss conditions of passive, replication-coupled loss of DNA methylation. How relevant do the authors consider the differences in UHRF1 RNA expression under the two conditions? Are there differences in the degree of residual DNA methylation residing at promoters of key meiotic genes?

Panel E: Escapers from DNA de-methylation in both BMP2 treated hPGCLC lines show an increase in Satellite sequences (though not in xrOvary cultured cells). How extensive is the extent of this escape within the genome and how do these elements behave if plotted as in Figure 4A?

Figure 5:

Panel C: What does the expression of ER genes look like in TET1 KO cells? Given the lack of a differentiation reporter in these cells it is not possible to isolate them, but expression analysis in the BTAG population would be interesting.

Figure 6

Panel D: The authors do not provide clear definitions of how “poised” vs. “bivalent” states are defined in their analysis.

Panel F: How are enhancers assigned to specific genes for expression analysis?

Panel F: The effect of TET1-deficiency on gene expression seems small in magnitude. Is this driven by variation in how many cells induce expression vs. low levels of increase in all cells. The authors should examine the expression levels of genes associated with these promoters/enhancers in their single-cell RNA-Seq analysis to address this point.

The result section of the manuscript is written in a very dense manner and is from time to time hard to follow, particularly also because of the use of many acronyms. In contrast, the discussion is written in a clear and understandable manner. The authors are recommended to improve the clarity of the results section.

Referee #2 (Remarks to the Author):

The manuscript by Murase et al. addresses a pertinent issue regarding the use of human primordial germ cell models, specifically the scarcity and quality of epigenetic reprogramming. Building upon their prior experiments, the authors have identified conditions that enable robust expansion of both male and female human primordial germ cell-like cells (hPGCLCs) in a simple 2D culture system through the addition of BMP ligands. Importantly, this approach not only maintains the expression of key germline transcription factors but also induces a robust DNA demethylation process similar to that observed when hPGCLCs are re-aggregated with ovary/testes soma.

Utilizing this model, along with RNAseq, scRNAseq, and epigenomic profiling, as well as genetic manipulations, the authors present several novel findings. They propose that the addition of BMP after the specification stage induces germline epigenetic reprogramming, attenuates the MAPK/ERK pathway, and consequently represses de novo and maintenance DNA methylation machinery. Furthermore, they suggest that DNA demethylation occurs through a passive replication-dependent process, while highlighting the role of TET1 in safeguarding hPGCLCs against differentiation towards amnion-like cells.

In general, this is a well-executed study that establishes a model for expanding hPGCLCs up to the mitotic spermatogonia/oogonia stage. While the technical advancements made are somewhat incremental, as they build upon previously published protocols, the study's practical utility will be of broad interest. In my opinion, the mechanistic aspects of the study are of high quality; however, there are some key validation experiments that are missing. The most significant concern with the paper lies in the interpretation of the TET1-KO phenotype, as the experiments conducted are insufficient to assess the role of hydroxymethylation in global DNA demethylation in PGCLCs. Instead, the authors should focus on the amnion/TE differentiation phenotype, which is interesting

in its own right.

Overall, with some textual changes and the inclusion of a few additional experiments, I believe this paper has the potential to be published in high-impact journals such as Nature Cell Biology or even Nature. Detailed criticism is provided below

Major criticism:

1. The authors claim that the addition of BMP2 allows for PGCLC expansion with minimal sorting purification, resulting in MAPK/ERK modulation and repression of DNA methylation machinery. However, several controls are missing. Firstly, it is necessary to demonstrate the long-term effects of BMP2 treatment in both male and female cells compared to a control group without BMP2. Currently, these controls are only included for the AGDT line and not the AGVT line. Additionally, it is unclear why different markers are used for female and male cells, despite both *Dazl* and *Vasa* being expressed in both sexes. Moreover, the MAPK/ERK effect of BMP stimulation is solely shown through RNAseq, which is insufficient to support the claim made in the abstract. To support this claim, it is necessary to perform pERK staining to conclusively demonstrate BMP-dependent ERK regulation.
2. Figure 3 presents single-cell RNAseq analysis integrated with in vivo samples. The analysis clearly indicates that only the EM1/M1/2/3/PLL1 clusters are shared between in vivo and in vitro samples, with clear off-target populations in the PLL2 cluster. While this is somewhat discussed in the text, it should be more explicitly stated in the results section rather than only in the discussion. Specifically, lines 299-303 should be revised to provide a more nuanced and clear explanation.
3. The epigenomic analysis shows that DNA demethylation in vitro is significantly delayed compared to the dynamics observed in vivo. Although this is mentioned in the discussion, it is not adequately addressed in the results section. Additionally, male cells appear to aberrantly demethylate their imprints (*PEG3*; *IGF2R*), which is linked to hypermethylation in the iPSC state. This observation should be mentioned, and the claim made in lines 344-345 should be nuanced accordingly.
4. The authors demonstrate poor X chromosome reactivation (XCR) in their system, as observed in other in vitro models. While this limitation is appropriately discussed, the authors suggest that it is likely due to the aberrant epigenetic status of iPSCs (lines 367-369; 423). However, the cited papers fail to provide functional evidence supporting this hypothesis. While this speculative hypothesis can be mentioned in the discussion, it should be removed from the results section since the authors do not investigate other marks besides 5mC. Additionally, the claim that XCR depends on promoter demethylation on Xi (line 421) is not accurate. While there is a correlation between demethylation and reactivation, the functional link has not been established yet, as it could be related to the delayed erasure of other marks, such as histone modifications.
5. One of the more intriguing claims is that the majority of 5mC is passively lost during PGCLC differentiation. Although the results broadly support this statement, some nuances should be added. Firstly, the depletion of *UHRF1* and *DNMT3B* in vitro is less pronounced compared to in vivo conditions (Figure 3a). Secondly, there is no conclusive experiment demonstrating that active demethylation does not drive the demethylation of ER or imprinted genes, as suggested by Hackett et al. in mice in 2013. The TET1KO experiments are inconclusive regarding the dynamics of DNA demethylation, as there is off-target differentiation and limited conclusions can be drawn. Furthermore, there appears to be delayed activation of early ER genes (*GTSF1*, *SPATA22*, *PRAME*) in the KO group as early as c12, which may indicate TET1 involvement at CGI and/or imprints. Therefore, the conclusions in this area should be more nuanced.

6. The authors make an exciting discovery that TET1KO PGCLCs de-differentiate into TE/Amnion-like fates. However, they do not integrate their datasets with in vivo counterparts. It is suggested to utilize the recent pre-print (10.1101/2021.05.07.442980), which includes a data integration tool for examining TE and amnion fates. Another crucial point is that TET1KO PGCLCs already appear significantly different from the controls at c0 and c12, as observed in Figure 5C, with upregulation of genes like HAND1. This raises the question of whether TET1 is involved in PGC specification and if this phenotype becomes stronger with passaging of PGCLCs. Additionally, it seems that the regulation of bivalent genes occurs during specification, not only during expansion. Thus, statements like those in lines 620-623 or 734-736 should be revisited. Similarly, the statement in lines 633-635 is speculative at this point. To challenge the function of TET1 in epigenetic reprogramming, it would be necessary to conduct conditional loss-of-function experiments during the expansion stage. Overall, the final conclusions of this paper should be softened, as it is premature to state that there is no contribution of active DNA demethylation, especially at imprints and germline genes.

Minor comments:

1. Extended Figure 1g: The color scale is not visible.
2. Some statistical analyses and quantifications are missing, such as in Fig1b, Fig1h, Fig6f/h.
3. In line 175-177, it should be clarified that at this point, the claim can only be made that BMP is involved in PGCLC expansion/differentiation, and not yet in epigenetic reprogramming. The evidence for epigenetic reprogramming will be presented later.
4. In Figure 2c (bottom), the color code is incorrect.
5. In Fig2b, it is unclear which gates were used for the purification of the culture. Did you sort for the AG+ve cells or double positive cells?
6. In line 242, there seems to be a mistake in the cells mentioned. Shouldn't it be AG+DT/VT+ cells?
7. In Extended Figure 5a, there appears to be a gene wrongly annotated as an early core ER gene, as it is highly expressed in female iPSCs.
8. In Extended Figure 5e, the expression of Xist from Xi is not visible. Why is this?
9. In Extended Figure 5h, the labeling of the y-axis is confusing. What is the scale on the right? The legend does not explain this graph clearly.
10. In Extended Figure 6f, there are significant mRNA levels of TET1 remaining. Therefore, an additional western blot should be performed to demonstrate the loss of the protein using an N-terminus specific antibody. If some protein is still present, it is possible that the authors are observing partial penetrance or a dominant negative effect. This should be discussed.
11. The statement in line 673 should be rephrased. PGCLC expansion under standard conditions results in the epigenetic reprogramming of H3K9 and H3K27 methylation, with only DNA methylation showing inefficient reprogramm

Referee #3 (Remarks to the Author):

Mammalian primordial germ cells (PGCs) arise during gastrulation, and then migrate to the gonads, whereupon they turn on hallmark markers DAZL and DDX4/VASA; later they undergo sex determination and subsequent steps of germ cell differentiation (PMID 31754036, DOI 10.1007/978-981-10-7941-2_1). Current human pluripotent stem cell (hPSC) differentiation methods yield DAZL-

DDX4- PGC-like cells (PGCLCs) that likely correspond to early PGCs prior to their arrival at the gonads (e.g., PMID 26189426, 25543152, 28607482). Consequently, the next major challenge for the field is to understand how to convert them into DAZL+ DDX4+ gonadal-stage PGCs, and the extracellular signals that drive this pivotal developmental transition.

The authors previously showed that hPSC-derived PGCLCs cocultured with ovarian cells turned on DDX4 at extremely low efficiency (less than 2%, after 120 days of culture), within “reconstituted” ovarian cultures (PMID 30237246), thus raising the question of whether it would be possible to produce gonadal-stage PGCLCs without ovarian support cells. Alternatively, the authors previously developed a system to numerically expand hPSC-derived early-stage PGCLCs, but in this system, PGCLCs did not seem to spontaneously mature to a later, gonadal-like state (PMID 32954504).

Here the authors report three important advances. First, starting from hPSC-derived early-stage PGCLCs, they further differentiate them into enriched populations of DAZL+ and DDX4+ PGCLCs that likely correspond to gonadal-stage PGCLCs (Fig. 2). Second, during this differentiation process, the authors achieve massive expansion in cell numbers (>10 billion-fold), thus enabling the mass production of gonad-stage PGCLCs for basic research and other practical applications. Third, during this differentiation process, the hPSC-derived PGCLCs not only turn on later stage markers (e.g., DAZL and DDX4), but they achieve impressive genome-wide DNA demethylation, which represents a key component of the epigenetic reprogramming that occurs in the developing germline. However, there are still two downsides of the authors’ system: it still relies on (1) mouse M220 feeder cells and (2) different dosages of BMP across different hPSC lines.

Altogether, the paper is extremely thorough. The authors have not yet arrived at the “holy grail” of efficiently reconstituting later-stage germ cell differentiation (e.g., sex determination and meiosis). Nevertheless, they have developed an important method to produce gonadal-stage PGCLCs in massive numbers, and the entire field will undoubtedly seize upon this method to eventually generate later-stage germ cells.

Major comments: the paper is technically comprehensive and in this reviewer’s opinion, there are no overt experiments to suggest. This reviewer’s thoughts were more focused on the claims and presentation of the manuscript.

1. Main message of the manuscript. This paper reported an impressive advance, but this reviewer thought that the significance could be more clearly conveyed to a broader audience. In the Abstract and Introduction, the authors talk about how they recapitulated “epigenetic reprogramming” in their PGCLCs. This is indeed important, but the significance of “epigenetic reprogramming genes (ER genes)” (e.g., DAZL and DDX4) might be harder for a general audience to follow. Perhaps the authors could consider broadening the significance of their study by explaining in the Introduction (1) the transition from gastrulation- to gonadal-stage PGCs in vivo, (2) that DAZL and DDX4 are markers of the gonadal stage, (3) that they therefore generated DAZL and DDX4 reporters and screened for signals to create gonadal-stage PGCLCs, which represents a major milestone. Explaining and emphasizing the transition from gastrulation- to gonadal-stage PGCLCs could make it easier for a general audience to follow, in addition to the “epigenetic reprogramming” aspect.
2. Maturation stage. It is impressive that the authors generate gonadal-stage PGCLCs, but it is

important to (1) clarify which exact stage of gonadal maturation they are mimicking and (2) be consistent in their claims (they variously claim to generate pro-oogonia, oogonia, or oocytes in different sections). For instance, the authors write “c107 AG-VT+ cells displayed features closer to that of ag120 AG-VT+ cells in xrOvaries, which represent a retinoic acid (RA)-responsive, pre-leptotene state of the first meiotic prophase” and “emergence of PLL cells during BMP-driven hPGCLC differentiation agrees with the finding that c107 AG-VT+ cells showed advanced characteristics closer to RA-responsive cells (Fig. 3b), suggesting the role of BMP signaling not only in hPGCLC differentiation and mitotic pro-spermatogonia/oogonia propagation, but also in oogonia-to-oocyte differentiation”. After the BMP-driven expansion culture, the authors’ germ cells could potentially correspond to mitotic, early gonadal-stage germ cells (“FGC1” in PMID 28457750), as they are DDX4+ DAZL+ but are negative/low for ZGLP1, STRA8 and SYCP1 (which mark “FGC2/FGC3” in PMID 28457750), as shown in Fig. 3a of the present submission. Given that the cells are negative/low for ZGLP1, STRA8 and SYCP1, would they appear to mimic an earlier stage of differentiation prior to the RA-responsive “FGC2-like” stage? Alternatively, are these markers turned on in a small subset of cells, such that population heterogeneity masks the extent of marker expression? In Fig. 3g, the authors claim that a small subset of their day 117 in vitro-derived cells correspond to the “yellow” PLL1 cluster, and it would be nice if the authors could show expression of PLL1 signature markers at the single-cell level to substantiate this claim.

3. BMP ligand expression in vivo. It very interesting that BMP seems to be the key signal that matures hPSC-derived PGCLCs to a gonadal stage, as across model organisms, BMP ligands are known to be expressed in the posterior end of the embryo (roughly where the gonads are located). The authors’ analysis of published scRNAseq data shows that gastrulation-stage human embryos express BMP ligands. However, wouldn’t this correspond to the initial BMP input required for PGC specification during gastrulation (e.g., PMID 19410550)? At later developmental stages, are BMP ligands similarly expressed in hindgut endoderm cells or gonadal niche cells? This might be easily answered using published scRNAseq datasets (e.g., PMID 28457750). Additionally, while the authors emphasize the importance of BMP signaling in human PGCLC maturation, there is a considerable body of work on BMP signaling and PGC specification in mouse (as the authors themselves have contributed to, including PMID 19410550). Is there any evidence that in non-human model organisms that (1) BMP is functionally required for later-stage maturation of PGCLCs in the gonads or that (2) BMP ligands are expressed in the gonads? To be clear, this is not a request to perform experiments in non-human model organisms, but this might be worth mentioning in the Discussion (this is mentioned briefly in the second paragraph of the Discussion, but could be elaborated on).

4. Overall concision. Overall, parts of the main text and figures could be made more concise. This would greatly help readers to follow the paper. To highlight one example, the section “TET1 safeguards human germ cells from differentiation into amnion” could be shortened to more concisely convey the message that TET1 deletion leads to spurious amniotic differentiation. In this section, paragraphs on trajectory analysis and gene ontology term analysis should be drastically shortened, and this would not really change the overall interpretation of this section. Likewise, various panels of the figures repeatedly show the same thing, including Figs. 1d-g (in vivo expression of BMP ligands in human embryos by scRNAseq), so the authors should weigh what data should be allocated to Figures vs. Extended Data Figures.

Minor comments:

1. Acronyms. The extensive use of acronyms makes the paper hard to follow, and the authors could

consider introducing a simple cartoon somewhere in the main figures to introduce the various acronyms (e.g., for differentiation stages, cell lines, and fluorescent reporters). For instance, acronyms for multiple differentiation stages are introduced in Fig. 1 and more acronyms for various cell lines and fluorescent reporters are in Fig. 2, which make the figures difficult to decode.

2. Comparing the extent of epigenetic reprogramming. The authors show an impressive loss of genome-wide DNA methylation in their later-stage PGCLCs. However, to the best of this reviewer's knowledge, other hPSC differentiation protocols to create early-stage PGCLCs have also shown some degree of epigenetic changes, although probably more modest. Can the authors discuss, or even directly compare, the extent of epigenetic reprogramming that they achieve against past efforts that made early-stage PGCLCs?

3. Early-stage PGCLC differentiation. Minor comment: what purity of BLIMP1+ and/or TFAP2A+ PGCLCs were generated at day 6 of differentiation? This was mentioned in some (e.g., Fig. 5a) but not all experiments.

4. Fig. 1c: Could the authors list somewhere in the Methods section the list of molecules screened, and their concentrations?

5. Fig. 1c: In this screening experiment, did the authors assess expression of DAZL and DDX4?

6. The authors make the interesting discovery that WNT inhibitor (IWR1) supports PGCLC expansion and blocks de-differentiation into pluripotent cells. Could this be because WNT drives naïve pluripotency, so WNT inhibition restricts the re-acquisition of pluripotency?

7. Fig. 2a: The authors should label the flow cytometry plots with the percentage of cells present in each population/gate, so these data can be quantified and interpreted more readily.

8. Fig. 2a: It is impressive that the authors can upregulate DAZL and VASA. However, it still takes weeks or months in the BMP agonist/WNT inhibitor-based culture system to upregulate these markers, perhaps suggesting that there may be other signals important for DAZL and VASA upregulation?

9. "Morphologically, both AG+DT- and AG+DT+ cells exhibited a spindle shape with a large, ovoid nucleus, indicative of their motile phenotype" – if cell motility is not directly assessed, this claim should be toned down.

10. Fig. 2c, bottom panel: "AGVT" is used twice in the legend to refer to different things. Is this a typo?

11. Fig. 2e: What are the cells with fewer than 46 chromosomes?

12. A previous study (PMID 31754036) highlighted a core set of 13 genes (DAZL, PDLIM1, BAIAP2L1, ZNF800, REC114, MAGEB16, TDRD12, SYCP3, FKBP6, MAEL, DDX4, SMC1B, SLC25A31) that are upregulated upon PGC entry into the gonads, which were conserved across multiple animal species; are these 13 genes likewise turned on by BMP treatment?

13. "The up-regulated genes included ER genes and were enriched for gene ontology (GO) terms such as "BMP signaling pathway/negative regulation of transcription from RNA polymerase II promoter" (e.g., ID1-4, GATA2/3/6, TBX1/3), and "inactivation of MAPK activity" (e.g., DUSP4, DUSP6) (Extended Data Fig. 3d, e)." – Maybe the authors could clarify by saying "BMP-upregulated genes" (which might be clearer than just "upregulated genes")

14. Fig. 3a: Interestingly, DNMT3L is still repressed even in the absence of BMP.

15. Fig. 3c: To make it easier to interpret this figure, the authors should consider directly defining what the acronyms are in the figure itself.

16. Fig. 3d: To make it easier to interpret this figure, could the authors explicitly label the top row "in vivo" and the bottom row "in vitro"?

17. Fig. 3g: It might be helpful to include the cluster proportion percentage proportions somewhere, especially for key samples like c117.
18. Fig. 3h: This figure is somewhat redundant with Fig. 3g, and the authors could consider moving it to the Extended Data figures.
19. Fig. 5b: Maybe it would be clearer if the authors labeled the y-axis as “Cell number fold change”, which is more specific than “fold change”.
20. “On the other hand, TET1 KO hPGCLC-derived cells could be cultured at least until c42 (Fig. 5b), indicating that TET1 is not necessarily essential for hPGCLC propagation.” – The meaning of this sentence is not clear, as it seems to suggest that TET1 is dispensable, and seems to contradict the preceding sentence.
21. Fig. 5e – This figure is hard to interpret. Where do the wild-type and TET1^{-/-} cells reside in this plot? Would it be clearer to show two different UMAP plots of wild-type vs. TET1^{-/-} cells to depict expression of amniotic markers (e.g., ISL1) in the latter, but not the former?
22. The authors claim that trophoctoderm-like cells can emerge after TET1 deletion (e.g., Fig. 6i). While they show expression of some markers (“SDC1, KRT7 [trophoctoderm (TE), TFAP2A, GATA3, and HAND1 [AM/TE markers]”), what about other markers like ELF5 or additional trophoctoderm markers (e.g., those reported by PMID 26862703)?
23. Fig. 5h – This panel could be moved into the Extended Data Figures, as it does not add much to the identity of the various cell-types.
24. Claims about TET1 repressing bivalent promoters are interesting but may need to be toned down slightly, since the authors did not show direct binding of TET1 to those genomic elements using ChIP-seq or another technique.
25. In the Discussion, the authors claim that expansion occurs over ~4 months, but in Fig. 6i they indicate ~5 months: maybe these time durations can be standardized.
26. In the Discussion, the authors make the interesting comment that epigenetic reprogramming and PGCLC specification happen contemporaneously in mouse, but are separable in human. As the authors’ work focuses on in vitro system, as opposed to making broad claims about human and mouse development, maybe they could highlight that these two processes are separable in an in vitro human system, as it is less clear whether the same claim could be made in vivo.

Author Rebuttals to Initial Comments:

Rebuttal: *Nature* manuscript 2023-15-08623

We would like to sincerely thank the Referees for their constructive comments, which we have used as the basis for revising our manuscript. We would like to begin by providing a summary of the manuscript revision.

Summary of the manuscript revision

The major requests made by the Referees in regard to our manuscript were:

for additional experiments:

1. to reproduce BMP-driven hPGCLC differentiation in additional male and female cell lines;
2. to provide an orthogonal validation of the expression of fluorescent reporters;
3. to validate the reduction of MAPK/ERK signaling during BMP-driven hPGCLC differentiation; and

for additional analyses:

4. to compare gene expression and the dynamics of DNA methylation reprogramming between BMP-driven and xrOvary-based hPGCLC differentiation;
5. to perform additional analyses on the impact of *TET1* KO on the demethylation of promoters of ER genes and imprint DMRs; and
6. to perform further characterization of amnion (AM)/trophectoderm (TE)-like cells in the *TET1* KO culture.

We here provide our responses to these points as **General Responses 1–6**.

General Response 1. We performed BMP-driven hPGCLC differentiation using one additional hiPSC subline and two new hiPSC lines: male 585B1 hiPSCs bearing *TFAP2C-EGFP* (AG) and *DDX4/human VASA homologue-tdTomato* (VT) alleles (M1-AGVT), male 1383D6 hiPSCs with no reporters (M2)¹, and female 1390G3 AGVT hiPSCs (F2-AGVT)². See Fig. 1a in the revised manuscript for the summary of acronyms used in this study.

Under the condition for male 585B1 AGDT (*DAZL-tdTomato*: M1-AGDT) hPGCLC differentiation, hPGCLCs from M1-AGVT exhibited a robust propagation and began to up-regulate VT from ~c32, and ~95% of the cells became VT⁺ at c92, with an overall expansion of ~>3×10⁷-fold and an average doubling time of ~4.2 days (Fig. 2b, Extended Data Fig. 4e, f in the revised manuscript). Under the same condition, hPGCLCs from M2 exhibited a good propagation and IF analysis revealed that at c54

and c86, ~55% and ~92% of TFAP2C⁺ cells became DDX4⁺, respectively, with an overall expansion of ~10⁵-fold and the average doubling time of ~5.1 days (Fig. 2b, Extended Data Fig. 4g–i in the revised manuscript).

Under the condition for female NCLCN AGVT (F1-AGVT) hPGCLC differentiation, hPGCLCs from F2-AGVT exhibited a slow but stable propagation and up-regulated VT as early as ~c26, and nearly all the cells became VT⁺ at c98, with an overall expansion of ~>3×10³-fold and an average doubling time of ~8.8 days (Fig. 2b, Extended Data Fig. 4n, o in the revised manuscript).

RNA-seq and EM-seq analyses of hPGCLC-derived cells from these additional three hiPSC lines revealed their appropriate transcriptome maturation and DNA methylome reprogramming (Fig. 3a, b, Fig. 4 in the revised manuscript). These findings demonstrate that BMP signaling reproducibly promotes the differentiation of hPGCLCs from at least four independent hiPSC lines (M1-AGDT/M1-AGVT, M2, F1-AGVT, F2-AGVT) into mitotic pro-spermatogonia or oogonia, while their propagation/differentiation dynamics show a degree of line-dependent heterogeneity. We provided relevant data and statements in the revised manuscript (Fig. 2–4, Extended Data Fig. 4, “**BMP signaling induces hPGCLC differentiation,**” “**Transcriptome dynamics,**” and “**DNA methylome reprogramming**” sections).

General Response 2. First, we performed IF analysis of DAZL and DDX4 expression in M1-AGDT hPGCLC-derived cells at c89. This revealed that 1) essentially all DT⁺ cells were DAZL⁺, with the expression levels of DT and DAZL showing an excellent correlation ($r = \sim 0.65$); 2) on the other hand, ~one-third of DT⁻ cells exhibited low/middle-level DAZL positivity; and 3) essentially all DAZL⁻ cells were DT⁻ (Supplementary Figure 1a). In accord with these findings, 4) essentially all DT⁺ cells were DDX4⁺, with the expression levels of DT and DDX4 showing a strong correlation ($r = \sim 0.61$); 5) a fraction (~15%) of DT⁻ cells exhibited low/middle-level DDX4 positivity; and 6) the vast majority of DDX4⁻ cells were DT⁻ (Supplementary Figure 1a). These findings demonstrate that the DT positivity is a powerful quantitative indicator for DAZL (and DDX4) expression, while on the other hand, the DT⁻ cells (at a late stage) include a fraction of DAZL- (and DDX4-) expressing cells at low/middle levels, which may be due to a sporadic selective transcriptional/post-transcriptional silencing of the DT allele during BMP-driven hPGCLC differentiation.

Next, we performed IF analysis of DAZL and DDX4 expression in M1-AGVT hPGCLC-derived cells at c89. This revealed that 1) the expression levels of VT and DDX4 were highly correlated in all expression-level ranges ($r = \sim 0.73$) and 2) DAZL was broadly expressed from VT^{-/low} to VT^{high} cells, with the expression levels of VT and DAZL showing a mild correlation ($r = \sim 0.58$) (Supplementary Figure 1b). These findings demonstrate that VT is a faithful reporter for DDX4 expression, and are consistent with the notion that DAZL begins to be expressed earlier than VT. The IF analysis for F1-AGVT hPGCLC-derived cells at c91 gave essentially the same results (Supplementary Figure 1c).

Furthermore, we performed IF analysis of TFAP2C and DDX4 expression in F1-AGVT hPGCLC-derived cells at c109. This revealed that 1) essentially all AG⁺ cells were TFAP2C⁺ and 2) all AG⁻VT⁺ cells we detected were TFAP2C⁻ and DDX4⁺ (Supplementary Figure 1d), demonstrating that AG is also a faithful reporter for TFAP2C expression.

Collectively, these findings demonstrate that both DT and VT positivity monitor DAZL and DDX4 expression in a highly quantitative manner, while care should be taken for DT⁻ cells, which include a fraction of DAZL⁺ (and DDX4⁺) cells, although the majority are indeed DAZL⁻ (and DDX4⁻). Accordingly, we assume that the detection of a relatively high level of *DDX4* in one, but not the other, replicate for the c82 AG⁺DT⁻ cells (Fig. 2f) was due to a relatively large proportion of *DDX4*-expressing cells in the former, and that the detection of *DDX4* at a low level in AG⁺VT⁻ cells at c44, c65, and c86 and of *TFAP2C* at a low level in the AG⁻VT⁺ cells at c107 (Fig. 3a) was due to an inclusion of VT^{low} and AG^{low} cells, respectively, upon FACS of the AG⁺VT⁻ and AG⁻VT⁺ cell population (Fig. 2a) [note also that the “yield” mode for FACS inevitably sorts in a fraction of non-gated cells]. We provided relevant data and discussion in the revised manuscript (“**BMP signaling induces hPGCLC differentiation**” section, 2nd paragraph, Supplementary Figure 1 and Discussion).

General Response 3. We quantified the level of phosphorylated ERK (pERK) in hPGCLC-derived cells cultured with or without BMP2 at c32 by Western blot analysis. The results revealed that in the presence of BMP2, the pERK level was markedly reduced in hPGCLC-derived cells (Extended Data Fig. 5h, i in the revised manuscript), demonstrating that the MAPK/ERK signaling is attenuated by BMP signaling. We provided relevant data and statements in the revised manuscript (Extended Data Fig. 5h, i, “**Transcriptome dynamics**” section, 2nd paragraph).

General Response 4. As to the transcriptomes, as shown in Fig. 3a, b in the revised manuscript, BMP-driven and xrOvary-based hPGCLC differentiation exhibited a remarkable similarity with regard to the genes contributing to PC1 (~31%) and PC2 (~17%), but interestingly, showed contrasting expression profiles for the genes contributing to PC3 (~8.6%) (Extended Data Fig. 5b). The genes contributing to the negative scores for PC3 (up-regulated during BMP-driven hPGCLC differentiation) were enriched with GO terms such as “angiogenesis,” “extracellular matrix organization,” and “positive/negative regulation of transcription from RNA polymerase II promoter,” and included *ID1/2/3*, *GATA2/3*, *MSX2*, and *HAND1*, while those contributing to the positive scores for PC3 (up-regulated during xrOvary-based hPGCLC differentiation) were enriched with GO terms such as “brain development,” “skeletal muscle cell differentiation,” and “positive/negative regulation of transcription from RNA polymerase II promoter,” and included *KHDC3L*, *SMARCA2*, and *DPPA5*. We provided relevant data and statements in the revised manuscript (Extended Data Fig. 5b, “**Transcriptome dynamics**” section, 1st paragraph).

As to the DNA methylome, first, we would like to clarify that we measured the DNA methylome for BMP-driven hPGCLC differentiation (this study) using enzymatic methyl sequencing (EM-seq) with paired-end sequencing^{3,4}, whereas, in the previous studies^{1,2,5}, we and others measured the DNA methylome for xrOvary-based hPGCLC differentiation and fetal germ cells *in vivo*, respectively, using whole genome bisulfite sequencing (WGBS) with single-end sequencing. The DNA methylomes of hiPSCs (F1 and F2) measured by these two methods were highly similar (Extended Data Fig. 7a in the revised manuscript)⁴, making quantitative comparison of the DNA methylomes measured by these two methods appropriate. On the other hand, we noted that compared to WGBS with single-end sequencing, EM-seq with paired-end sequencing, but not single-end reads (simulated computationally), detects 5mC on the satellite elements (mainly the α satellites on the centromeres) with a higher coverage/sensitivity (Extended Data Fig. 8c, d in the revised manuscript).

To compare the dynamics of DNA methylation reprogramming between BMP-driven and xrOvary-based hPGCLC differentiation, we included the DNA methylomes of hPGCLC-derived cells from M1-BTAG at aggregation day (ag) 35 and 77 in xrOvaries² for analysis. Scatter-plot comparisons and PCA revealed that BMP-driven c32 and c82 cells (M1-AGDT) were very similar to ag35 and ag77 cells (M1-BTAG) in both genome-wide 2 kb bins and promoters, and, as shown in the original manuscript, c122 cells (M1-AGDT) were similar to ag120 cells (F2-AGVT) (Fig. 4a, Extended Data Fig. 7a, c in the revised manuscript). On the other hand, BMP-driven hPGCLC differentiation with M1-AGVT, M2, F1-AGVT, and F2-AGVT cells exhibited DNA methylation reprogramming with faster kinetics: c82 (M1-AGVT), c76 (M2), c72 (F1-AGVT), and c68 (F2-AGVT) cells were similar to ag120 cells (F2-AGVT) (Fig. 4a, Extended Data Fig. 7a, c in the revised manuscript).

As shown and discussed in the original manuscript, the final state of DNA methylome reprogramming can be compared based on DNA demethylation escapees. As described earlier, while EM-seq and WGBS provide highly consistent outcomes, the former detects satellite sequences with a higher coverage [note, however, that the satellite sequence detected by WGBS exhibited 5mC levels similar to those detected by EM-seq (Extended Data Fig. 8b in the revised manuscript)]. Accordingly, except the satellite sequences, the escapees in c122 (M1-AGDT), c82 (M1-AGVT), c76 (M2), mitotic pro-spermatogonia *in vivo*, c127 (F1-AGVT), c68 (F2-AGVT), ag120 (F2-AGVT), and oogonia *in vivo* were highly similar to one another in terms of both genomic regions (predominantly transposable elements) and their proportions (Fig. 4c–e in the revised manuscript). Additionally, we compared the 5mC levels in relevant regulatory elements, i.e., promoters of ER genes and genes for “meiotic cell cycle” in these cells, which revealed that such elements were nearly fully demethylated in all cell types (Extended Data Fig. 8a in the revised manuscript).

Collectively, these findings indicate that 1) BMP-driven and xrOvary-based hPGCLC differentiation can induce DNA methylation reprogramming with similar kinetics; 2) on the other hand, depending on the hiPSC lines used or culture conditions, including BMP concentrations, BMP-driven hPGCLC differentiation can also induce DNA methylation reprogramming with faster kinetics; and 3) BMP-driven and xrOvary-based hPGCLC differentiation lead to a similar final DNA methylome, which recapitulates that in mitotic pro-spermatogonia/oogonia *in vivo*. We provided relevant data and

statements in the revised manuscript (Fig.4, Extended Data Fig. 7, 8, “**DNA methylome reprogramming**” section).

General Response 5. We measured the 5mC-level differences of the ER-gene promoters and imprint DMRs between wild-type and *TET1* KO cells, and compared them to those of other promoter classes. At c12, before which wild-type and *TET1* KO cells exhibited an equivalent expansion and relatively similar transcriptome profiles (Fig. 5a and Extended Data Fig. 10g), in *TET1* KO cells, all promoter classes, and particularly poised promoters, showed elevated 5mC levels (Fig. 6a, e, f). We found that ER genes, which were enriched with poised promoters, particularly early ER genes, exhibited highly elevated promoter 5mC levels, while imprint DMRs showed 5mC-level elevation only slightly higher than the genome-wide average (Fig. 6f). Accordingly, in *TET* KO cells, ER genes exhibited a trend for down-regulation and were highly over-represented in down-regulated DEGs (Fig. 6h, Extended Data Fig. 10l).

At c42, while direct evaluation of the primary role of TET1 in the *TET1* KO phenotype would not be straightforward due to the impaired growth and aberrant transcriptome of *TET1* KO cells (Fig. 5a and Extended Data Fig. 10g), *TET1* KO cells nevertheless failed to properly demethylate all genomic elements, and despite substantial genome-wide 5mC-level differences, DMRs between wild-type and *TET1* KO cells were enriched in promoters, CGIs, and coding sequences (CDSs) (Extended Data Fig. 12a). Interestingly, the 5mC-level differences in poised and ER-gene promoters were large, and those of imprint DMRs were less than the genome-wide average (Extended Data Fig. 12b, c). As to the expression, the vast majority of ER genes were down-regulated, with a significant enrichment in down-regulated DEGs, while interestingly, genes with poised promoters/enhancers, while exhibiting failure of demethylation, did not show a general trend for down-regulation (Extended Data Fig. 10l, 12d, e). Collectively, these findings favor the notion that TET1 primarily demethylates regulatory elements, particularly poised and ER-gene promoters, and such demethylation is correlated with the up-regulation of ER genes, but not genes with poised promoters in general. We provided relevant data and statements in the revised manuscript (Fig. 6, Extended Data Fig. 12, “**TET1 represses bivalent promoters and demethylates regulatory elements**” section).

General Response 6. We analyzed the scRNA-seq data of wild-type and *TET1* KO hPGCLC-derived cells at c18 using the cell-state prediction tool suggested by the Referee⁶. Consistent with our analysis in the original manuscript, this revealed that cluster 8 and 7 cells were similar to TE and AM, respectively, and cluster 4–6 cells were similar to extraembryonic mesoderm (Extended Data Fig. 11h in the revised manuscript). We provided relevant data and statements in the revised manuscript (Extended Data Fig. 11h, “**TET1 safeguards human germ cells from differentiation into amnion**” section, 3rd paragraph).

Referees' comments:

Referee #1 (Remarks to the Author):

The results presented by Murase et al. are interesting and add information about the process by which human primordial germ cells undergo their fetal differentiation and – crucially - global DNA de-methylation. The authors identify a heretofore unknown role for BMP signalling in proliferation and epigenetic re-programming of PGCLCs (adding to its already known roles in PGC specification and oogenesis). The authors also conclude that human PGC de-methylation occurs largely through a passive, replication-dependent process.

Generally, the work has been performed in a very extensive and careful manner.

Response 1. We sincerely thank the Referee for the encouraging comments.

Nonetheless, this study suffers from a limitation. The main work is performed in cells derived from two iPSC lines, which differ from each other in quite substantial ways. This makes conclusions regarding robustness challenging. The two lines differ in chromosome complement (XX vs. XY), reporter gene complement (AGDT vs. AGVT) and culture requirements (differing levels of BMP2 used per cell line, presumably related to the cellular state of the initial iPSC lines). The work examining TET1 function in PGCs was performed in two sub-cultured lines but uses a third reporter complement (BTAG) further complicating analysis. It is strongly recommended to reproduce the key results (kinetics of reporter gene activation and DNA de-methylation timing) in at least one additional cell line of each sex chromosome complement (with matching reporter gene constellations) to determine if the differences in dynamics relate to XX vs. XY cell lines or are more strongly related to the variation in starting states from different iPSCs.

Response 2. Please see **General Responses 1** and **4** above.

Other points:

An important point relates to nomenclature. The authors refer to the BMP2 treated cells as being “oogonia-like and mitotic pro-spermatogonia-like,” however given no evidence for sexually dimorphic gene expression in these cells, would not the sex neutral term “gonocyte” be a more appropriate label?

Response 3. We analyzed the expression of Y-chromosome-specific genes, which revealed that such genes, including *RPS4Y1*, *ZFY*, *TTY15*, *USP9Y*, *DDX3Y*, *UTY*, *KDM5D*, and *EIF1AY*, were specifically expressed during male hPGCLC differentiation (Fig. 3a and Extended Data Fig. 5a in the revised manuscript), demonstrating that male and female hPGCLC-derived cells show sexually dimorphic gene expression. We provided relevant data and statements in the revised manuscript (Fig. 3a, Extended Data Fig. 5a, “**Transcriptome dynamics**” section, 1st paragraph).

Figure 1: Panel H: what fraction of cells become DAZL positive?

Explain to the reader why you generated reporter strains for DAZL and DDX4/VASA expression.

Response 3. The percentage of DAZL⁺ cells in Fig. 1h in the original manuscript (Extended Data Fig. 2h in the revised manuscript) was ~19.2%. The reasons why we generated reporter lines for *DAZL* and *DDX4* were: 1) they are two representative genes signifying the differentiation of hPGCs into mitotic pro-spermatogonia/oogonia; 2) their expression dynamics are different, i.e., *DAZL* is expressed earlier and at a higher level than *DDX4*; and 3) it was not necessarily obvious which gene works better as a reporter for mitotic pro-spermatogonia/oogonia differentiation. We provided relevant data and statements in the revised manuscript (Legend to Extended Data Fig. 2h, “**BMP signaling induces hPGCLC differentiation**” section, 1st paragraph).

Figure 2:

Panel A: The mis-match in timepoints between the two cell lines makes this figure hard to read (some timepoints line up, while others do not). The differing culture conditions also make this.

Response 4. As described in **General Response 1**, BMP signaling reproducibly promotes the differentiation of hPGCLCs from at least four independent hiPSC lines into mitotic pro-spermatogonia or oogonia, although their propagation/differentiation dynamics show a degree of line-dependent heterogeneity.

In response to the Referee’s comment, to make Fig. 2a easier to read, for the flow cytometric plots, we presented only representative timepoints in the revised manuscript. As in the original manuscript, for the growth curve and for the proportion of the respective fluorescent-marker⁺ cells, we presented all timepoints during the BMP-driven hPGCLC differentiation (Fig. 2b, 2c).

Panel B: Why not show the two cell lines on the same graph? It is hard to compare the dynamics as is.

Response 5. We plotted the growth curve of all cell lines we used on the same graph in the revised manuscript (Fig. 2b).

Panel C: Legend for second plot: should the grey bar be labelled VT?

Response 6. We would like to thank the Referee for pointing this error out. We corrected the label in the revised manuscript (Fig. 2c).

Panel G: At c82 the results for DAZL-Tomato selection seem strange. For DAZL itself the result is clear (AG selected cells express substantially less DAZL mRNA than AGDT cells). However, the results for DDX4 do not agree (AG selected cells express nearly identical levels of DDX4 mRNA as AGDT cells). Given that the authors argue that DDX4 is a later marker of differentiation than DAZL this seems counterintuitive.

Response 7. Please see **General Response 2.**

Figure 3:

Panel A:

Reporter vs. marker expression:

VT cells are selected to be AG(-), if so, why do they show equivalent levels of TFAP2C mRNA? Why do certain AG cells show similar levels of DDX4 mRNA as AGVT do? Is the DDX4 protein under post-transcriptional regulation in these cells? Immunofluorescence with DDX4 antibody to examine the level and frequency of expression of endogenous protein in later cultures of AG and AGVT cells would help to clarify this point.

Response 8. Please see **General Response 2.**

Expression dynamics:

In RNA-Seq it appears that selection for DAZL also selects for DDX4 (low level of DAZL mRNA appears in the Tomato-negative population). Subsequent culture of AGDT cells leads to expression of SYCP3. Selection for DDX4 expression (via AGVT) also selects for high DAZL mRNA levels (though at later timepoints AG cells also express DAZL mRNA). In the AGVT harboring cell line, all DAZL expressing cells also express SYCP3. Is the repression of SCP3 transcription less potent in female cells? What about DNA methylation of its promoter?

Response 9. Please see **General Response 2.**

We examined the relationship between the expression and the promoter methylation of SYCP3. As shown in Figure 1 for the Referee, SYCP3 expression showed a clear negative correlation with its promoter 5mC levels in both male (M1-AGDT/VT, M2) and female (F1/F2-AGVT) hPGCLC-derived

cells. To more strictly evaluate whether the repression of *SYCP3* is less potent in female cells would require an analysis of additional male and female lines, which we consider beyond the scope of this manuscript.

Panel B: The relationship between experiments in this PCA is unclear. The labelling from 1390G3-AGVT (from previous experiments cultured with xrOvaries?) is not at all clear from the legend. Also, which cells are selected for which reporters? A more direct comparison between BMP-driven hPGCLC differentiation and differentiation in xrOvaries is needed.

Response 10. We are sorry for the confusing labeling and legend to Fig. 3b in the original manuscript. In response to the Referee's request, we revised the figure and its legend so that the labeling is clearer and to clarify which cell lines were selected for which reporters (Fig. 3b).

For a more direct comparison between BMP-driven and xrOvary-based hPGCLC differentiations, please see **General Response 4**.

Panel C: Have the authors observed any batch effects between in vivo and in vitro data sets? If yes, how have they been resolved?

Response 11. Please note that the method for batch correction is described in the "**Single cell RNA-seq (scRNA-seq) library preparation, data processing and analysis**" section of the **Methods**.

Panel I: How comparable are the enrichments and statistical significances of the different GO terms identified under in vitro versus in vivo conditions?

Response 12. We showed the statistical significance of the enrichment of each GO term for the DEGs between *in vitro* and *in vivo* cells in the revised manuscript (Extended Data Fig. 6f, h, Supplementary Table 4).

Figure 4:

Panel A: Separating promoters into CGI-rich and CpG-poor classes would be useful to compare the dynamics.

Response 13. We would like to thank the Referee for this comment. We classified the promoters into high, intermediate, and low CpG promoters (HCP, ICP, and LCP, respectively), and presented their 5mC-level dynamics in the revised manuscript (Fig. 4a, 6a).

Panel A: The authors identified two ways to reprogram the hPGCLC DNA methylome in vitro, via BMP-supported growth versus induction in xenotransplanted ovaries. It is important to perform a careful genome wide comparison between the temporal kinetics and ultimate methylome states between the two conditions. It is particularly interesting to address/discuss conditions of passive, replication-coupled loss of DNA methylation. How relevant do the authors consider the differences in UHRF1 RNA expression under the two conditions? Are there differences in the degree of residual DNA methylation residing at promoters of key meiotic genes?

Response 14. Please see **General Response 4**.

As the Referee noted, the mRNA expression level of *UHRF1* was slightly lower in hPGCLC-derived cells in xrOvaries [$\log_2(\text{RPM}+1) = \sim 3$, i.e., ~ 10 copies/cell⁷] than those cultured with BMP2 [$\log_2(\text{RPM}+1) = \sim 4$, i.e., ~ 20 copies/cell] (Fig. 3a). Given the findings described in **General Response 4**, we assume that the expression level of *UHRF1* is sufficiently low under both the BMP-driven and xrOvary-based hPGCLC differentiation conditions [also, note that UHRF1 translocated into the cytoplasm with BMP2 (Extended Data Fig. 9j, k)] to promote replication-coupled, passive DNA demethylation. The DNA demethylation kinetics would depend on the level of UHRF1 and the extent of the cells' propagation, the latter of which is extensive during BMP-driven hPGCLC differentiation but would be modest during xrOvary-based hPGCLC differentiation². We provided a relevant discussion in the revised manuscript ("**Discussion**" section, 2nd paragraph).

Panel E: Escapers from DNA de-methylation in both BMP2 treated hPGCLC lines show an increase in Satellite sequences (though not in xrOvary cultured cells). How extensive is the extent of this escape within the genome and how do these elements behave if plotted as in Figure 4A?

Response 15. Please see **General Response 4**. We plotted the DNA demethylation dynamics of satellite sequences in Extended Data Fig. 8b in the revised manuscript.

Figure 5:

Panel C: What does the expression of ER genes look like in TET1 KO cells? Given the lack of a differentiation reporter in these cells it is not possible to isolate them, but expression analysis in the BTAG population would be interesting.

Response 16. Please note that we have presented the expression of ER genes in *TET1* KO cells in Extended Data Fig. 6j in the original manuscript (Extended Data Fig. 10l in the revised manuscript); ER genes were not up-regulated in a proper manner in *TET1* KO cells. Please also see **General Response 5**.

Figure 6:

Panel D: The authors do not provide clear definitions of how “poised” vs. “bivalent” states are defined in their analysis.

Panel F: How are enhancers assigned to specific genes for expression analysis?

Response 17. We thank the Referee for these comments. We provided detailed information on the clustering, annotation, and enhancer assignment in the “**Epigenome-based clustering of cis-regulatory elements**” section of the **Methods**. Furthermore, we provided an additional heatmap showing averaged levels of epigenetic modifications of each cluster in the revised manuscript (Fig. 6d).

Panel F: The effect of TET1-deficiency on gene expression seems small in magnitude. Is this driven by variation in how many cells induce expression vs. low levels of increase in all cells. The authors should examine the expression levels of genes associated with these promoters/enhancers in their single-cell RNA-Seq analysis to address this point.

Response 18. Please note that Fig. 6f in the original manuscript shows the expression-level differences between wild-type and *TET1* KO c12 cells of all the genes, irrespective of their expression in the relevant cell types, bearing promoters/enhancers categorized as active (10,751/3,087), bivalent (4,940/11,652), poised (4,173/111,546), or silent (17,676/265). This data demonstrates that genes broadly categorized as bearing bivalent promoters/enhancers (4,940/11,652) show statistically significant up-regulation compared to genes in other categories. 226 genes were up-regulated [fold-change ≥ 2] in the *TET1* KO c12 cells (Extended Data Fig. 10i) and among such genes, those with bivalent promoters (99) were indeed both enriched (Fig. 6h) and substantially up-regulated [fold-change ≥ 2] (Supplementary Table 8 in the revised manuscript). We clarified these points in the revised manuscript (Fig. 6f, Supplementary Table 8, “**TET1 represses bivalent promoters and demethylates regulatory elements**” section, 2nd and 5th paragraph).

The result section of the manuscript is written in a very dense manner and is from time to time hard to follow, particularly also because of the use of many acronyms. In contrast, the discussion is written in a clear and understandable manner. The authors are recommended to improve the clarity of the results section.

Response 19. We have made a substantial revision of the entire manuscript to improve both the clarity and concision to meet with the requests from the Referees and the journal.

Referee #2 (Remarks to the Author):

The manuscript by Murase et al. addresses a pertinent issue regarding the use of human primordial germ cell models, specifically the scarcity and quality of epigenetic reprogramming. Building upon their prior experiments, the authors have identified conditions that enable robust expansion of both male and female human primordial germ cell-like cells (hPGCLCs) in a simple 2D culture system through the addition of BMP ligands. Importantly, this approach not only maintains the expression of key germline transcription factors but also induces a robust DNA demethylation process similar to that observed when hPGCLCs are re-aggregated with ovary/testes soma.

Utilizing this model, along with RNAseq, scRNAseq, and epigenomic profiling, as well as genetic manipulations, the authors present several novel findings. They propose that the addition of BMP after the specification stage induces germline epigenetic reprogramming, attenuates the MAPK/ERK pathway, and consequently represses de novo and maintenance DNA methylation machinery. Furthermore, they suggest that DNA demethylation occurs through a passive replication-dependent process, while highlighting the role of TET1 in safeguarding hPGCLCs against differentiation towards amnion-like cells.

In general, this is a well-executed study that establishes a model for expanding hPGCLCs up to the mitotic spermatogonia/oogonia stage. While the technical advancements made are somewhat incremental, as they build upon previously published protocols, the study's practical utility will be of broad interest. In my opinion, the mechanistic aspects of the study are of high quality; however, there are some key validation experiments that are missing. The most significant concern with the paper lies in the interpretation of the TET1-KO phenotype, as the experiments conducted are insufficient to assess the role of hydroxymethylation in global DNA demethylation in PGCLCs. Instead, the authors should focus on the amnion/TE differentiation phenotype, which is interesting in its own right.

Overall, with some textual changes and the inclusion of a few additional experiments, I believe this paper has the potential to be published in high-impact journals such as Nature Cell Biology or even Nature. Detailed criticism is provided below.

Response 1. We sincerely thank the Referee for the encouraging comments.

Major criticism:

1. The authors claim that the addition of BMP2 allows for PGCLC expansion with minimal sorting purification, resulting in MAPK/ERK modulation and repression of DNA methylation machinery. However, several controls are missing. Firstly, it is necessary to demonstrate the long-term effects of BMP2 treatment in both male and female cells compared to a control group without BMP2. Currently, these controls are only included for the AGDT line and not the AGVT line.

Response 2. In response to the Referee's comment, we cultured hPGCLCs from F1-AGVT (see **General Response 1**) without BMP2 and found that they showed a poor expansion, exhibited a substantial level of de-differentiation as early as c12, and were difficult to maintain with a simple dilution after c42, at which point they did not show up-regulation of VT (Extended Data Fig. 4l, m in the revised manuscript), indicating that as in the case for hPGCLC from M1-AGDT, BMP signaling plays a key role in stabilizing the germ-cell fate and promoting hPGCLC differentiation. We provided relevant data and statements in the revised manuscript (Extended Data Fig. 4l, m, "**BMP signaling induces hPGCLC differentiation**" section, 3rd paragraph).

Additionally, it is unclear why different markers are used for female and male cells, despite both Dazl and Vasa being expressed in both sexes.

Response 3. Please see **General Response 1**. The reasons why we generated reporter lines for *DAZL* and *DDX4* were: 1) they are two representative genes signifying the differentiation of hPGCs into mitotic pro-spermatogonia/oogonia; 2) their expression dynamics are different, i.e., *DAZL* is expressed earlier and at a higher level than *DDX4*; and 3) it was not necessarily obvious which gene works better as a reporter for mitotic pro-spermatogonia/oogonia differentiation. We provided relevant data and statements in the revised manuscript (Fig. 1b, "**BMP signaling induces hPGCLC differentiation**" section, 1st paragraph).

Moreover, the MAPK/ERK effect of BMP stimulation is solely shown through RNAseq, which is insufficient to support the claim made in the abstract. To support this claim, it is necessary to perform pERK staining to conclusively demonstrate BMP-dependent ERK regulation.

Response 4. Please see **General Response 3**.

2. Figure 3 presents single-cell RNAseq analysis integrated with *in vivo* samples. The analysis clearly indicates that only the EM1/M1/2/3/PLL1 clusters are shared between *in vivo* and *in vitro* samples, with clear off-target populations in the PLL2 cluster. While this is somewhat discussed in the text, it should be more explicitly stated in the results section rather than only in the discussion. Specifically, lines 299-303 should be revised to provide a more nuanced and clear explanation.

Response 5. In response to the Referee's comment, we discussed that the PLL2 cluster is an aberrantly differentiated cell population and revised the relevant sentence in the revised manuscript ("**Transcriptome dynamics**" section, 4th paragraph).

3. The epigenomic analysis shows that DNA demethylation in vitro is significantly delayed compared to the dynamics observed in vivo. Although this is mentioned in the discussion, it is not adequately addressed in the results section. Additionally, male cells appear to aberrantly demethylate their imprints (PEG3; IGF2R), which is linked to hypermethylation in the iPSC state. This observation should be mentioned, and the claim made in lines 344-345 should be nuanced accordingly.

Response 6. In response to the Referee's comment, we discussed the DNA demethylation kinetics of *in vitro* and *in vivo* cells and the remaining aberrant imprints in the **Results** section in the revised manuscript ("**DNA methylome reprogramming**" section, 1st paragraph).

4. The authors demonstrate poor X chromosome reactivation (XCR) in their system, as observed in other in vitro models. While this limitation is appropriately discussed, the authors suggest that it is likely due to the aberrant epigenetic status of iPSCs (lines 367-369; 423). However, the cited papers fail to provide functional evidence supporting this hypothesis. While this speculative hypothesis can be mentioned in the discussion, it should be removed from the results section since the authors do not investigate other marks besides 5mC. Additionally, the claim that XCR depends on promoter demethylation on Xi (line 421) is not accurate. While there is a correlation between demethylation and reactivation, the functional link has not been established yet, as it could be related to the delayed erasure of other marks, such as histone modifications.

Response 7. In response to the Referee's comment, we removed our speculation on the potential mechanism of poor XCR in hPGCLC-derived cells from the **Results** section and provided an appropriate description of the correlation between promoter demethylation and XCR in the revised manuscript ("**ER gene regulation and X-chromosome reactivation**" section, 2nd paragraph).

5. One of the more intriguing claims is that the majority of 5mC is passively lost during PGCLC differentiation. Although the results broadly support this statement, some nuances should be added. Firstly, the depletion of UHRF1 and DNMT3B in vitro is less pronounced compared to in vivo conditions (Figure 3a). Secondly, there is no conclusive experiment demonstrating that active demethylation does not drive the demethylation of ER or imprinted genes, as suggested by Hackett et al. in mice in 2013. The TET1KO experiments are inconclusive regarding the dynamics of DNA demethylation, as there is off-target differentiation and limited conclusions can be drawn. Furthermore, there appears to be delayed activation of early ER genes (GTSF1, SPATA22, PRAME) in the KO group as early as c12, which may indicate TET1 involvement at CGI and/or imprints. Therefore, the conclusions in this area should be more nuanced.

Response 8. We would like to thank the Referee for these comments. As to the first point, the expression levels of *UHRF1* and *DNMT3B* were at a sufficiently low level [$\log_2(\text{RPM}+1) = \sim 4$ and ~ 3 (i.e., ~ 20 and ~ 10 copies/cell), respectively (Fig. 3a) ⁷], *UHRF1* was translocated into the cytoplasm, and *de novo* DNMT activity monitored by CpA methylation was extremely low in hPGCLC-derived cells (Extended Data Fig. 9i–k). On the other hand, as pointed out by the Referee, the down-regulation of *UHRF1* and *DNMT3B* mRNA was somewhat more pronounced in mitotic pro-spermatogonia/oogonia *in vivo* (Fig. 3a). This might be the reason why the DNA demethylation kinetics was slower *in vitro*. We discussed this point in the revised manuscript (“**Discussion**” section, 2nd paragraph).

As to the second point on the ER genes and imprint DMRs, please see **General Response 5**.

6. The authors make an exciting discovery that TET1KO PGCLCs de-differentiate into TE/Amnion-like fates. However, they do not integrate their datasets with in vivo counterparts. It is suggested to utilize the recent pre-print (10.1101/2021.05.07.442980), which includes a data integration tool for examining TE and amnion fates.

Response 9. Please see **General Response 6**.

Another crucial point is that TET1KO PGCLCs already appear significantly different from the controls at c0 and c12, as observed in Figure 5C, with upregulation of genes like HAND1. This raises the question of whether TET1 is involved in PGC specification and if this phenotype becomes stronger with passaging of PGCLCs. Additionally, it seems that the regulation of bivalent genes occurs during specification, not only during expansion. Thus, statements like those in lines 620-623 or 734-736 should be revisited. Similarly, the statement in lines 633-635 is speculative at this point. To challenge the function of TET1 in epigenetic reprogramming, it would be necessary to conduct conditional loss-of-function experiments during the expansion stage. Overall, the final conclusions of this paper should be softened, as it is premature to state that there is no contribution of active DNA demethylation, especially at imprints and germline genes.

Response 10. We would like to thank the Referee for these comments. As we described in the original manuscript, *TET1* KO hiPSCs were induced into BT⁺AG⁺ cells in an apparently normal fashion (Extended Data Fig. 10f); the induced cells showed a robust expansion until c12 (Extended Data Fig. 10g), and were clustered with their wild-type counterparts in their transcriptomes (Extended Data Fig. 10h). On the other hand, as the Referee pointed out, they did show DEGs compared to their wild-type counterparts. We therefore revised the relevant statements to account for this fact (“**TET1 safeguards human germ cells from differentiation into amnion**” section).

Please note that we did show the involvement of TET1 in the DNA demethylation of promoters, CGIs, CTCF binding sites, exons, and CDS (Fig. 6c). Please see **General Response 5** regarding the role of TET1 in the demethylation of ER-gene promoters and imprint DMRs.

Minor comments:

1. *Extended Figure 1g: The color scale is not visible.*

Response 11. We revised the color scale in Extended Data Fig. 1g so that it should now be clear.

2. *Some statistical analyses and quantifications are missing, such as in Fig1b, Fig1h, Fig6f/h.*

Response 12. We performed quantifications and statistical analyses for the data in Fig. 1d, Extended Data Fig. 2h, and 6g, j, and described the results in the corresponding legends in the revised manuscript.

3. *In line 175-177, it should be clarified that at this point, the claim can only be made that BMP is involved in PGCLC expansion/differentiation, and not yet in epigenetic reprogramming. The evidence for epigenetic reprogramming will be presented later.*

Response 13. We consider the up-regulation of DAZL as an index for epigenetic reprogramming. In response to the Referee's comment, we moderated the relevant sentence in the revised manuscript ("**Screening of signaling for hPGCLC differentiation**" section, 3rd paragraph).

4. *In Figure 2c (bottom), the color code is incorrect.*

Response 14. We thank the Referee for pointing this out. We corrected the label in Fig. 2c in the revised manuscript.

5. *In Fig2b, it is unclear which gates were used for the purification of the culture. Did you sort for the AG+ve cells or double positive cells?*

Response 15. We thank the Referee for pointing this out. We collected the entire population of the AG⁺ cells, including AG⁺DT⁺/VT⁺ cells where relevant, for the counting of the cell numbers. We explained this in the legend to Fig. 2b in the revised manuscript.

6. In line 242, there seems to be a mistake in the cells mentioned. Shouldn't it be AG+DT/VT+ cells?

Response 16. We corrected this error in the revised manuscript ("**Transcriptome dynamics**" section, 1st paragraph).

7. In Extended Figure 5a, there appears to be a gene wrongly annotated as an early core ER gene, as it is highly expressed in female iPSCs.

Response 17. We thank the Referee for pointing this out. We corrected the ER gene list and re-analyzed all the relevant data in the revised manuscript (Fig. 6f, h, Extended Data Fig. 9a, b, 10l, 12c, e).

8. In Extended Figure 5e, the expression of Xist from Xi is not visible. Why is this?

Response 18. We thank the Referee for pointing this out. We found that our statement that "while *XIST* (class 3) was expressed from Xi" (line 403 in the original manuscript) was an error. Based on our previous study using cynomolgus monkeys⁴ and many other publications, *XIST* was expressed most likely from Xi, but we did not find informative single nucleotide polymorphisms (SNPs) that discriminate *XIST* expression from parental alleles in regions sequenced with the 3-prime RNA-seq analysis⁷ (Figure 2 for the Referee). We corrected the error and explained this in the legend to Extended Data Fig. 9e in the revised manuscript (Legend to Extended Data Fig. 9e, "**ER gene regulation and X-chromosome reactivation**" section, 2nd paragraph).

9. In Extended Figure 5h, the labeling of the y-axis is confusing. What is the scale on the right? The legend does not explain this graph clearly.

Response 19. The scale in Extended Data Fig. 9h on the left was the log scale, whereas that on the right was the linear scale. We added this information to the revised manuscript (legend to Extended Data Fig. 9h).

10. In Extended Figure 6f, there are significant mRNA levels of TET1 remaining. Therefore, an additional western blot should be performed to demonstrate the loss of the protein using an N-terminus specific antibody. If some protein is still present, it is possible that the authors are observing partial penetrance or a dominant negative effect. This should be discussed.

Response 20. In response to the Referee's comment, we performed Western blot analysis for TET1 in wild-type and *TET1* KO cells using several anti-TET1 N-terminus antibodies, but found that no antibodies detected TET1 in a specific manner. We therefore performed mass spectrometric (MS) analysis for TET1 and its truncated version potentially derived from the *TET1* KO allele. This revealed that peptides from the full-length, but not the truncated, TET1 were reproducibly detected from the wild-type cells, whereas neither form was detected from the *TET1* KO cells (Extended Data Fig. 10e in the revised manuscript). We provided the data and relevant statements in the revised manuscript ("**TET1 safeguards human germ cells from differentiation into amnion**" section, 1st paragraph).

11. The statement in line 673 should be rephrased. PGCLC expansion under standard conditions results in the epigenetic reprogramming of H3K9 and H3K27 methylation, with only DNA methylation showing inefficient reprogramming.

Response 21. We would like to thank the Referee for this comment. In the course of our revision, we decided to remove the sentence in question ("**Discussion**" section).

Referee #3 (Remarks to the Author):

Mammalian primordial germ cells (PGCs) arise during gastrulation, and then migrate to the gonads, whereupon they turn on hallmark markers DAZL and DDX4/VASA; later they undergo sex determination and subsequent steps of germ cell differentiation (PMID 31754036, DOI 10.1007/978-981-10-7941-2_1). Current human pluripotent stem cell (hPSC) differentiation methods yield DAZL-DDX4- PGC-like cells (PGCLCs) that likely correspond to early PGCs prior to their arrival at the gonads (e.g., PMID 26189426, 25543152, 28607482). Consequently, the next major challenge for the field is to understand how to convert them into DAZL+ DDX4+ gonadal-stage PGCs, and the extracellular signals that drive this pivotal developmental transition.

The authors previously showed that hPSC-derived PGCLCs cocultured with ovarian cells turned on DDX4 at extremely low efficiency (less than 2%, after 120 days of culture), within "reconstituted" ovarian cultures (PMID 30237246), thus raising the question of whether it would be possible to produce gonadal-stage PGCLCs without ovarian support cells. Alternatively, the authors previously developed a system to numerically expand hPSC-derived early-stage PGCLCs, but in this system, PGCLCs did not seem to spontaneously mature to a later, gonadal-like state (PMID 32954504).

Here the authors report three important advances. First, starting from hPSC-derived early-stage PGCLCs, they further differentiate them into enriched populations of DAZL+ and DDX4+ PGCLCs that likely correspond to gonadal-stage PGCLCs (Fig. 2). Second, during this differentiation process, the authors achieve massive expansion in cell numbers (>10 billion-fold), thus enabling the mass production of gonad-stage PGCLCs for basic research and other practical applications. Third, during this differentiation process, the hPSC-derived PGCLCs not only turn on later stage markers (e.g., DAZL and DDX4), but they achieve impressive genome-wide DNA demethylation, which represents a key component of the epigenetic reprogramming that occurs in the developing germline. However, there are still two downsides of the authors' system: it still relies on (1) mouse M220 feeder cells and (2) different dosages of BMP across different hPSC lines.

Altogether, the paper is extremely thorough. The authors have not yet arrived at the "holy grail" of efficiently reconstituting later-stage germ cell differentiation (e.g., sex determination and meiosis). Nevertheless, they have developed an important method to produce gonadal-stage PGCLCs in massive numbers, and the entire field will undoubtedly seize upon this method to eventually generate later-stage germ cells.

Response 1. We sincerely thank the Referee for the encouraging comments.

Major comments: the paper is technically comprehensive and in this reviewer's opinion, there are no overt experiments to suggest. This reviewer's thoughts were more focused on the claims and presentation of the manuscript.

1. Main message of the manuscript. This paper reported an impressive advance, but this reviewer thought that the significance could be more clearly conveyed to a broader audience. In the Abstract and Introduction, the authors talk about how they recapitulated "epigenetic reprogramming" in their PGCLCs. This is indeed important, but the significance of "epigenetic reprogramming genes (ER genes)" (e.g., DAZL and DDX4) might be harder for a general audience to follow. Perhaps the authors could consider broadening the significance of their study by explaining in the Introduction (1) the transition from gastrulation- to gonadal-stage PGCs in vivo, (2) that DAZL and DDX4 are markers of the gonadal stage, (3) that they therefore generated DAZL and DDX4 reporters and screened for signals to create gonadal-stage PGCLCs, which represents a major milestone. Explaining and emphasizing the transition from gastrulation- to gonadal-stage PGCLCs could make it easier for a general audience to follow, in addition to the "epigenetic reprogramming" aspect.

Response 2. We thank the Referee for this helpful comment. We believe that epigenetic reprogramming is a hallmark event of general interest during germ-cell development: it resets parental epigenetic memories through genome-wide DNA demethylation and histone modification remodeling and consequently differentiates PGCs ("gastrulation-stage germ cells") into mitotic pro-spermatogonia or oogonia ("gonadal-stage germ cells"). Accordingly, in response to the Referee's comment, we revised the **Abstract** section to explain in a more explicit manner that epigenetic

reprogramming differentiates PGCs (gastrulation-stage germ cells) into mitotic pro-spermatogonia or oogonia (gonadal-stage germ cells) and our work demonstrates an *in vitro* reconstitution of these key processes.

*2. Maturation stage. It is impressive that the authors generate gonadal-stage PGCLCs, but it is important to (1) clarify which exact stage of gonadal maturation they are mimicking and (2) be consistent in their claims (they variously claim to generate pro-oogonia, oogonia, or oocytes in different sections). For instance, the authors write “c107 AG-VT+ cells displayed features closer to that of ag120 AG-VT+ cells in xrOvaries, which represent a retinoic acid (RA)-responsive, pre-leptotene state of the first meiotic prophase” and “emergence of PLL cells during BMP-driven hPGCLC differentiation agrees with the finding that c107 AG-VT+ cells showed advanced characteristics closer to RA-responsive cells (Fig. 3b), suggesting the role of BMP signaling not only in hPGCLC differentiation and mitotic pro-spermatogonia/oogonia propagation, but also in oogonia-to-oocyte differentiation”. After the BMP-driven expansion culture, the authors’ germ cells could potentially correspond to mitotic, early gonadal-stage germ cells (“FGC1” in PMID 28457750), as they are DDX4+ DAZL+ but are negative/low for ZGLP1, STRA8 and SYCP1 (which mark “FGC2/FGC3” in PMID 28457750), as shown in Fig. 3a of the present submission. Given that the cells are negative/low for ZGLP1, STRA8 and SYCP1, would they appear to mimic an earlier stage of differentiation prior to the RA-responsive “FGC2-like” stage? Alternatively, are these markers turned on in a small subset of cells, such that population heterogeneity masks the extent of marker expression? In Fig. 3g, the authors claim that a small subset of their day 117 *in vitro*-derived cells correspond to the “yellow” PLL1 cluster, and it would be nice if the authors could show expression of PLL1 signature markers at the single-cell level to substantiate this claim.*

Response 3. As the Referee pointed out, BMP-driven differentiation induces hPGCLCs into mitotic pro-spermatogonia or oogonia as the major cell type (M1/2/3 cells in Fig. 3c, d), which correspond to mitotic, early gonadal-stage germ cells [FGC1 in ⁸]. On the other hand, we noted that a prolonged culture leads to the differentiation of cells categorized as germ cells at the pre-leptotene stage of the meiotic prophase as a minor cell type [PLL1/2 cells (Fig. 3c, d in the original manuscript)], and these cells show some similarity to RA-responsive cells [FGC2 in ⁸].

Accordingly, in response to the Referee’s comment, we performed a detailed analysis of both M and PLL cells: For M, *in vivo* and *in vitro* cells contributed to each subcluster (M1: G1/S; M2: S/G2/M; M3: G2/M) in an equivalent manner (Fig. 3c, d, Extended Data Fig. 6b, c) and the DEGs between *in vivo* and *in vitro* cells were small in number, with those up-regulated *in vivo* enriched for “positive regulation of double-strand break repair” and those up-regulated *in vitro* enriched for “angiogenesis” and “negative regulation of transcription from RNA polymerase II promoter” (Extended Data Fig. 6d–f, Fig. 3i, Supplementary Table 4). In contrast, for PLL, *in vivo* cells contributed only to PLL1 (S/G2/M/meiotic) and although *in vitro* cells also contributed to PLL1, they were plotted in its periphery adjacent to PLL2, and *in vitro* cells were the exclusive source for PLL2 (Fig. 3c, Extended Data Fig. 6b, c). The PLL1 signature genes included ZGLP1, STRA8, REC8, and SMC1B, and exhibited strong up-regulation in PLL1 *in vivo* cells, but showed only moderate elevation in PLL1 or 2 *in vitro* cells (Extended Data Fig. 6g). We identified DEGs between PLL1 *in vivo* and PLL1

or 2 *in vitro* cells, and found that those up-regulated in PLL1 *in vivo* cells were enriched for “meiotic nuclear division,” whereas those up-regulated in PLL1 or 2 *in vitro* cells were enriched for “regulation of cell differentiation,” “cell development,” and “regulation of signal transduction” (Extended Data Fig. 6d, e, h, Supplementary Table 4). Collectively, these findings indicate that BMP-driven hPGCLC differentiation recapitulates the transcriptome dynamics of hPGC differentiation into mitotic pro-spermatogonia/oogonia [FGC1 in ⁸] and their robust propagation. On the other hand, the continued culture also results in differentiation of aberrant cells as a minor population; these cells show low expression of genes for meiotic entry [up-regulated in FGC2 in ⁸] and ectopic up-regulation of developmental regulators. We added these data and accompanying descriptions to the revised manuscript (“**Transcriptome dynamics**” section, 3rd and 4th paragraph).

3. BMP ligand expression in vivo. It very interesting that BMP seems to be the key signal that matures hPSC-derived PGCLCs to a gonadal stage, as across model organisms, BMP ligands are known to be expressed in the posterior end of the embryo (roughly where the gonads are located). The authors’ analysis of published scRNAseq data shows that gastrulation-stage human embryos express BMP ligands. However, wouldn’t this correspond to the initial BMP input required for PGC specification during gastrulation (e.g., PMID 19410550)? At later developmental stages, are BMP ligands similarly expressed in hindgut endoderm cells or gonadal niche cells? This might be easily answered using published scRNAseq datasets (e.g., PMID 28457750). Additionally, while the authors emphasize the importance of BMP signaling in human PGCLC maturation, there is a considerable body of work on BMP signaling and PGC specification in mouse (as the authors themselves have contributed to, including PMID 19410550). Is there any evidence that in non-human model organisms that (1) BMP is functionally required for later-stage maturation of PGCLCs in the gonads or that (2) BMP ligands are expressed in the gonads? To be clear, this is not a request to perform experiments in non-human model organisms, but this might be worth mentioning in the Discussion (this is mentioned briefly in the second paragraph of the Discussion, but could be elaborated on).

Response 4. We thank the Referee for these comments. As the Referee would agree, the mechanism of PGC specification in humans has been unknown due to an inaccessibility to human embryos at the relevant stage. Using cynomolgus monkeys (cy) as a primate model, we have provided evidence suggesting that cyPGCs are specified in the nascent amnion in response to the BMP signaling originating from the amnion itself as early as embryonic day (E) 11 prior to gastrulation ⁹. We therefore assume that BMP signals in the human endoderm, including the definitive endoderm, yolk sac endoderm, and hypoblast in human embryos at ~E17 (Extended Data Fig. 2a–d) ¹⁰, are not the BMP inputs required for hPGC specification, but would be those that induce epigenetic reprogramming during hPGC migration and hPGC differentiation into mitotic pro-spermatogonia or oogonia. This is in line with the observation that PGCs initiate epigenetic reprogramming during their migratory period and up-regulate genes such as *DAZL* and *DDX4* prior to their colonization of the gonads ^{5,8,9,11-13}. We therefore reason that *BMPs* expressed in the gonads are not the BMP inputs inducing epigenetic reprogramming. In response to the Referee’s comment, we analyzed the expression of *BMPs* in the hindgut endoderm at a later developmental stage (week 6.1) ¹⁴ and found that such cells indeed express *BMPs* (Extended Data Fig. 2e in the revised manuscript). We provided this data and a discussion of these points in the revised manuscript (“**Screening of signaling for hPGCLC differentiation**” section, 2nd paragraph).

4. Overall concision. Overall, parts of the main text and figures could be made more concise. This would greatly help readers to follow the paper. To highlight one example, the section “TET1 safeguards human germ cells from differentiation into amnion” could be shortened to more concisely convey the message that TET1 deletion leads to spurious amniotic differentiation. In this section, paragraphs on trajectory analysis and gene ontology term analysis should be drastically shortened, and this would not really change the overall interpretation of this section. Likewise, various panels of the figures repeatedly show the same thing, including Figs. 1d-g (in vivo expression of BMP ligands in human embryos by scRNAseq), so the authors should weigh what data should be allocated to Figures vs. Extended Data Figures.

Response 5. We thank the Referee for these comments. We have made a substantial revision of the entire manuscript to improve both the clarity and concision to meet with the requests from the Referees and the journal.

Minor comments:

1. Acronyms. The extensive use of acronyms makes the paper hard to follow, and the authors could consider introducing a simple cartoon somewhere in the main figures to introduce the various acronyms (e.g., for differentiation stages, cell lines, and fluorescent reporters). For instance, acronyms for multiple differentiation stages are introduced in Fig. 1 and more acronyms for various cell lines and fluorescent reporters are in Fig. 2, which make the figures difficult to decode.

Response 6. We thank the Referee for these comments. In response to the Referee’s comments, we provided a simple panel introducing the relevant acronyms used in our study in Fig. 1a in the revised manuscript.

2. Comparing the extent of epigenetic reprogramming. The authors show an impressive loss of genome-wide DNA methylation in their later-stage PGCLCs. However, to the best of this reviewer’s knowledge, other hPSC differentiation protocols to create early-stage PGCLCs have also shown some degree of epigenetic changes, although probably more modest. Can the authors discuss, or even directly compare, the extent of epigenetic reprogramming that they achieve against past efforts that made early-stage PGCLCs?

Response 7. We compared the genome-wide 5mC profiles in hPGCLCs reported in other studies^{15,16} with those in hPGCLC-derived cells in this study and discussed the results in the revised manuscript (Extended Data Fig. 8e, “DNA methylome reprogramming” section, 1st paragraph).

3. *Early-stage PGCLC differentiation. Minor comment: what purity of BLIMP1+ and/or TFAP2A+ PGCLCs were generated at day 6 of differentiation? This was mentioned in some (e.g., Fig. 5a) but not all experiments.*

Response 8. We provided information on the hPGCLC induction ratio in all relevant experiments/figures in the revised manuscript (Fig. 1c, Extended Data Fig. 1a, 3f, 10f).

4. *Fig. 1c: Could the authors list somewhere in the Methods section the list of molecules screened, and their concentrations?*

Response 9. We listed the molecules screened and their concentrations in the “**hPGCLC expansion/differentiation culture**” section in the **Methods** in the revised manuscript.

5. *Fig. 1c: In this screening experiment, did the authors assess expression of DAZL and DDX4?*

Response 10. For this experiment, we only assessed the expression of the four genes (*PRDM1*, *GTSF1*, *PRAME*, *MEG3*).

6. *The authors make the interesting discovery that WNT inhibitor (IWR1) supports PGCLC expansion and blocks de-differentiation into pluripotent cells. Could this be because WNT drives naïve pluripotency, so WNT inhibition restricts the re-acquisition of pluripotency?*

Response 11. De-differentiated cells in the hPGCLC culture were transcriptionally similar to hiPSCs in primed pluripotency or iMeLCs¹, and the mechanism by which WNT inhibition blocks hPGCLC differentiation is currently unclear.

7. *Fig. 2a: The authors should label the flow cytometry plots with the percentage of cells present in each population/gate, so these data can be quantified and interpreted more readily.*

Response 12. We provided the percentages of cells in each gate of the flow cytometry plots in all relevant figures in the revised manuscript (Fig. 2a, Extended Data Fig. 1a, 2f, 3g, 4c, e, g, l, n).

8. *Fig. 2a: It is impressive that the authors can upregulate DAZL and VASA. However, it still takes weeks or months in the BMP agonist/WNT inhibitor-based culture system to upregulate these*

markers, perhaps suggesting that there may be other signals important for DAZL and VASA upregulation?

Response 13. We assume that the speed of the epigenetic reprogramming, hPGCLC expansion, and differentiation would depend on the balance between BMP and MAPK/ERK signaling and genetic/epigenetic properties of the original hiPSC line, but as the Referee pointed out, the existence of other relevant signals would not be excluded. We discussed these points in the revised manuscript ("**Discussion**" section, 2nd paragraph).

9. "Morphologically, both AG+DT- and AG+DT+ cells exhibited a spindle shape with a large, ovoid nucleus, indicative of their motile phenotype" – if cell motility is not directly assessed, this claim should be toned down.

Response 14. We deleted the statement in the revised manuscript.

10. Fig. 2c, bottom panel: "AGVT" is used twice in the legend to refer to different things. Is this a typo?

Response 15. We thank the Referee for this comment. We corrected the error in the revised manuscript.

11. Fig. 2e: What are the cells with fewer than 46 chromosomes?

Response 16. We assume that these are cells that have either somewhat lost the sex chromosomes or lost some chromosomes during the spread analysis as an experimental artefact.

12. A previous study (PMID 31754036) highlighted a core set of 13 genes (DAZL, PDLIM1, BAIAP2L1, ZNF800, REC114, MAGEB16, TDRD12, SYCP3, FKBP6, MAEL, DDX4, SMC1B, SLC25A31) that are upregulated upon PGC entry into the gonads, which were conserved across multiple animal species; are these 13 genes likewise turned on by BMP treatment?

Response 17. We provided the data for the expression of the 13 genes and a relevant statement in the revised manuscript (Extended Data Fig. 5a, "**Transcriptome dynamics**" section, 1st paragraph).

13. *“The up-regulated genes included ER genes and were enriched for gene ontology (GO) terms such as “BMP signaling pathway/negative regulation of transcription from RNA polymerase II promoter” (e.g., ID1–4, GATA2/3/6, TBX1/3), and “inactivation of MAPK activity” (e.g., DUSP4, DUSP6) (Extended Data Fig. 3d, e).” – Maybe the authors could clarify by saying “BMP-upregulated genes” (which might be clearer than just “upregulated genes”)*

Response 18. We revised the phrase as suggested in the revised manuscript (“**Transcriptome dynamics**” section, 2nd paragraph).

14. *Fig. 3a: Interestingly, DNMT3L is still repressed even in the absence of BMP.*

Response 19. We commented on this point in the revised manuscript (“**BMP signaling attenuates DNMT activities**” section, 1st paragraph).

15. *Fig. 3c: To make it easier to interpret this figure, the authors should consider directly defining what the acronyms are in the figure itself.*

Response 20. In response to the Referee’s comment, we defined the acronyms directly in the figure panel in the revised manuscript (Fig. 3c).

16. *Fig. 3d: To make it easier to interpret this figure, could the authors explicitly label the top row “in vivo” and the bottom row “in vitro”?*

Response 21. We added these labels as suggested (Fig. 3d).

17. *Fig. 3g: It might be helpful to include the cluster proportion percentage proportions somewhere, especially for key samples like c117.*

Response 22. We included the cluster proportion in Extended Data Fig. 6b, c in the revised manuscript.

18. *Fig. 3h: This figure is somewhat redundant with Fig. 3g, and the authors could consider moving it to the Extended Data figures.*

Response 23. We moved this panel to the Extended Data Fig. 6c in the revised manuscript.

19. *Fig. 5b: Maybe it would be clearer if the authors labeled the y-axis as “Cell number fold change”, which is more specific than “fold change”.*

Response 24. We did so in the revised manuscript (Extended Data Fig. 10g).

20. *“On the other hand, TET1 KO hPGCLC-derived cells could be cultured at least until c42 (Fig. 5b), indicating that TET1 is not necessarily essential for hPGCLC propagation.” – The meaning of this sentence is not clear, as it seems to suggest that TET1 is dispensable, and seems to contradict the preceding sentence.*

Response 25. In the course of revising this section, we deleted the sentence in question (“**TET1 safeguards human germ cells from differentiation into amnion**” section, 1st paragraph).

21. *Fig. 5e – This figure is hard to interpret. Where do the wild-type and TET1-/- cells reside in this plot? Would it be clearer to show two different UMAP plots of wild-type vs. TET1-/- cells to depict expression of amniotic markers (e.g., ISL1) in the latter, but not the former?*

Response 26. We thank the Referee for this comment. We provided additional UMAP plots showing wild-type and TET1 KO cells and expression of key AM/TE markers in the revised manuscript (Fig. 5b).

22. *The authors claim that trophectoderm-like cells can emerge after TET1 deletion (e.g., Fig. 6i). While they show expression of some markers (“SDC1, KRT7 [trophectoderm (TE), TFAP2A, GATA3, and HAND1 [AM/TE markers]”), what about other markers like ELF5 or additional trophectoderm markers (e.g., those reported by PMID 26862703)?*

Response 27. In response to the Referee’s comment, we added information on the expression of additional TE markers reported in ^{17,18}, including ELF5, in the revised manuscript (Fig. 5c).

23. *Fig. 5h – This panel could be moved into the Extended Data Figures, as it does not add much to the identity of the various cell-types.*

Response 28. We moved this panel to Extended Data Fig. 11c in the revised manuscript.

24. Claims about TET1 repressing bivalent promoters are interesting but may need to be toned down slightly, since the authors did not show direct binding of TET1 to those genomic elements using ChIP-seq or another technique.

Response 29. We re-analyzed the TET1 binding sites in hESCs¹⁹ and found that among the DEGs between wild-type and *TET1* KO cells at c12 and c42, up-regulated genes with bivalent promoters were highly enriched with the targets of TET1 in hESCs (Fig. 6j, Extended Data Fig. 12f in the revised manuscript). We adjusted the claims accordingly ("**TET1 represses bivalent promoters and demethylates regulatory elements**" section, 2nd and 4th paragraph).

25. In the Discussion, the authors claim that expansion occurs over ~4 months, but in Fig. 6i they indicate ~5 months: maybe these time durations can be standardized.

Response 30. We thank the Referee for this comment. We deleted these statements in the course of substantially revising and thinning the **Discussion** section.

26. In the Discussion, the authors make the interesting comment that epigenetic reprogramming and PGCLC specification happen contemporaneously in mouse, but are separable in human. As the authors' work focuses on in vitro system, as opposed to making broad claims about human and mouse development, maybe they could highlight that these two processes are separable in an in vitro human system, as it is less clear whether the same claim could be made in vivo.

Response 31. We thank the Referee for this comment. We also deleted these statements in the course of our revision of the **Discussion**.

References

- 1 Murase, Y. *et al.* Long-term expansion with germline potential of human primordial germ cell-like cells in vitro. *EMBO J* **39**, e104929, doi:10.15252/embj.2020104929 (2020).
- 2 Yamashiro, C. *et al.* Generation of human oogonia from induced pluripotent stem cells in vitro. *Science* **362**, 356-360, doi:10.1126/science.aat1674 (2018).
- 3 Vaisvila, R. *et al.* Enzymatic methyl sequencing detects DNA methylation at single-base resolution from picograms of DNA. *Genome Res*, doi:10.1101/gr.266551.120 (2021).
- 4 Gyobu-Motani, S. *et al.* Induction of fetal meiotic oocytes from embryonic stem cells in cynomolgus monkeys. *EMBO J*, e112962, doi:10.15252/embj.2022112962 (2023).
- 5 Tang, W. W. *et al.* A Unique Gene Regulatory Network Resets the Human Germline Epigenome for Development. *Cell* **161**, 1453-1467, doi:10.1016/j.cell.2015.04.053 (2015).
- 6 Zhao, C. *et al.* Reprogrammed blastoids contain amnion-like cells but not trophoblast. *bioRxiv*, doi:10.1101/2021.05.07.442980 (2021).
- 7 Nakamura, T. *et al.* SC3-seq: a method for highly parallel and quantitative measurement of single-cell gene expression. *Nucleic Acids Res* **43**, e60, doi:10.1093/nar/gkv134 (2015).
- 8 Li, L. *et al.* Single-Cell RNA-Seq Analysis Maps Development of Human Germline Cells and Gonadal Niche Interactions. *Cell Stem Cell* **20**, 858-873 e854, doi:10.1016/j.stem.2017.03.007 (2017).
- 9 Sasaki, K. *et al.* The Germ Cell Fate of Cynomolgus Monkeys Is Specified in the Nascent Amnion. *Dev Cell* **39**, 169-185, doi:10.1016/j.devcel.2016.09.007 (2016).
- 10 Tyser, R. C. V. *et al.* Single-cell transcriptomic characterization of a gastrulating human embryo. *Nature* **600**, 285-289, doi:10.1038/s41586-021-04158-y (2021).
- 11 Gkoutela, S. *et al.* DNA Demethylation Dynamics in the Human Prenatal Germline. *Cell* **161**, 1425-1436, doi:10.1016/j.cell.2015.05.012 (2015).
- 12 Guo, F. *et al.* The Transcriptome and DNA Methylome Landscapes of Human Primordial Germ Cells. *Cell* **161**, 1437-1452, doi:10.1016/j.cell.2015.05.015 (2015).
- 13 Alves-Lopes, J. P. *et al.* Specification of human germ cell fate with enhanced progression capability supported by hindgut organoids. *Cell reports* **42**, 111907, doi:10.1016/j.celrep.2022.111907 (2023).
- 14 Elmentaite, R. *et al.* Single-Cell Sequencing of Developing Human Gut Reveals Transcriptional Links to Childhood Crohn's Disease. *Dev Cell* **55**, 771-783 e775, doi:10.1016/j.devcel.2020.11.010 (2020).
- 15 von Meyenn, F. *et al.* Comparative Principles of DNA Methylation Reprogramming during Human and Mouse In Vitro Primordial Germ Cell Specification. *Dev Cell* **39**, 104-115, doi:10.1016/j.devcel.2016.09.015 (2016).
- 16 Kobayashi, M. *et al.* Expanding homogeneous culture of human primordial germ cell-like cells maintaining germline features without serum or feeder layers. *Stem cell reports* **17**, 507-521, doi:10.1016/j.stemcr.2022.01.012 (2022).

- 17 Lee, C. Q. *et al.* What Is Trophoblast? A Combination of Criteria Define Human First-Trimester Trophoblast. *Stem cell reports* **6**, 257-272, doi:10.1016/j.stemcr.2016.01.006 (2016).
- 18 Io, S. *et al.* Capturing human trophoblast development with naive pluripotent stem cells in vitro. *Cell Stem Cell* **28**, 1023-1039 e1013, doi:10.1016/j.stem.2021.03.013 (2021).
- 19 Verma, N. *et al.* TET proteins safeguard bivalent promoters from de novo methylation in human embryonic stem cells. *Nat Genet* **50**, 83-95, doi:10.1038/s41588-017-0002-y (2018).

Reviewer Reports on the First Revision:

Referees' comments:

Referee #1 (Remarks to the Author):

The authors have performed a substantial amount of work to address the concerns raised during the initial review. A major part of the update includes repeating the differentiation process using three additional hiPSC lines. This work now shows that variation between cell lines, driven by the initial cellular state of the iPSC lines, plays a larger role in the differentiation of these cells into more differentiated germ cells than specific reporter transgene integrations or sex chromosome constitution. The work further examines the process of in vitro DNA de-methylation and finds a consistent process largely involving passive de-methylation to be at play. The replication of the key experiments represents a strong enhancement of the results.

The analysis of endogenous marker expression shows good agreement with reporter expression, with the additional finding that expression of low levels of DAZL and DDX4 are likely below the level needed to generate detectable reporter fluorescence. This is a very nice addition to the manuscript.

The improvement in nomenclature and labelling of the figures has made them much easier to understand. Additionally, the re-writing of the text has greatly improved the clarity and readability. Four textual changes outlined below remain to be considered. Other than that, I feel that the authors have satisfied the primary concerns of the reviewers and the paper is now fit for publication.

1. Figure 6: Can the authors comment on the apparent conundrum between gain of DNA methylation and gain in expression at bivalently marked genes in TET1 KO cells? It is advised to point this out in the paper.

2. Line 336 – 338 (related to extended data figure 9d): “Thus, the expression of ER genes is regulated in a coordinated manner relative to the extent of their promoter 5mC demethylation”. This statement appears to be too bold since it is only partially supported by the data. The change in DNA methylation is for most genes relatively small (less than 10%, as shown in the figure) and does only relate positively with a gain in expression for a few genes. What is the correlation coefficient for the data in the four plots? The reason for this rather modest relationship may be due to that the measurements were done in bulk populations of cells, not accommodating single cell heterogeneity for both molecular read-outs.

3. Line 386 – 397 (related to extended data figure 9l and 9m): in extended data figure 9l, the $\Delta\text{DNAm}/\text{cell division}$ parameter corresponds to a demethylation rate over developmental time. If understood correctly, the data in extended data figure 9m represents for different tiles the change in DNAm% between cell type “A” versus cell type “B” relative to the % of DNAm present within the original cell type A. Hence, this data represents a kind of normalized degree of DNA demethylation, that mathematically speaking is not related to development time, and hence does not represent a demethylation rate. Why are the fold-changes presented in extended data figure 9m negative in value? Should these not be positive values?

The concluding sentence of this paragraph is very speculative (“with a bias towards genomic tiles with less affinity/recruitment potential for residual DNMT activities”). There may be other reasons underlying protection of certain regions against demethylation or maintenance of residual methylation. It is recommended to be either more comprehensive, or alternatively, clearer in what is meant.

Finally, are the regions resistant to DNA demethylation *in vitro* also more prone in keeping DNA methylation *in vivo*? Is there any sequence composition underlying such regions? Or a chromatin state?

Referee #2 (Remarks to the Author):

The revised manuscript by Murase et al. represents a significant improvement over the initial version. The authors have adequately addressed the points I raised and provided additional analysis. The only outstanding concern for me pertains to the section discussing TET1KO differentiation into TE and amnion. The section is titled "TET1 safeguards human germ cells from differentiation into amnion." However, upon integration with reference datasets, it becomes evident that TET1KO cells not only lead to amnion but also exhibit a TE-like state and robust induction of extraembryonic mesoderm. In fact, it appears that only a small proportion of cells become similar to the amnion. Therefore, I recommend revising this interpretation throughout the paper. The authors could assert that TET1 safeguards human germ cells from differentiation into extraembryonic cell types. I find the current interpretation of the data to be misleading. With these text changes, I believe that the manuscript will be ready for publication.

Referee #3 (Remarks to the Author):

In this revised manuscript, the authors have addressed some of the reviewers' comments, although some questions remain, especially regarding the mechanistic work. Despite these remaining issues, it should still be emphasized that the current manuscript reports significant advances compared to past work in the field, including (1) generation of a considerably higher percentage (>80%) of DDX4/VASA+ TFAP2C+ or DAZL+ TFAP2C+ germ cells (Fig. 2A) and (2) massive expansion of hPSC-derived PGCLCs (up to 10 billion-fold; Fig. 2B). By comparison, past work differentiated hPSCs into DDX4+ or VASA+ germ cells with several percent efficiency, and relied on coculture with ovarian or hindgut cells (PMID 30237246, 36640324). In another important advance, the present manuscript describes that some hPSC-derived germ cells progress further and reach more developmentally advanced stages marked by SYCP3 and ZGLP1 (Extended Data Fig. 6A), which is quite impressive. They are therefore significantly moving the field forward with regard to the quantity and maturity of human germ cells that can be generated *in vitro* from hPSCs.

In their revised study, the authors now report several new observations that strengthen their case. First, they demonstrate that, across several independent hPSC lines, they can achieve significant

PGCLC expansion (Fig. 2B) and reproducible induction of DAZL and VASA (Fig. 3A). Second, they show that BMP ligands are putatively expressed at the right time and place within the human hindgut endoderm *in vivo* to mediate the PGCLC expansion and maturation effects observed *in vitro* (Extended Data Fig. 2E).

While altogether the study describes a useful method to generate hPSC-derived DDX4+ and VASA+ germ cells in massive quantities, several limitations should also be noted. First, the authors' differentiation system relies on coculture with mouse m220 feeder cells. Although the authors critique that the hindgut coculture system "introduces heterogeneity" (line 83), they themselves are also introducing heterogeneity through coculture with non-human feeder cells. (Additionally, serum is used in most experiments, although the authors have shown that it is dispensable; Extended Data Fig. 4Q). Second, as other reviewers have pointed out and as the authors appropriately acknowledge, there is still some line-to-line variability in PGCLC expansion and proliferation rates (Fig. 2B). Of the five hPSC lines or subclones examined: long-term expansion is demonstrated for two lines/subclones, whereas shorter-term expansion is reported for three lines/subclones, and one of these three (F2) shows drastically less expansion capacity (>1000-fold less expansion; Fig. 2B). These limitations do not necessarily require new experiments to address, but should be noted.

Finally, and most significantly, several aspects of the mechanistic work regarding BMP and TET1 are less clear-cut (see Major Comments below). While the authors disclose an impressive functional effect of BMP on PGCLC expansion and maturation, there is less evidence to support the mechanistic claims that BMP represses DNA methylation machinery or suppresses MAPK/ERK signaling. Toning down the mechanistic claims or even removing some of the less definitive mechanistic data could strengthen the work. Taken together, this study reports a useful advance, and if the authors could address several important remaining issues, they could even further strengthen their manuscript.

Major comments:

1. BMP mechanism: effects on DNA methylation. First, the authors should provide the absolute fold-change differences with regard to BMP-induced DNMT3A, DNMT3B, and UHRF1 downregulation. The heatmap in Fig. 3A suggests modest changes in DNMT3A, DNMT3B, and UHRF1 expression upon BMP treatment. It also seems like it takes weeks for the expression decreases to manifest, suggesting that these are not direct target genes of BMP-mediated repression. Second, the authors claim that BMP treatment leads to cytoplasmic translocation of UHRF1 protein, but the images do not strongly support this conclusion. Rather, Extended Data Fig. 9J gives the impression that UHRF1 levels are strongly decreased, and there is still some nuclear localization of the residual UHRF1 protein. To examine cytoplasmic translocation, the authors could consider biochemical fractionation of cells to separate cytoplasmic vs. nuclear fractions, followed by a Western blot for UHRF1. Addressing these points is important, since the authors claim in the Abstract that "BMP signaling attenuates ... *de novo* and maintenance DNMT activities". If necessary, this claim might be toned down or removed from the Abstract.

2. BMP mechanism: MAPK/ERK suppression. "Accordingly, Western blot analysis revealed that BMP signaling reduced the phosphorylated MAPK/ERK levels in hPGCLC-derived cells (Extended Data Fig. 5h, i)." – The Western Blot reveals a modest effect, even if it is statistically significant. Additionally, it

is unclear if this result reflects direct or indirect pathway crosstalk. The authors may want to consider toning down this claim and removing it from the Abstract.

3. TET1 mechanism: There are several questions about this section. First, the authors show that promoters display elevated 5-mC in TET1^{-/-} cells, but nevertheless become more transcriptionally active. This is quite a surprising result, as 5-mC at promoters is generally thought to enforce gene silencing. Can the authors explain this apparent contradiction? Or could this be an experimental artifact derived from bulk-population analyses where individual subpopulations harbor sharply different transcriptional or chromatin states? The authors could also consider omitting these data. Second, the authors could consider toning down their claims about TET1 and DNA demethylation, since they primarily examine 5-mC but not 5-hmC (the latter of which is the direct enzymatic product of 5-mC). Third, the authors could also consider including more in the Discussion section about what is already known about TET1/TET2 and DNA methylation/demethylation in the germline (e.g., PMID 29513657, 23415914, 37456839, 23223451), and how it compares to their present findings.

4. Writing. Some parts of the manuscript, such as the Introduction, are clearly written. However, several other sections of the manuscript, including the sections “DNA methylome reprogramming”, “ER gene regulation and X-chromosome reactivation”, and “TET1 represses bivalent promoters and demethylates regulatory elements” are dense and data-laden, making them difficult to follow, especially for Nature’s general readership. For the “DNA methylome reprogramming” and “ER gene regulation and X-chromosome reactivation” sections of the Results, it would be helpful if the authors could consider including 1-2 sentences of background regarding X chromosome activation/inactivation and DNA methylation dynamics present in vivo before jumping into their in vitro results, to signal to the reader what is the in vivo gold standard that they are trying to recapitulate. Generally speaking, it would be helpful if the authors could consider making their entire manuscript easier to read.

Minor comments:

1. Cell-type terminology: Multiple reviewers have asked about how the authors define “mitotic prospermatogonia or oogonia”, which are the crux of the authors’ claims. To help a general audience understand the various cell differentiation stages involved in this work (of which there are many, see “staircase plots” in Fig. 3A and Extended Data Fig. 6A), the authors could consider including in the main or supplementary figures a cartoon explaining (1) the various differentiation stages, (2) markers used to define each stage, and (3) which stage(s) are encompassed by their new differentiation system. There are many differentiation markers mentioned throughout this study, and a simple cartoon would help a general audience grasp the significance of each marker.

2. “Here, we embarked on establishing a system for hPGCLC differentiation under a defined condition” (Introduction): The authors should consider replacing the word “defined” with another word. Most of their experiments include FBS. Additionally, all their experiments entail Advanced DMEM/F12, which usually includes bovine serum albumin, which is not defined and often contains many impurities (PMID 28238792).

3. “Screening of signaling for hPGCLC differentiation” section (Results): Halfway through this paragraph, non-human cells are mentioned abruptly, which might be a bit hard for a general audience to follow. The authors should more directly explain in the beginning of the paragraph that mouse m220 feeder cells are employed as part of the culture system, as this is a potential limitation

of the current method that should be clearly stated.

4. Fig. 2F: In the figure legend, please kindly explain the statistical test that was performed.

5. Cell expansion terminology: Throughout the Results section, the authors mention the degree of cell expansion observed in various conditions. Do they refer to total cell number, or alternatively, the number of marker-defined germ cells that expanded? Both are fine, but clarification would be helpful.

6. Extended Data Fig. 2G: The authors should consider labeling the y-axis (“Fold change” and “Enrichment score”) with more details, to help the reader understand what is being measured.

7. “In response to BMP2, hPGCLC-derived cells maintained pluripotency/early PGC genes, and up-regulated ER genes and 13 previously reported genes that show up-regulation in gonadal germ cells 26” (Results): Extended Data Fig. 5A seems to show that only some of the signature genes reported by ref. 26 are upregulated, with genes like REC114 and SLC25A31 not being significantly expressed. If so, perhaps the authors should amend their comment to state “some of the 13 previously reported genes”?

8. Extended Data Fig. 9J: What GFP is being used, is GFP supposed to be nuclearly-localized?

9. Lines 386-397 of the main text, which talk about 5-mC and genomic titles, could be written more clearly.

10. Fig. 5B, Fig. 5D, Extended Data Fig. 11E, and Extended Data Fig. 11H: What samples (e.g., what conditions and genotypes or even datasets from what paper) are being shown in these UMAP plots should be more clearly annotated. At present, these figures are quite hard to follow

11. “Among the DEGs at c42, the up-regulated genes with bivalent promoters were large in number and highly enriched with the targets of TET1 in hESCs (Extended Data Fig. 12f)” – Are TET1 target genes in hESCs relevant to genomics analyses of PGCLCs? This is not an experimental request for the authors to perform TET1 ChIP-seq in PGCLCs, but merits a question nevertheless.

12. Extended Data Fig. 6A: This is quite an impressive figure showing reasonable correspondence between germ cells in vivo and in vitro, at least with regard to the markers shown. However, some important genes like ZGLP1 seem to be expressed significantly less in vitro than in vivo, but the heatmap color scale makes it difficult to ascertain the absolute difference in expression level. How differently are ZGLP1 and other genes expressed in vivo and in vitro? And is there a way to graphically display that?

13. “Fig. 3g: It might be helpful to include the cluster proportion percentage proportions somewhere, especially for key samples like c117. Response 22. We included the cluster proportion in Extended Data Fig. 6b, c in the revised manuscript” (Response to Reviewers): For certain key figures, including Extended Data Fig. 6B, readers may want to know the actual percentage numbers (not just a stacked histogram) of what percentage of the total population was comprised of a specific cell-type. This is especially important for data like Extended Data Fig. 6A,B wherein the authors report the generation of hitherto-difficult-to-produce developmentally advanced cells. Readers will want to know how efficient the process was.

14. “We compared the genome-wide 5mC profiles in hPGCLCs reported in other studies 15,16 with those in hPGCLC-derived cells in this study and discussed the results in the revised manuscript” (Extended Data Fig. 8e, “DNA methylome reprogramming” section, 1st paragraph): This plot was difficult to follow. How do the cells produced by other differentiation protocols compare with the gold standard of actual germ cells? What about the total genome-wide level of methylation, expressed in a percentage, not just a correlation plot? Do the other protocols yield cells with global DNA demethylation?

15. “hPGCLC-derived cells repressed de novo DNA methyltransferases (DNMTs) (DNMT3L is repressed even without BMP)” (Results): The authors could clarify what “repress” means, does it reflect downregulation, or alternatively, repression of enzymatic activity?
16. “We also analyzed the characteristics of DNA demethylation dynamics across genomic tiles with different 5mC levels” (Results): The paragraph starting with this topic sentence was difficult to follow, and should be rewritten.
17. BMP: It is also very interesting that the authors have discovered an important role for BMP signaling in driving epigenetic reprogramming in the germline. This may be somewhat reminiscent of the requirement for BMP to maintain naïve mouse pluripotent stem cells (PMID 14636556). As the authors are well aware, naïve pluripotent cells have a unique chromatin landscape that bears some similarities to PGCs. Maybe this point about BMP and naïve pluripotency would also be worth mentioning, if there is space in the manuscript?
18. Precise maturation stage: It takes weeks to months for the authors’ BMP-driven system to induce expression of DDX4/VASA or DAZL (Fig. 2). What is the identity of the cells prior to DDX4/VASA or DAZL expression? Do their cells express CDH5/VE-CADHERIN and DMRT1, which were previously reported to mark migratory PGCs prior to their lodgment in the gonads (PMID 37709822)?
19. For a general audience, it might be helpful for the authors to compare their present results to those reported by another recent study (PMID 37709822), the latter of which seemed to generate less mature PGCLCs.

Author Rebuttals to First Revision:

Rebuttal: Nature manuscript 2023-15-08623A

We would like to sincerely thank the Referees for their constructive comments, which we have used as the basis for revising our manuscript.

Referees' comments:

Referee #1 (Remarks to the Author):

The authors have performed a substantial amount of work to address the concerns raised during the initial review. A major part of the update includes repeating the differentiation process using three additional hiPSC lines. This work now shows that variation between cell lines, driven by the initial cellular state of the iPSC lines, plays a larger role in the differentiation of these cells into more differentiated germ cells than specific reporter transgene integrations or sex chromosome constitution. The work further examines the process of in vitro DNA de-methylation and finds a consistent process largely involving passive de-methylation to be at play. The replication of the key experiments represents a strong enhancement of the results.

The analysis of endogenous marker expression shows good agreement with reporter expression, with the additional finding that expression of low levels of DAZL and DDX4 are likely below the level needed to generate detectable reporter fluorescence. This is a very nice addition to the manuscript.

The improvement in nomenclature and labelling of the figures has made them much easier to understand. Additionally, the re-writing of the text has greatly improved the clarity and readability. Four textual changes outlined below remain to be considered. Other than that, I feel that the authors have satisfied the primary concerns of the reviewers and the paper is now fit for publication.

Response 1. We sincerely thank Referee #1 for the encouraging comments.

1. Figure 6: Can the authors comment on the apparent conundrum between gain of DNA methylation and gain in expression at bivalently marked genes in TET1 KO cells? It is advised to point this out in the paper.

Response 2. Previous studies in other contexts, including mouse ESCs, have shown that *TET* deficiency leads to hyper-methylation of bivalent promoters with significant up-regulation of transcription, which is most likely due to impaired PRC recruitment¹⁻³. It is therefore conceivable that a similar mechanism operates in *TET1*-deficient hPGCLC-derived cells. We added passages to clarify this possibility (the "***TET1* deficiency leads to de-repression of bivalent genes and**

hypermethylation of regulatory elements” section, 2nd paragraph; and **Discussion** section, 1st paragraph).

2. Line 336 – 338 (related to extended data figure 9d): “Thus, the expression of ER genes is regulated in a coordinated manner relative to the extent of their promoter 5mC demethylation”. This statement appears to be too bold since it is only partially supported by the data. The change in DNA methylation is for most genes relatively small (less than 10%, as shown in the figure) and does only relate positively with a gain in expression for a few genes. What is the correlation coefficient for the data in the four plots? The reason for this rather modest relationship may be due to that the measurements were done in bulk populations of cells, not accommodating single cell heterogeneity for both molecular read-outs.

Response 3. We would like to thank the Referee for this insightful comment. In our revision, we added the correlation coefficients for the data in the four plots and moderated the statement as follows: “Thus, ER-gene expression may occur, at least in part, in a coordinated manner in response to 5mC demethylation of the ER-gene promoter, though single-cell analysis would be necessary to come to any definitive conclusion,” (Extended Data Fig. 9d and its legend; and the “**ER gene regulation and X-chromosome reactivation**” section, 2nd paragraph). We sincerely hope that these revisions meet with the Referee’s approval.

3. Line 386 – 397 (related to extended data figure 9l and 9m): in extended data figure 9l, the $\Delta\text{DNAm}/\text{cell division}$ parameter corresponds to a demethylation rate over developmental time. If understood correctly, the data in extended data figure 9m represents for different tiles the change in DNAm% between cell type “A” versus cell type “B” relative to the % of DNAm present within the original cell type A. Hence, this data represents a kind of normalized degree of DNA demethylation, that mathematically speaking is not related to development time, and hence does not represent a demethylation rate. Why are the fold-changes presented in extended data figure 9m negative in value? Should these not be positive values?

Response 4. Again, we would like to sincerely thank the Referee for this comment. The Referee’s understanding is correct: the data represent a normalized DNA demethylation ratio [% mC original cells – % mC cultured cells/% mC original cells] and should be positive in value. We corrected Extended Data Fig. 9m and its legend in the revised manuscript accordingly.

The concluding sentence of this paragraph is very speculative (“with a bias towards genomic tiles with less affinity/recruitment potential for residual DNMT activities”). There may be other reasons underlying protection of certain regions against demethylation or maintenance of residual methylation. It is recommended to be either more comprehensive, or alternatively, clearer in what is meant.

Response 5. We thank the Referee for the comment. To address this point, we amended the concluding sentence to read as follows: “Taken together, these findings suggest that BMP-driven hPGCLC specification and differentiation are accompanied by an attenuation of both *de novo* and maintenance DNMT activities, promoting a replication-coupled passive genome-wide DNA demethylation, which occurs in a heterogeneous manner depending on the properties of the genomic regions.” (the “**BMP signaling and DNMT activities**” section, 2nd paragraph).

Finally, are the regions resistant to DNA demethylation in vitro also more prone in keeping DNA methylation in vivo? Is there any sequence composition underlying such regions? Or a chromatin state?

Response 6. The genomic bins showing DNA demethylation resistance corresponded to DNA demethylation “escapees” (Extended Data Fig. 9n), which show a great overlap between *in vitro* and *in vivo* cell types and consisted primarily of evolutionarily young retrotransposons (Fig. 4c–e). We have added this information in the revised manuscript (Extended Data Fig. 9n; and the “**BMP signaling and DNMT activities**” section, 2nd paragraph).

Referee #2 (Remarks to the Author):

The revised manuscript by Murase et al. represents a significant improvement over the initial version. The authors have adequately addressed the points I raised and provided additional analysis. The only outstanding concern for me pertains to the section discussing TET1KO differentiation into TE and amnion. The section is titled “TET1 safeguards human germ cells from differentiation into amnion.” However, upon integration with reference datasets, it becomes evident that TET1KO cells not only lead to amnion but also exhibit a TE-like state and robust induction of extraembryonic mesoderm. In fact, it appears that only a small proportion of cells become similar to the amnion. Therefore, I recommend revising this interpretation throughout the paper. The authors could assert that TET1 safeguards human germ cells from differentiation into extraembryonic cell types. I find the current interpretation of the data to be misleading. With these text changes, I believe that the manuscript will be ready for publication.

Response 1. We sincerely thank the Referee for the encouraging comments and for pointing this out. To address this concern, we replaced “amnion” with “extraembryonic cells/cell types” where relevant throughout the manuscript (the **Summary**; the “**TET1-deficient hPGCLCs aberrantly differentiate into extraembryonic cells**” section, 3rd paragraph; the “**TET1 deficiency leads to de-repression of bivalent genes and hypermethylation of regulatory elements**” section, 5th paragraph; and the “**Discussion**” section, 2nd paragraph).

Referee #3 (Remarks to the Author):

In this revised manuscript, the authors have addressed some of the reviewers' comments, although some questions remain, especially regarding the mechanistic work. Despite these remaining issues, it should still be emphasized that the current manuscript reports significant advances compared to past work in the field, including (1) generation of a considerably higher percentage (>80%) of DDX4/VASA+ TFAP2C+ or DAZL+ TFAP2C+ germ cells (Fig. 2A) and (2) massive expansion of hPSC-derived PGCLCs (up to 10 billion-fold; Fig. 2B). By comparison, past work differentiated hPSCs into DDX4+ or VASA+ germ cells with several percent efficiency, and relied on coculture with ovarian or hindgut cells (PMID 30237246, 36640324). In another important advance, the present manuscript describes that some hPSC-derived germ cells progress further and reach more developmentally advanced stages marked by SYCP3 and ZGLP1 (Extended Data Fig. 6A), which is quite impressive. They are therefore significantly moving the field forward with regard to the quantity and maturity of human germ cells that can be generated in vitro from hPSCs.

In their revised study, the authors now report several new observations that strengthen their case. First, they demonstrate that, across several independent hPSC lines, they can achieve significant PGCLC expansion (Fig. 2B) and reproducible induction of DAZL and VASA (Fig. 3A). Second, they show that BMP ligands are putatively expressed at the right time and place within the human hindgut endoderm in vivo to mediate the PGCLC expansion and maturation effects observed in vitro (Extended Data Fig. 2E).

While altogether the study describes a useful method to generate hPSC-derived DDX4+ and VASA+ germ cells in massive quantities, several limitations should also be noted. First, the authors' differentiation system relies on coculture with mouse m220 feeder cells. Although the authors critique that the hindgut coculture system "introduces heterogeneity" (line 83), they themselves are also introducing heterogeneity through coculture with non-human feeder cells. (Additionally, serum is used in most experiments, although the authors have shown that it is dispensable; Extended Data Fig. 4Q). Second, as other reviewers have pointed out and as the authors appropriately acknowledge, there is still some line-to-line variability in PGCLC expansion and proliferation rates (Fig. 2B). Of the five hPSC lines or subclones examined: long-term expansion is demonstrated for two lines/subclones, whereas shorter-term expansion is reported for three lines/subclones, and one of these three (F2) shows drastically less expansion capacity (>1000-fold less expansion; Fig. 2B). These limitations do not necessarily require new experiments to address, but should be noted.

Finally, and most significantly, several aspects of the mechanistic work regarding BMP and TET1 are less clear-cut (see Major Comments below). While the authors disclose an impressive functional effect of BMP on PGCLC expansion and maturation, there is less evidence to support the mechanistic claims that BMP represses DNA methylation machinery or suppresses MAPK/ERK signaling. Toning down the mechanistic claims or even removing some of the less definitive mechanistic data could strengthen the work. Taken together, this study reports a useful advance, and if the authors could address several important remaining issues, they could even further strengthen their manuscript.

Response 1. We sincerely thank Referee #3 for the encouraging comments and for reviewing the manuscript carefully. We agree with the Referee that our study, by demonstrating *in vitro* reconstitution of epigenetic reprogramming and the generation of abundant mitotic pro-spermatogonia and oogonia-like cells in humans, represents a significant advance in human *in vitro* gametogenesis research and its potential translation into reproductive medicine. We also feel that our study provides a key insight into the mechanism of epigenetic reprogramming. On the other hand, we appreciate the concerns raised by Referee #3 regarding some of the statements in our manuscript. To address these concerns, we have provided additional data to further support the statements in question, while also toning down the mechanistic claims where appropriate, as detailed below in our responses to the specific comments. We sincerely hope that our revision meets with approval by the Referee.

Major comments:

1. *BMP mechanism: effects on DNA methylation.* First, the authors should provide the absolute fold-change differences with regard to BMP-induced DNMT3A, DNMT3B, and UHRF1 downregulation. The heatmap in Fig. 3A suggests modest changes in DNMT3A, DNMT3B, and UHRF1 expression upon BMP treatment. It also seems like it takes weeks for the expression decreases to manifest, suggesting that these are not direct target genes of BMP-mediated repression. Second, the authors claim that BMP treatment leads to cytoplasmic translocation of UHRF1 protein, but the images do not strongly support this conclusion. Rather, Extended Data Fig. 9J gives the impression that UHRF1 levels are strongly decreased, and there is still some nuclear localization of the residual UHRF1 protein. To examine cytoplasmic translocation, the authors could consider biochemical fractionation of cells to separate cytoplasmic vs. nuclear fractions, followed by a Western blot for UHRF1. Addressing these points is important, since the authors claim in the Abstract that “BMP signaling attenuates ... *de novo* and maintenance DNMT activities”. If necessary, this claim might be toned down or removed from the Abstract.

Response 2. We would like to thank the Referee for pointing these out. With regard to the first point raised, we have now provided the absolute fold-change differences of the expression of DNMT3A, DNMT3B, and UHRF1 during BMP-driven hPGCLC specification and differentiation (Extended Data Fig. 9i). This shows that the three genes were down-regulated upon BMP-driven hPGCLC specification, and thereafter remained low and appeared to show further down-regulation during BMP-driven hPGCLC differentiation (the differences between c32 cells cultured with or without BMP2 were minor, if any). We apologize for the brevity of the statement “hPGCLC-derived cells repressed *de novo* DNA methyltransferases (DNMTs)” (line 370 in the previous version). This may have been too condensed to adequately describe our observations.

With regard to the second point, we have now provided magnified immunofluorescence (IF) images of UHRF1 and a line-plot analysis of such data for the levels of UHRF1 in the nucleus and the cytoplasm from 10 randomly chosen cells during hPGCLC culture with or without BMP2. This analysis shows that without BMP2, the UHRF1 signal was highly enriched in the nucleus, whereas with BMP2, the signal exhibited a more uniform distribution between the nucleus and the cytoplasm

(Extended Data Fig. 9k, left bottom, right top). This is in good agreement with the analysis of all the cells originally shown in Extended Data Fig. 9k (Extended Data Fig. 9k, right bottom, in the revised manuscript), which showed that the nuclear : cytoplasmic ratio of UHRF1 was significantly lower in hPGCLCs cultured with BMP2 ($p < 5 \times 10^{-6}$), and was even further reduced upon further hPGCLC culture with BMP2 ($p < 5 \times 10^{-6}$). Here again, we apologize that our statement “in response to BMP2, UHRF1 was translocated from the nucleus to the cytoplasm (line 376-377 in the previous version)” might have been a bit too brief to properly describe our observations.

Taken together, our data indicate that in hPGCLCs cultured with BMP2, the expression levels of *DNMT3A*, *DNMT3B*, and *UHRF1* remain low and appear to decrease even further (Extended Data Fig. 9i), the *de novo* DNMT activities represented by genome-wide CpA methylation levels decrease [Effect sizes (Cohen’s d-values) > 0.2 : Extended Data Fig. 9j], and the subcellular localization of UHRF1 changes from a predominately nuclear to a more cytoplasmic localization (Extended Data Fig. 9k).

On the other hand, considering that these processes occur over weeks and the presented data do not address whether this is a direct effect of BMP signaling, which the Referee correctly points out, we have modified our claims and toned them down to reflect our results more accurately (in the **Summary**: “...BMP-driven hPGCLC differentiation is accompanied by an attenuation of ... both *de novo* and maintenance DNA methyltransferase (DNMT) activities,...”; the “**BMP signaling and DNMT activities**” section; and the “**Discussion**” section, 1st paragraph: “...hPGCLCs cultured with BMP2 propagate stably, displaying reduced levels of ... both *de novo* and maintenance DNMT activities/machineries...,” “The precise mechanism of action of BMP signalling ... remains unclear and warrants further investigation”). We sincerely hope that these revisions meet with the Referee’s approval.

2. BMP mechanism: MAPK/ERK suppression. “Accordingly, Western blot analysis revealed that BMP signaling reduced the phosphorylated MAPK/ERK levels in hPGCLC-derived cells (Extended Data Fig. 5h, i).” – The Western Blot reveals a modest effect, even if it is statistically significant. Additionally, it is unclear if this result reflects direct or indirect pathway crosstalk. The authors may want to consider toning down this claim and removing it from the Abstract.

Response 3. We would like the Referee to note that the quantification of the Western blot signals for pERK1 and pERK2 revealed their ~4.5-fold and ~2.9-fold reduction, respectively ($p < 0.02$ and $p < 0.05$, respectively), in hPGCLCs cultured with BMP2 (Extended Data Fig. 5i, j). We performed the same experiments and obtained similar results: ~4.3-fold and ~3.9-fold reduction of pERK1 and pERK2 signals, respectively ($p < 0.01$ and $p < 0.03$, respectively), in hPGCLCs cultured with BMP2 (Extended Data Fig. 5i, j). Combined with the RNA-seq data indicating the negative regulation of the ERK1 and ERK2 pathways at the mRNA levels (Extended Data Fig. 5g, h), these findings provide strong supporting evidence that the ERK signaling is attenuated in hPGCLCs cultured with BMP2.

However, as the Referee pointed out, the presented data do not address whether this is a direct effect of BMP signaling. Therefore, as suggested by the Referee, we have toned this statement down and revised the phrasing to more appropriately reflect our results [**Summary**: "...BMP-driven hPGCLC differentiation is accompanied by an attenuation of the mitogen-activated protein kinase/extracellular-regulated kinase (MAPK/ERK) pathway"; "**Discussion**" section, 1st paragraph: "...hPGCLCs cultured with BMP2 propagate stably, displaying reduced levels of MAPK/ERK signalling...", "The precise mechanism of action of BMP signalling ... remains unclear and warrants further investigation"].

3. TET1 mechanism: There are several questions about this section. First, the authors show that promoters display elevated 5-mC in TET1-/- cells, but nevertheless become more transcriptionally active. This is quite a surprising result, as 5-mC at promoters is generally thought to enforce gene silencing. Can the authors explain this apparent contradiction? Or could this be an experimental artifact derived from bulk-population analyses where individual subpopulations harbor sharply different transcriptional or chromatin states? The authors could also consider omitting these data. Second, the authors could consider toning down their claims about TET1 and DNA demethylation, since they primarily examine 5-mC but not 5-hmC (the latter of which is the direct enzymatic product of 5-hmC). Third, the authors could also consider including more in the Discussion section about what is already known about TET1/TET2 and DNA methylation/demethylation in the germline (e.g., PMID 29513657, 23415914, 37456839, 23223451), and how it compares to their present findings.

Response 4. With regard to the first point, previous studies in other contexts, including mouse ESCs, have shown that *TET* deficiency leads to hyper-methylation of bivalent promoters with significant up-regulation of transcription, which is most likely due to impaired PRC recruitment¹⁻³. It is therefore conceivable that a similar mechanism operates in *TET1*-deficient hPGCLC-derived cells. We have added relevant discussion describing these possibilities [the "***TET1* deficiency leads to de-repression of bivalent genes and hypermethylation of regulatory elements**" section, 2nd paragraph: "The hyper-methylation of bivalent promoters associated with significant up-regulation of transcription might have been due to impaired recruitment of Polycomb repressive complexes (PRCs), as previously shown in mouse embryonic stem cells (mESCs)^{48, 49}"].

With regard to the second point, it is generally well accepted that TET proteins are involved in DNA demethylation through oxidation of 5mC to generate 5hmC or further oxidized species that can then be either passively or enzymatically removed from DNA⁴⁻⁶. On the other hand, we appreciate the Referee's point that we did not directly examine 5hmC, except in wild-type and *TET1 KO* hiPSCs with a dot blot analysis (Extended Data Fig. 10c). Accordingly, in light of the Referee's suggestion, we have provided more appropriate statements explaining our experimental results where relevant in the revised manuscript [**Summary**: "...hPGCLCs deficient in tens-eleven translocation (TET) 1, an active DNA demethylase abundant in human germ cells^{2,3}, differentiate into extraembryonic cells, including amnion, with de-repression of key genes bearing bivalent promoters; these cells fail to fully activate genes vital for spermatogenesis and oogenesis, with their promoters remaining

methyated”; the “**TET1 deficiency leads to de-repression of bivalent genes and hypermethylation of regulatory elements**” section; and the “**Discussion**” section, 2nd paragraph].

With regard to the third point, the manuscript by Hill et al. (PMID 29513657) ⁷ reports findings encompassing those reported by Hackett et al. (PMID 23223451) ⁸ and Vincent et al. (PMID 23415914) ⁹, and the consensus view of the role of *Tet1* in mice is that it is dispensable for genome-wide DNA demethylation *per se*, but is critical in maintaining demethylation of key genes for spermatogenesis and oogenesis and of imprint DMRs ^{7,10,11}. In response to the Referee’s comments, we have therefore added a concise discussion regarding this point and the relevance of our findings on the role of *TET1* in hPGCLC differentiation (the “**Discussion**” section, 2nd paragraph).

On the other hand, the manuscript by Li et al. (PMID 36333732) ¹² reports that *TET1* knockout hiPSCs are induced into hPGCLCs in an apparently normal manner, while the manuscript by Hsu et al. (PMID 37456839) ¹³ reports that *TET1* knockout hESCs cultured on mouse embryonic feeders (MEFs) show a reduced hPGCLC induction efficiency, while those cultured without MEFs (the same condition as in our experiments) are induced into hPGCLCs in an apparently normal manner, with no further analysis on the properties of hPGCLCs induced under the latter condition. Neither study addresses the role of *TET1* in subsequent hPGCLC differentiation. We touched on these points in the revised manuscript (“**Discussion**” section, 2nd paragraph).

4. Writing. Some parts of the manuscript, such as the Introduction, are clearly written. However, several other sections of the manuscript, including the sections “DNA methylome reprogramming”, “ER gene regulation and X-chromosome reactivation”, and “TET1 represses bivalent promoters and demethylates regulatory elements” are dense and data-laden, making them difficult to follow, especially for Nature’s general readership. For the “DNA methylome reprogramming” and “ER gene regulation and X-chromosome reactivation” sections of the Results, it would be helpful if the authors could consider including 1-2 sentences of background regarding X chromosome activation/inactivation and DNA methylation dynamics present in vivo before jumping into their in vitro results, to signal to the reader what is the in vivo gold standard that they are trying to recapitulate. Generally speaking, it would be helpful if the authors could consider making their entire manuscript easier to read.

Response 5. In response to the Referee’s specific comment, we have provided some brief background regarding DNA demethylation and X-chromosome inactivation/reactivation in the human germline in the “**DNA methylation reprogramming**” and “**ER gene regulation and XCR**” sections of the revised manuscript.

Furthermore, we have incorporated additional textual changes to the revised manuscript to try to improve its readability whenever possible. The entire manuscript was proofread by a professional

English editing service. We sincerely hope that with these changes it meets with the Referee's approval.

Minor comments:

1. Cell-type terminology: Multiple reviewers have asked about how the authors define "mitotic pro-spermatogonia or oogonia", which are the crux of the authors' claims. To help a general audience understand the various cell differentiation stages involved in this work (of which there are many, see "staircase plots" in Fig. 3A and Extended Data Fig. 6A), the authors could consider including in the main or supplementary figures a cartoon explaining (1) the various differentiation stages, (2) markers used to define each stage, and (3) which stage(s) are encompassed by their new differentiation system. There are many differentiation markers mentioned throughout this study, and a simple cartoon would help a general audience grasp the significance of each marker.

Response 6. In response to the Referee's suggestion, we provided a scheme that illustrates the requested three points in Fig. 1a in the revised manuscript. We sincerely hope this scheme adequately address the Referee's request.

2. "Here, we embarked on establishing a system for hPGCLC differentiation under a defined condition" (Introduction): The authors should consider replacing the word "defined" with another word. Most of their experiments include FBS. Additionally, all their experiments entail Advanced DMEM/F12, which usually includes bovine serum albumin, which is not defined and often contains many impurities (PMID 28238792).

Response 7. In response to the Referee's suggestion, we removed the term "defined," replacing it with the phrase "signalling molecule-driven" (the last sentence of **the introductory paragraphs**).

3. "Screening of signaling for hPGCLC differentiation" section (Results): Halfway through this paragraph, non-human cells are mentioned abruptly, which might be a bit hard for a general audience to follow. The authors should more directly explain in the beginning of the paragraph that mouse m220 feeder cells are employed as part of the culture system, as this is a potential limitation of the current method that should be clearly stated.

Response 8. In response to the Referee's suggestion, we now describe that we used m220 feeders in our culture system in the first sentence of the **"Screening of signaling for hPGCLC differentiation"** section in the revised manuscript.

4. Fig. 2F: In the figure legend, please kindly explain the statistical test that was performed.

Response 9. We thank the Referee for pointing this out. To avoid any confusion by the readers, we stated that “Both male DT⁺ cells and female VT⁺ cells expressed key ER genes at a high level” when citing Fig. 2f in the revised manuscript (the “**BMP signaling promotes hPGCLC differentiation**” section, 4th paragraph).

5. Cell expansion terminology: Throughout the Results section, the authors mention the degree of cell expansion observed in various conditions. Do they refer to total cell number, or alternatively, the number of marker-defined germ cells that expanded? Both are fine, but clarification would be helpful.

Response 10. To calculate the expansion of hPGCLC-derived cells, we used the sum of reporter/surface marker⁺ cells, i.e., the sum of BT⁺AG⁺ cells for M1-BTAG, the sum of AG⁺DT⁻/VT⁻, AG⁺DT⁺/VT⁺, and AG⁻DT⁺/VT⁺ cells for M1-AGDT/VT and F1/F2-AGVT, and the sum of EpCAM⁺/ITGB6⁺ cells for M2. We explained this point clearly in the “**Cell number count of hPGCLC-derived cells**” section of the **Methods** in the revised manuscript (“**BMP signaling promotes hPGCLC differentiation**” section, 2nd paragraph; the “**Cell number count of hPGCLC-derived cells**” section of the **Methods**).

6. Extended Data Fig. 2G: The authors should consider labeling the y-axis (“Fold change” and “Enrichment score”) with more details, to help the reader understand what is being measured.

Response 11. We thank the Referee for this suggestion. We have now labeled the y-axis as “BTAG-cell/reporter-positive fold change” and “BTAG-cell/reporter-positive enrichment score” in all relevant figures in the revised manuscript (Extended Data Fig. 1b–e, 2g, 4d, j, m, q, 10g).

7. “In response to BMP2, hPGCLC-derived cells maintained pluripotency/early PGC genes, and up-regulated ER genes and 13 previously reported genes that show up-regulation in gonadal germ cells 26” (Results): Extended Data Fig. 5A seems to show that only some of the signature genes reported by ref. 26 are upregulated, with genes like REC114 and SLC25A31 not being significantly expressed. If so, perhaps the authors should amend their comment to state “some of the 13 previously reported genes”?

Response 12. We thank the Referee for pointing this out. We amended the sentence as suggested by the Referee in the revised manuscript (“**Transcriptome dynamics**” section, 1st paragraph).

8. Extended Data Fig. 9J: What GFP is being used, is GFP supposed to be nuclearly-localized?

Response 13. We used enhanced GFP (EGFP), which was knocked in onto the 3' end of *TFAP2C* via a 2A peptide sequence so that EGFP was expressed under the control of the regulatory elements of *TFAP2C* (*TFAP2C-EGFP*: AG) (Extended Data Fig. 3a)¹⁴. The reason why AG appeared to be more concentrated in the nucleus is unclear. We explained this in the legend to Extended Data Fig. 9j.

9. Lines 386-397 of the main text, which talk about 5-mC and genomic titles, could be written more clearly.

Response 14. In response to the Referee's suggestion, we revised the suggested paragraph to improve its clarity.

10. Fig. 5B, Fig. 5D, Extended Data Fig. 11E, and Extended Data Fig. 11H: What samples (e.g., what conditions and genotypes or even datasets from what paper) are being shown in these UMAP plots should be more clearly annotated. At present, these figures are quite hard to follow.

Response 15. As the Referee recommended, we improved the labelling of the relevant figures to make the panels easier to understand (Fig. 5b, d, Extended Data Fig. 11e, h).

11. "Among the DEGs at *c42*, the up-regulated genes with bivalent promoters were large in number and highly enriched with the targets of *TET1* in hESCs (Extended Data Fig. 12f)" – Are *TET1* target genes in hESCs relevant to genomics analyses of PGCLCs? This is not an experimental request for the authors to perform *TET1* ChIP-seq in PGCLCs, but merits a question nevertheless.

Response 16. We thank the Referee for mentioning this. We consider that this would be useful information for the readers, since it suggests a possibility that, as shown between hESCs and neuronal precursors¹⁵, key targets of *TET1* can be shared between hPGCLCs and hESCs. We have now included a sentence in the revised manuscript to discuss this point (the "***TET1* deficiency leads to de-repression of bivalent genes and hypermethylation of regulatory elements**" section, 2nd paragraph).

12. Extended Data Fig. 6A: This is quite an impressive figure showing reasonable correspondence between germ cells *in vivo* and *in vitro*, at least with regard to the markers shown. However, some important genes like *ZGLP1* seem to be expressed significantly less *in vitro* than *in vivo*, but the heatmap color scale makes it difficult to ascertain the absolute difference in expression level. How differently are *ZGLP1* and other genes expressed *in vivo* and *in vitro*? And is there a way to graphically display that?

Response 17. We apologize if it was not clear, but Extended Data Fig. 6g shows the expression levels/expression-level differences of key genes potentially involved in the initiation of meiosis, including *ZGLP1* (PLL1 signature genes), between *in vivo* and *in vitro* cells in a readily accessible manner (lines 267–269 in the previous version). The PLL1 signature genes included *ZGLP1*, *STRA8*, *REC8*, and *SMC1B*, and exhibited strong up-regulation in PLL1 *in vivo* cells, but showed only moderate elevation in PLL1 or 2 *in vitro* cells (Extended Data Fig. 6g, h). In response to the Referee’s comment, we have also provided the exact expression values of the genes shown in Extended Data Fig. 6a in Source Data Extended Data Fig. 6. We hope these data adequately address the Referee’s request.

13. *“Fig. 3g: It might be helpful to include the cluster proportion percentage proportions somewhere, especially for key samples like c117. Response 22. We included the cluster proportion in Extended Data Fig. 6b, c in the revised manuscript” (Response to Reviewers): For certain key figures, including Extended Data Fig. 6B, readers may want to know the actual percentage numbers (not just a stacked histogram) of what percentage of the total population was comprised of a specific cell-type. This is especially important for data like Extended Data Fig. 6A,B wherein the authors report the generation of hitherto-difficult-to-produce developmentally advanced cells. Readers will want to know how efficient the process was.*

Response 18. In response to the Referee’s comments, we have provided the actual percentages of major clusters/culture days/weeks post-fertilization (wpf) within the histogram (Extended Data Fig. 6b, c). We have provided the full information in Source Data Extended Data Fig. 6.

14. *“We compared the genome-wide 5mC profiles in hPGCLCs reported in other studies 15,16 with those in hPGCLC-derived cells in this study and discussed the results in the revised manuscript” (Extended Data Fig. 8e, “DNA methylome reprogramming” section, 1st paragraph): This plot was difficult to follow. How do the cells produced by other differentiation protocols compare with the gold standard of actual germ cells? What about the total genome-wide level of methylation, expressed in a percentage, not just a correlation plot? Do the other protocols yield cells with global DNA demethylation?*

Response 19. The 5mC levels of cells reported by von Meyenn et al. and Kobayashi et al. were 57.9% and 61.4%, respectively^{16,17}, and did not reach the demethylation levels of mitotic pro-spermatogonia or oogonia *in vivo*. In response to the Referee’s comment, we revised this figure panel and added the above information (Extended Data Fig. 8e and its legend).

15. *“hPGCLC-derived cells repressed de novo DNA methyltransferases (DNMTs) (DNMT3L is repressed even without BMP)” (Results): The authors could clarify what “repress” means, does it reflect downregulation, or alternatively, repression of enzymatic activity?*

Response 20. In response to the Referee's comment, we added the following statement to clarify this point: "We observed that during BMP-driven hPGCLC specification and differentiation, hPGCLCs and their progeny down-regulated the expression of *de novo* DNA methyltransferases (DNMTs) (Fig. 3a, Extended Data Fig. 9i)" (the "**BMP signaling and DNMT activities**" section, 1st paragraph).

16. "We also analyzed the characteristics of DNA demethylation dynamics across genomic tiles with different 5mC levels" (Results): The paragraph starting with this topic sentence was difficult to follow, and should be rewritten.

Response 21. We revised the paragraph for clarity as recommended.

17. *BMP: It is also very interesting that the authors have discovered an important role for BMP signaling in driving epigenetic reprogramming in the germline. This may be somewhat reminiscent of the requirement for BMP to maintain naïve mouse pluripotent stem cells (PMID 14636556). As the authors are well aware, naïve pluripotent cells have a unique chromatin landscape that bears some similarities to PGCs. Maybe this point about BMP and naïve pluripotency would also be worth mentioning, if there is space in the manuscript?*

Response 22. In response to the Referee's comment, we added the following statement: "The finding that BMP signalling stabilizes the germ-cell fate is reminiscent of a known role of BMP signalling to sustain the self-renewal of mESCs via a blockade of their differentiation⁵¹" (the "**Discussion**" section, 1st paragraph).

18. *Precise maturation stage: It takes weeks to months for the authors' BMP-driven system to induce expression of DDX4/VASA or DAZL (Fig. 2). What is the identity of the cells prior to DDX4/VASA or DAZL expression? Do their cells express CDH5/VE-CADHERIN and DMRT1, which were previously reported to mark migratory PGCs prior to their lodgment in the gonads (PMID 37709822)?*

Response 23. The hPGC-to-mitotic pro-spermatogonia or oogonia differentiation *in vivo* also takes ~5–7 weeks¹⁸⁻²³. During hPGCLC-to-mitotic pro-spermatogonia or oogonia-like cell differentiation, both the transcriptome and DNA methylome show a progressive maturation (Fig. 3b, Fig. 4a). Therefore, we assume that cells prior to *DDX4/VASA* or *DAZL* expression are cells during such differentiation. They indeed expressed both *CDH5* and *DMRT1* (Extended Data Fig. 5b). We provided relevant data and statement in the revised manuscript (Extended Data Fig. 5b, the "**Transcriptome dynamics**" section, 1st paragraph).

19. For a general audience, it might be helpful for the authors to compare their present results to those reported by another recent study (PMID 37709822), the latter of which seemed to generate less mature PGCLCs.

Response 24. The most developmentally advanced cells reported by Irie et al. [DZ⁺ PGCLCs (*DAZL-tdTomato*-positive PGCLCs)] were induced by retinoic acid, Activin A, and overexpression of *SOX17* and *DMRT1*, and did express *DAZL*, but did not express *DDX4*, and showed genome-wide 5mC levels of ~76–79%²⁴ (Extended Data Fig. 8e). Therefore, these cells appeared to be a distinct entity from mitotic pro-spermatogonia/oogonia-like cells. We have now included this information in the revised manuscript (the “**Discussion**” section, 1st paragraph).

References

- 1 Williams, K. *et al.* TET1 and hydroxymethylcytosine in transcription and DNA methylation fidelity. *Nature* **473**, 343-348, doi:nature10066 [pii] 10.1038/nature10066 (2011).
- 2 Wu, H. *et al.* Dual functions of Tet1 in transcriptional regulation in mouse embryonic stem cells. *Nature* **473**, 389-393, doi:nature09934 [pii] 10.1038/nature09934 (2011).
- 3 Lu, F., Liu, Y., Jiang, L., Yamaguchi, S. & Zhang, Y. Role of Tet proteins in enhancer activity and telomere elongation. *Genes Dev* **28**, 2103-2119, doi:10.1101/gad.248005.114 (2014).
- 4 Tahiliani, M. *et al.* Conversion of 5-methylcytosine to 5-hydroxymethylcytosine in mammalian DNA by MLL partner TET1. *Science* **324**, 930-935, doi:1170116 [pii] 10.1126/science.1170116 (2009).
- 5 Rasmussen, K. D. & Helin, K. Role of TET enzymes in DNA methylation, development, and cancer. *Genes Dev* **30**, 733-750, doi:10.1101/gad.276568.115 (2016).
- 6 Wu, X. & Zhang, Y. TET-mediated active DNA demethylation: mechanism, function and beyond. *Nat Rev Genet* **18**, 517-534, doi:10.1038/nrg.2017.33 (2017).
- 7 Hill, P. W. S. *et al.* Epigenetic reprogramming enables the transition from primordial germ cell to gonocyte. *Nature* **555**, 392-396, doi:10.1038/nature25964 (2018).
- 8 Hackett, J. A. *et al.* Synergistic Mechanisms of DNA Demethylation during Transition to Ground-State Pluripotency. *Stem cell reports* **1**, 518-531, doi:10.1016/j.stemcr.2013.11.010 (2013).
- 9 Vincent, J. J. *et al.* Stage-specific roles for tet1 and tet2 in DNA demethylation in primordial germ cells. *Cell Stem Cell* **12**, 470-478, doi:10.1016/j.stem.2013.01.016 (2013).
- 10 Yamaguchi, S. *et al.* Tet1 controls meiosis by regulating meiotic gene expression. *Nature* **492**, 443-447, doi:10.1038/nature11709 (2012).
- 11 Yamaguchi, S., Shen, L., Liu, Y., Sendler, D. & Zhang, Y. Role of Tet1 in erasure of genomic imprinting. *Nature* **504**, 460-464, doi:10.1038/nature12805 (2013).
- 12 Li, Z. *et al.* The balance between NANOG and SOX17 mediated by TET proteins regulates specification of human primordial germ cell fate. *Cell Biosci* **12**, 181, doi:10.1186/s13578-022-00917-0 (2022).
- 13 Hsu, F. M. *et al.* TET1 facilitates specification of early human lineages including germ cells. *iScience* **26**, 107191, doi:10.1016/j.isci.2023.107191 (2023).
- 14 Sasaki, K. *et al.* Robust In Vitro Induction of Human Germ Cell Fate from Pluripotent Stem Cells. *Cell Stem Cell* **17**, 178-194, doi:10.1016/j.stem.2015.06.014 (2015).
- 15 Verma, N. *et al.* TET proteins safeguard bivalent promoters from de novo methylation in human embryonic stem cells. *Nat Genet* **50**, 83-95, doi:10.1038/s41588-017-0002-y (2018).
- 16 von Meyenn, F. *et al.* Comparative Principles of DNA Methylation Reprogramming during Human and Mouse In Vitro Primordial Germ Cell Specification. *Dev Cell* **39**, 104-115, doi:10.1016/j.devcel.2016.09.015 (2016).
- 17 Kobayashi, M. *et al.* Expanding homogeneous culture of human primordial germ cell-

- like cells maintaining germline features without serum or feeder layers. *Stem cell reports* **17**, 507-521, doi:10.1016/j.stemcr.2022.01.012 (2022).
- 18 Baker, T. G. A Quantitative and Cytological Study of Germ Cells in Human Ovaries. *Proc R Soc Lond B Biol Sci* **158**, 417-433 (1963).
- 19 Fukuda, T., Hedinger, C. & Groscurth, P. Ultrastructure of developing germ cells in the fetal human testis. *Cell Tissue Res* **161**, 55-70 (1975).
- 20 Gkoutela, S. *et al.* DNA Demethylation Dynamics in the Human Prenatal Germline. *Cell* **161**, 1425-1436, doi:10.1016/j.cell.2015.05.012 (2015).
- 21 Guo, F. *et al.* The Transcriptome and DNA Methylome Landscapes of Human Primordial Germ Cells. *Cell* **161**, 1437-1452, doi:10.1016/j.cell.2015.05.015 (2015).
- 22 Tang, W. W. *et al.* A Unique Gene Regulatory Network Resets the Human Germline Epigenome for Development. *Cell* **161**, 1453-1467, doi:10.1016/j.cell.2015.04.053 (2015).
- 23 Li, L. *et al.* Single-Cell RNA-Seq Analysis Maps Development of Human Germline Cells and Gonadal Niche Interactions. *Cell Stem Cell* **20**, 858-873 e854, doi:10.1016/j.stem.2017.03.007 (2017).
- 24 Irie, N. *et al.* DMRT1 regulates human germline commitment. *Nat Cell Biol* **25**, 1439-1452, doi:10.1038/s41556-023-01224-7 (2023).

Reviewer Reports on the Second Revision:

Referees' comments:

Referee #1 (Remarks to the Author):

The authors addressed my remaining points in a suitable manner. I recommend publication of the manuscript in its current form.

Referee #2 (Remarks to the Author):

The revision of the manuscript now fully addresses my concerns and I support its publication.

Referee #3 (Remarks to the Author):

In this revised manuscript, the authors have satisfactorily addressed all remaining comments. They have appropriately toned down their mechanistic claims regarding BMP-induced repression of DNA methylation machinery and BMP-induced MAPK/ERK pathway attenuation. The authors should be congratulated on their study, which significantly raises the bar with regard to the quantity and maturity of human germ cells that can be generated in vitro from hPSCs.

Author Rebuttals to Second Revision:

Rebuttal: *Nature* manuscript 2023-15-08623B

Referees' comments:

Referee #1 (Remarks to the Author):

The authors addressed my remaining points in a suitable manner. I recommend publication of the manuscript in its current form.

Referee #2 (Remarks to the Author):

The revision of the manuscript now fully addresses my concerns and I support its publication.

Referee #3 (Remarks to the Author):

In this revised manuscript, the authors have satisfactorily addressed all remaining comments. They have appropriately toned down their mechanistic claims regarding BMP-induced repression of DNA methylation machinery and BMP-induced MAPK/ERK pathway attenuation. The authors should be congratulated on their study, which significantly raises the bar with regard to the quantity and maturity of human germ cells that can be generated in vitro from hPSCs.

Response. We would like to sincerely thank the Referees for their constructive and encouraging comments throughout the review process.